# Triangulation supports agricultural spread of the Transeurasian languages

Martine Robbeets[1✉], Remco Bouckaert[1,2], Matthew Conte[3], Alexander Savelyev[1,4], Tao Li[1,5,6], Deog-Im An[7], Ken-ichi Shinoda[8], Yinqiu Cui[9,10], Takamune Kawashima[11], Geonyoung Kim[3], Junzo Uchiyama[12,13], Joanna Dolińska[1], Sofia Oskolskaya[1,14], Ken-Yōjiro Yamano[15], Noriko Seguchi[16,17], Hirotaka Tomita[18,19], Hiroto Takamiya[20], Hideaki Kanzawa-Kiriyama[8], Hiroki Oota[21], Hajime Ishida[22], Ryosuke Kimura[22], Takehiro Sato[23], Jae-Hyun Kim[24], Bingcong Deng[1], Rasmus Bjørn[1], Seongha Rhee[25], Kyou-Dong Ahn[25], Ilya Gruntov[4,26], Olga Mazo[4,26], John R. Bentley[27], Ricardo Fernandes[1,28,29], Patrick Roberts[1], Ilona R. Bausch[12,30,31], Linda Gilaizeau[1], Minoru Yoneda[32], Mitsugu Kugai[33], Raffaela A. Bianco[1], Fan Zhang[9], Marie Himmel[1], Mark J. Hudson[1,34✉] & Chao Ning[1,35✉]

The origin and early dispersal of speakers of Transeurasian languages—that is, Japanese, Korean, Tungusic, Mongolic and Turkic—is among the most disputed issues of Eurasian population history[1–3]. A key problem is the relationship between linguistic dispersals, agricultural expansions and population movements[4,5]. Here we address this question by 'triangulating' genetics, archaeology and linguistics in a unified perspective. We report wide-ranging datasets from these disciplines, including a comprehensive Transeurasian agropastoral and basic vocabulary; an archaeological database of 255 Neolithic–Bronze Age sites from Northeast Asia; and a collection of ancient genomes from Korea, the Ryukyu islands and early cereal farmers in Japan, complementing previously published genomes from East Asia. Challenging the traditional 'pastoralist hypothesis'[6–8], we show that the common ancestry and primary dispersals of Transeurasian languages can be traced back to the first farmers moving across Northeast Asia from the Early Neolithic onwards, but that this shared heritage has been masked by extensive cultural interaction since the Bronze Age. As well as marking considerable progress in the three individual disciplines, by combining their converging evidence we show that the early spread of Transeurasian speakers was driven by agriculture.

Recent breakthroughs in ancient DNA sequencing have made us rethink the connections between human, linguistic and cultural expansions across Eurasia. Compared to western Eurasia[9–11], however, eastern Eurasia remains poorly understood. Northeast Asia—the vast region encompassing Inner Mongolia, the Yellow, Liao and Amur River basins, the Russian Far East, the Korean peninsula and the Japanese Islands—remains especially under-represented in the recent literature. With a few exceptions that are heavily focused on genetics[12–14] or limited to reviewing existing datasets[4], truly interdisciplinary approaches to Northeast Asia are scarce.

The linguistic relatedness of the Transeurasian languages—also known as 'Altaic'—is among the most disputed issues in linguistic prehistory. Transeurasian denotes a large group of geographically adjacent languages stretching across Europe and northern Asia, and includes five uncontroversial linguistic families: Japonic, Koreanic, Tungusic, Mongolic, and Turkic (Fig. 1a). The question of whether

[1]Max Planck Institute for the Science of Human History, Jena, Germany. [2]Centre of Computational Evolution, University of Auckland, Auckland, New Zealand. [3]Department of Archaeology and Art History, Seoul National University, Seoul, South Korea. [4]Institute of Linguistics, Russian Academy of Sciences, Moscow, Russia. [5]Department of Archaeology, Wuhan University, Wuhan, China. [6]Archaeological Institute for Yangtze Civilization (AIYC), Wuhan University, Wuhan, China. [7]Department of Conservation of Cultural Heritage, Hanseo University, Seosan, Korea. [8]Department of Anthropology, National Museum of Nature and Science, Tsukuba, Japan. [9]School of Life Sciences, Jilin University, Changchun, China. [10]Research Center for Chinese Frontier Archaeology of Jilin University, Jilin University, Changchun, China. [11]Hiroshima University Museum, Higashi-Hiroshima, Japan. [12]Sainsbury Institute for the Study of Japanese Arts and Cultures, Norwich, UK. [13]Center for Cultural Resource Studies, Kanazawa University, Kanazawa, Japan. [14]Institute for Linguistic Studies, Russian Academy of Sciences, Saint Petersburg, Russia. [15]Research Center for Buried Cultural Properties, Kumamoto University, Kumamoto, Japan. [16]Department of Environmental Changes, Faculty of Social and Cultural Studies, Kyushu University, Fukuoka, Japan. [17]Department of Anthropology, The University of Montana, Missoula, MT, USA. [18]Hokkaido Government Board of Education, Sapporo, Japan. [19]Graduate School of Integrated Sciences of Global Society, Kyushu University, Fukuoka, Japan. [20]Research Center for the Pacific Islands, Kagoshima University, Kagoshima, Japan. [21]Department of Biological Sciences, Graduate School of Science, The University of Tokyo, Tokyo, Japan. [22]Graduate School of Medicine, University of the Ryukyus, Nishihara, Japan. [23]Department of Bioinformatics and Genomics, Graduate School of Medical Sciences, Kanazawa University, Kanazawa, Japan. [24]Department of Archaeology and Art History, Donga University, Busan, South Korea. [25]Hankuk University of Foreign Studies, Seoul, South Korea. [26]National Research University Higher School of Economics, Moscow, Russia. [27]Department of World Languages and Cultures, Northern Illinois University, DeKalb, IL, USA. [28]Faculty of Arts, Masaryk University, Brno, Czech Republic. [29]School of Archaeology, University of Oxford, Oxford, UK. [30]Leiden University Institute of Area Studies, Leiden, The Netherlands. [31]Kokugakuin University Museum, Tokyo, Japan. [32]University Museum, University of Tokyo, Tokyo, Japan. [33]Miyakojima City Board of Education, Miyakojima, Japan. [34]Institut d'Asie Orientale, ENS de Lyon, Lyon, France. [35]School of Archaeology and Museology, Peking University, Beijing, China. ✉e-mail: robbeets@shh.mpg.de; hudson@shh.mpg.de; ning@shh.mpg.de

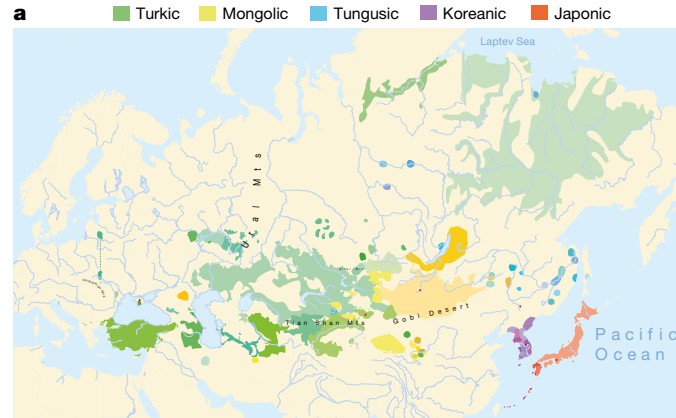

**a** [Legend:] Turkic | Mongolic | Tungusic | Koreanic | Japonic

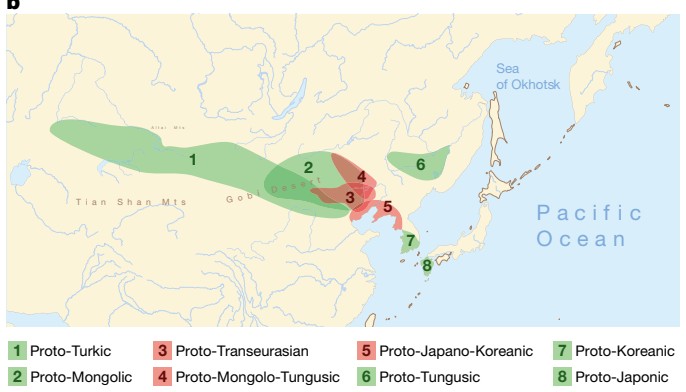

**b**

| | | |
|---|---|---|
| 1 Proto-Turkic | 3 Proto-Transeurasian | 5 Proto-Japano-Koreanic | 7 Proto-Koreanic |
| 2 Proto-Mongolic | 4 Proto-Mongolo-Tungusic | 6 Proto-Tungusic | 8 Proto-Japonic |

**Fig. 1 | Distribution of Transeurasian languages in the past and in the present. a**, Geographical distribution of the 98 Transeurasian language varieties included in this study. Contemporary languages are represented by coloured surfaces, historical varieties by red dots. For legend, see Extended Data Fig. 1. **b**, Reconstructed locations of Transeurasian ancestral languages spoken during the Neolithic (red) and the Bronze Age and later (green). For detailed homeland detection, see Supplementary Data 4. The estimated time-depth is based on Bayesian inference presented in Supplementary Data 24.

these five groups descend from a single common ancestor has been the topic of a long-standing debate between supporters of inheritance and borrowing. Recent assessments show that even if many common properties between these languages are indeed due to borrowing[15–17], there is nonetheless a core of reliable evidence for the classification of Transeurasian as a valid genealogical group[1,2,18,19].

Accepting this classification, however, gives rise to new questions about the time depth, location, cultural identity and dispersal routes of ancestral Transeurasian speech communities. Here we challenge the traditional 'pastoralist hypothesis' that identifies the primary dispersals of the Transeurasian languages with nomadic expansions starting in the eastern steppe in the fourth millennium before present (BP)[6–8], by proposing a 'farming hypothesis', which places those dispersals within the scope of the 'farming/language dispersal hypothesis'[5,20,21]. As these issues reach far beyond linguistics, we address them by integrating archaeology and genetics in a single approach termed 'triangulation'.

## Linguistics

We collected a new dataset of 3,193 cognate sets that represent 254 basic vocabulary concepts for 98 Transeurasian languages, including dialects and historical varieties (Supplementary Data 1). We applied Bayesian methods to infer a dated phylogeny of the Transeurasian languages (Supplementary Data 24). Our results indicate a time-depth

of 9181 BP (5595–12793 95% highest probability density (95% HPD)) for the Proto-Transeurasian root of the family; 6811 BP (4404–10166 95% HPD) for Proto-Altaic, the unity of Turkic, Mongolic and Tungusic languages; 4491 BP (2599–6373 95% HPD) for Mongolo-Tungusic; and 5458 BP (3335–8024 95% HPD) for Japano-Koreanic (Fig. 1b). These dates estimate the time-depth of the initial break-up of a given language family into more than one foundational subgroup.

We used our lexical dataset to model the expansion of Transeurasian languages in space (Supplementary Data 3, 4). We applied Bayesian phylogeography to complement classical approaches, such as lexicostatistics, the diversity hotspot principle and cultural reconstruction[1–3,8].

In contrast to previously proposed homelands, which range from the Altai[6–8] to the Yellow River[22] to the Greater Khingan Mountains[23] to the Amur basin[24], we find support for a Transeurasian origin in the West Liao River region in the Early Neolithic. After a primary break-up of the family in the Neolithic, further dispersals took place in the Late Neolithic and Bronze Age. The ancestor of the Mongolic languages expanded northwards to the Mongolian Plateau, Proto-Turkic moved westwards over the eastern steppe and the other branches moved eastwards: Proto-Tungusic to the Amur–Ussuri–Khanka region, Proto-Koreanic to the Korean Peninsula and Proto-Japonic over Korea to the Japanese islands (Fig. 1b).

Through a qualitative analysis in which we examined agropastoral words that were revealed in the reconstructed vocabulary of the proto-languages (Supplementary Data 5), we further identified items that are culturally diagnostic for ancestral speech communities in a particular region at a particular time. Common ancestral languages that separated in the Neolithic, such as Proto-Transeurasian, Proto-Altaic, Proto-Mongolo-Tungusic and Proto-Japano-Koreanic, reflect a small core of inherited words that relate to cultivation ('field', 'sow', 'plant', 'grow', 'cultivate', 'spade'); millets but not rice or other crops ('millet seed', 'millet gruel', 'barnyard millet'); food production and preservation ('ferment', 'grind', 'crush to pulp', 'brew'); wild foods suggestive of sedentism ('walnut', 'acorn', 'chestnut'); textile production ('sew', 'weave cloth', 'weave with a loom', 'spin', 'cut cloth', 'ramie', 'hemp'); and pigs and dogs as the only domesticated animals.

By contrast, individual subfamilies that separated in the Bronze Age, such as Turkic, Mongolic, Tungusic, Koreanic and Japonic, inserted new subsistence terms that relate to the cultivation of rice, wheat and barley; dairying; domesticated animals such as cattle, sheep and horses; farming or kitchen tools; and textiles such as silk (Supplementary Data 5). These words are borrowings that result from linguistic interaction between Bronze Age populations speaking various Transeurasian and non-Transeurasian languages.

In summary, the age, homeland, original agricultural vocabulary and contact profile of the Transeurasian family support the farming hypothesis and exclude the pastoralist hypothesis (Supplementary Data 5).

## Archaeology

Although Neolithic Northeast Asia was characterized by widespread plant cultivation[25], cereal farming expanded from several centres of domestication, the most important of which for Transeurasian was the West Liao basin, where cultivation of broomcorn millet started by 9000 BP[26–29]. Extracting data from the published literature, we scored 172 archaeological features for 255 Neolithic and Bronze Age sites (Supplementary Data 6, Fig. 2a) and compiled an inventory of 269 directly carbon-14-dated early crop remains (Supplementary Data 9) in northern China, the Primorye, Korea and Japan.

The main results of our Bayesian analysis (Supplementary Data 25), which clusters the 255 sites according to cultural similarity, are visualized in Fig. 2b. We find a cluster of Neolithic cultures in the West Liao basin, from which two branches associated with millet farming separate: a Korean Chulmun branch and a branch of Neolithic cultures covering the Amur, Primorye and Liaodong. This confirms previous

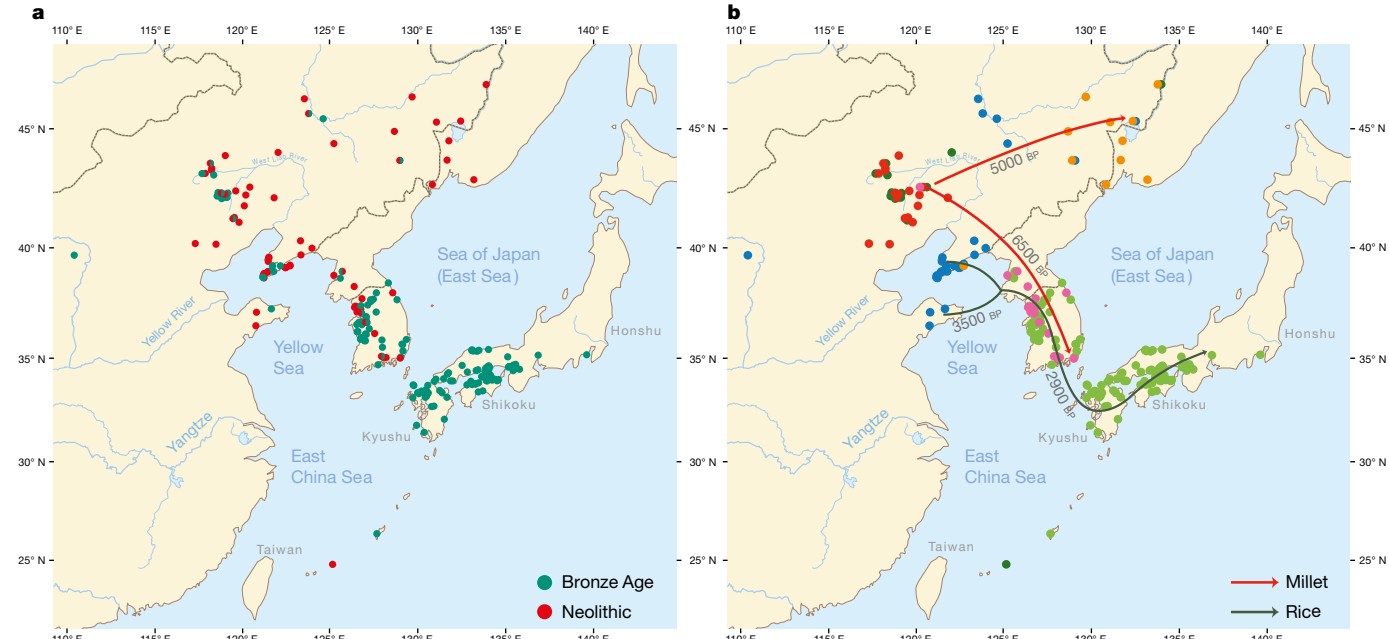

**Fig. 2 | Spatiotemporal distribution and clustering of sites included in the archaeological database. a**, Geographical distribution of 255 sites from the Neolithic (red) and the Bronze Age (green). **b**, Coloured dots cluster the investigated sites according to cultural similarity in line with Bayesian analysis in Supplementary Data 25, with indication of the spread of millet and rice in time and space. The distribution of archaeological sites in Fig. 2 is smaller than that of contemporary languages in Fig. 1 because we focus on the early dispersal of the linguistic subgroups in the Neolithic and the Bronze Age and on the links between the eastward spread of farming and language dispersal.

findings about the dispersal of millet agriculture to Korea by 5500 BP and via the Amur to the Primorye by 5000 BP[30,31].

Our analysis further clusters Bronze Age sites in the West Liao area with Mumun sites in Korea and Yayoi sites in Japan. This mirrors how during the fourth millennium BP, the agricultural package of the Liaodong–Shandong area was supplemented with rice and wheat. These crops were transmitted to the Korean Peninsula by the Early Bronze Age (3300–2800 BP) and from there to Japan after 3000 BP (Fig. 2b).

Although population movements were not linked with monothetic archaeological cultures, Neolithic farming expansions in Northeast Asia were associated with some diagnostic features, such as stone tools for cultivation and harvesting and textile technology[32] (Supplementary Data 7). Domesticated animals and dairying had an important role in the spread of the Neolithic in western Eurasia but, except for dogs and pigs, our database shows little evidence for animal domestication in Northeast Asia before the Bronze Age (Supplementary Data 6). The link between agriculture and population migrations is especially clear from similarities between ceramics, stone tools, and domestic and burial architecture between Korea and western Japan[33].

Building on previous studies, we provide an overview of demographic changes associated with the introduction of millet farming across the regions in our study (Extended Data Fig. 3). Having invested in elaborate paddy fields, wet rice farmers tended to stay in one place, absorbing population growth through extra labour, whereas millet farmers typically adopted a more expansionary settlement pattern[34]. Neolithic population densities increased across Northeast Asia before a population crash in the Late Neolithic[35,36]. The Bronze Age then saw exponential population increases in China, Korea and Japan.

## Genetics

We report genomic analyses of 19 authenticated ancient individuals from the Amur, Korea, Kyushu and the Ryukyus and combined them with published genomes that cover the eastern steppe, West Liao, Amur and Yellow River regions, Liaodong, Shandong, the Primorye

and Japan between 9500 and 300 BP (Fig. 3a, Extended Data Fig. 4, Supplementary Data 11, 13, 17). We projected them onto a principal component analysis (PCA) of 149 present-day Eurasian populations and 45 East Asian populations (Extended Data Figs. 5–8). Figure 3b models our key ancient populations as an admixture of five genetic components, whereby Jalainur represents Amur, Yangshao the Yellow River and Rokutsu the Jomon genome, whereas Hongshan and Upper Xiajiadian in the West Liao River are composed of Yellow River and Amur genomes (qpAdm admixture of various East Asian genetic components in Supplementary Data 16).

Contemporary Tungusic as well as Nivkh speakers in the Amur form a tight cluster[13] (Extended Data Fig. 5). Neolithic hunter-gatherers from Baikal, Primorye and the southeastern steppe, as well as farmers from the West Liao and Amur, all project within this cluster (Extended Data Figs. 8–10).

Late Neolithic Angangxi (Supplementary Data 12) show a high proportion of Amur-like ancestry, whereas West Liao Neolithic millet farmers show a considerable proportion of Amur-like ancestry with a gradual shift towards the Yellow River genome over time[12] (Extended Data Figs. 8–10, Fig. 3b). Although we lack Early Neolithic genomes in the West Liao River, Amur-like ancestry thus is likely to represent the original genetic profile of indigenous pre-Neolithic (or late Palaeolithic) hunter-gatherers covering Baikal, Amur, Primorye, the southeastern steppe and West Liao, continuing in the early farmers from this region. This contradicts a recent genetic study[13], which concludes that the absence of Yellow River influence in ancient genomes from Mongolia and the Amur does not support the West Liao genetic correlate of the Transeurasian language family.

The PCA (Extended Data Figs. 8–10) shows a general trend for Neolithic individuals from Mongolia to contain high Amur-like ancestry with extensive gene flow from western Eurasia increasing from the Bronze to Middle Ages[37]. Whereas the Turkic-speaking Xiongnu[38], Old Uyghur and Türk are extremely scattered, the Mongolic-speaking[39] Iron Age Xianbei fall closer to the Amur cluster than the Shiwei, Rouran, Khitan and Middle Mongolian Khanate from Antiquity and the Middle Ages.

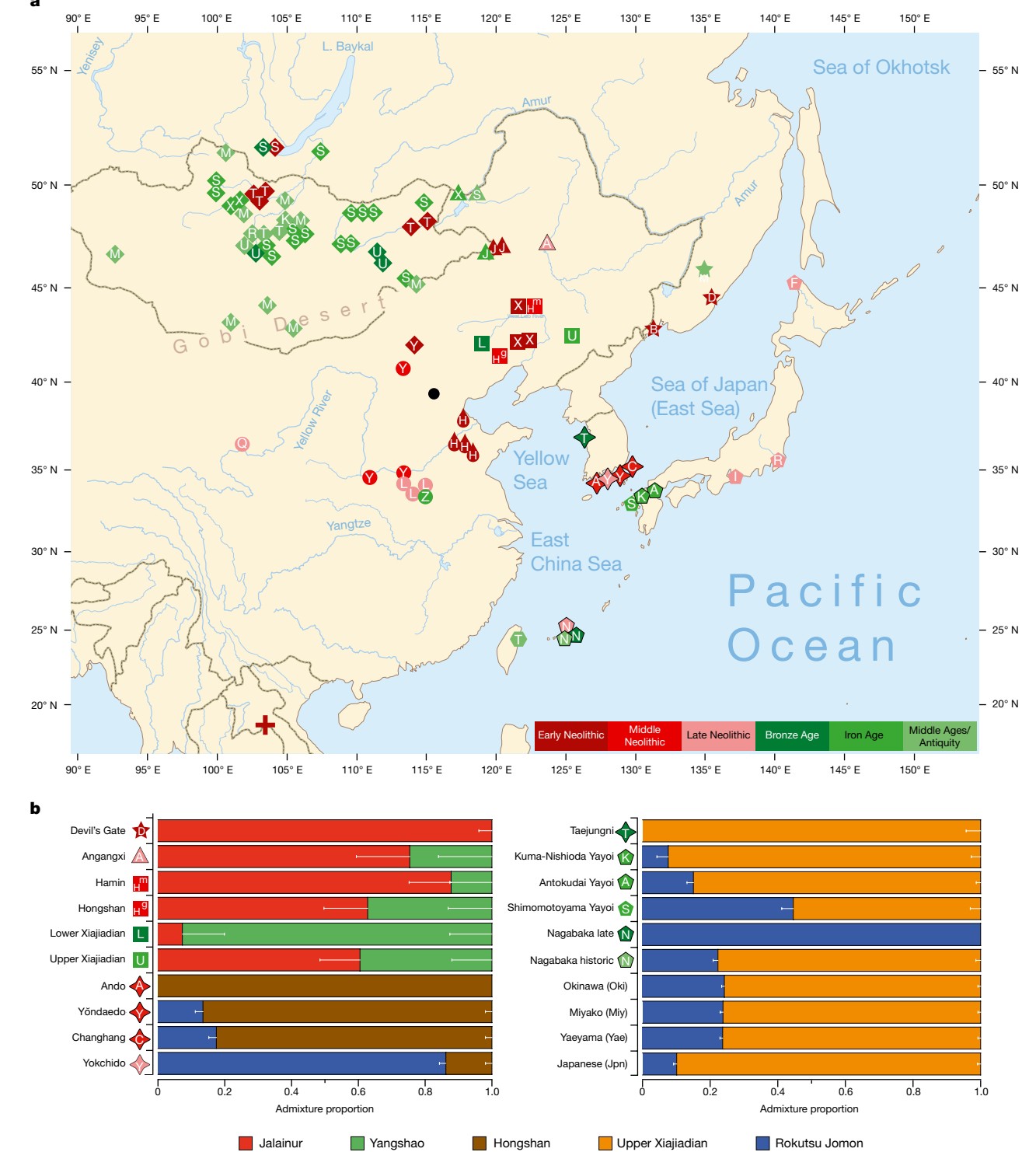

**Fig. 3 | Spatiotemporal distribution and admixture of ancient genomes.**
**a**, Ancient genomes located in time and space. For detailed legend, see Extended Data Fig. 4. **b**, QpAdm proximal admixture modelling of 20 key ancient populations from this study. The x axis shows ancestry proportion estimates for the target populations in the y axis; the error bars represent ±1 s.e.m. range, estimated by 5-cM block jackknifing.

As Amur-related ancestry can be traced down to speakers of Japanese and Korean[13], it appears to be the original genetic component common to all speakers of Transeurasian languages. By analysing ancient genomes from Korea (Supplementary Data 12), we find that Jomon ancestry was present on the Peninsula by 6000 BP (Fig. 3b, Supplementary Data 13).

The proximal qpAdm modelling (Supplementary Data 13) suggests that Neolithic Ando can be entirely derived from an ancestry related

to Hongshan, whereas Yŏndaedo and Changhang can be modelled as an admixture of Jomon with a high proportion of Hongshan ancestry, although Yŏndaedo has only limited resolution (Supplementary Data 16, Fig. 3b). Yokchido, on the southern coast of Korea, contains nearly 95% Jomon ancestry. Although our genetic analysis cannot itself distinguish between possible East Asian ancestries for Bronze Age Taejungni, given the Bronze Age date it can be best modelled as Upper Xiajiadian; a possible minor Jomon admixture is not statistically

significant (*P* = 0.228; Supplementary Data 16). We therefore observe a heterogeneous presence of Jomon ancestry in Neolithic Koreans (0–95%) and its eventual disappearance over time, as shown by a negligible Jomon contribution to present-day Koreans. The lack of a significant Jomon component in Taejungni indicates that early populations, without detectable Jomon ancestry linked to present-day Koreans, migrated to the Korean peninsula in association with rice farming, and replaced Neolithic populations with some Jomon admixture—although our genetic data currently do not have resolution to test this hypothesis, owing to limited sample size and coverage. We therefore associate the spread of farming to Korea with different waves of Amur and Yellow River gene flow, modelled by Hongshan for the Neolithic introduction of millet farming and by Upper Xiajiadian for the Bronze Age addition of rice agriculture.

Analysing the genomes from Yayoi farmers (Supplementary Data 12), we found that, like Taejungni, they can be modelled as indigenous Jomon ancestry admixed with Bronze Age Upper Xiajiadian ancestry. Our results support massive migration from Korea into Japan in the Bronze Age.

The Nagabaka genomes from Miyako Island (Supplementary Data 12) represent the first—to our knowledge—ancient genome-wide data from the Ryukyus. Contrary to previous findings that Holocene populations reached the southern Ryukyus from Taiwan[40], our results suggest that the prehistoric Nagabaka population originated in Jomon cultures to the north (Extended Data Fig. 7). The genetic turn-over from Jomon- to Yayoi-like ancestry before the early modern period mirrors the late arrival of agriculture and Ryukyan languages in this region.

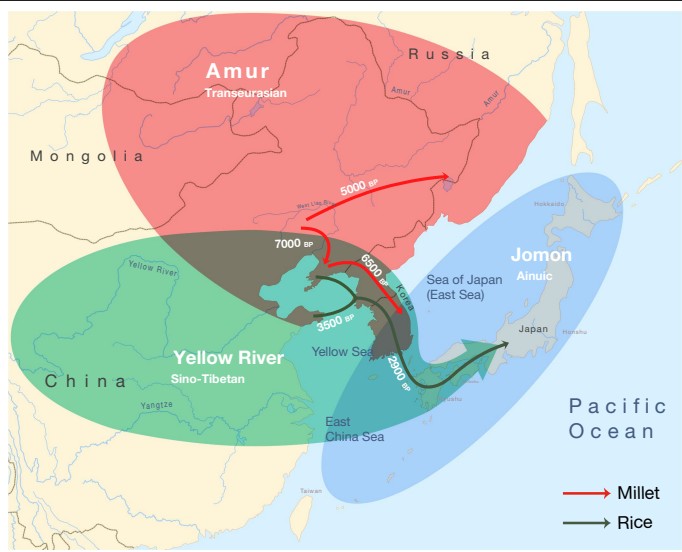

**Fig. 4 | Integration of linguistic, agricultural and genetic expansions in Northeast Asia.** Amur ancestry is marked in red, Yellow River ancestry in green and Jomon ancestry in blue. The red arrows show the eastward migrations of millet farmers in the Neolithic, bringing Koreanic and Tungusic languages to the indicated regions. The green arrows mark the integration of rice agriculture in the Late Neolithic and the Bronze Age, bringing the Japonic language over Korea to Japan.

## Discussion

Triangulation of linguistic, archaeological and genetic evidence shows that the origins of the Transeurasian languages can be traced back to the beginning of millet cultivation and the early Amur gene pool in Neolithic Northeast Asia. The spread of these languages involved two major phases that mirror the dispersal of agriculture and genes (Fig. 4). The first phase, represented by the primary splits in the Transeurasian family, goes back to the Early–Middle Neolithic, when millet farmers associated with Amur-related genes spread from the West Liao River to contiguous regions. The second phase, represented by linguistic contacts between the five daughter branches, goes back to the Late Neolithic, Bronze and Iron Ages, when millet farmers with substantial Amur ancestry gradually admixed with Yellow River, western Eurasian and Jomon populations and added rice, west Eurasian crops and pastoralism to the agricultural package.

Bringing together the spatiotemporal and subsistence patterns, we find clear links between the three disciplines (Supplementary Data 26). The onset of millet cultivation in the West Liao region around the ninth millennium BP can be associated with substantial Amur-related ancestry and overlaps in time and space with the ancestral Transeurasian speech community. In line with recent associations between the Sino-Tibetan family estimated at 8000 BP[41,42] and Neolithic farmers from the Upper and Middle Yellow River[13,14], our results associate the two centres of millet domestication in Northeast Asia with the origins of two major language families: Sino-Tibetan on the Yellow River and Transeurasian on the West Liao River. The lack of evidence for Yellow River influence in the ancestral Transeurasian language and genes is consistent with the multi-centric origins of millet cultivation suggested in archaeobotany[28].

The early stages of millet domestication in the ninth to seventh millennia BP are accompanied by population growth (Extended Data Fig. 3), leading to the formation of environmentally or socially separated subgroups in the West Liao region and broken connectivity between speakers of Altaic and Japano-Koreanic.

Around the mid-sixth millennium BP, some of these farmers started to migrate eastwards, around the Yellow Sea into Korea and northeast into the Primorye, bringing Koreanic and Tungusic languages to these regions and bringing from the West Liao region additional Amur ancestries to the Primorye and mixed Amur–Yellow River ancestries to Korea. Our newly analysed Korean genomes are notable in that they testify to the presence of and admixture with Jomon-related ancestries outside Japan.

The Late Bronze Age saw extensive cultural exchange across the Eurasian steppe, which resulted in the admixture of populations from the West Liao region and the Eastern steppe with western Eurasian genetic lineages. Linguistically, this interaction is mirrored in the borrowing of agropastoral vocabulary by Proto-Mongolic and Proto-Turkic speakers, especially relating to wheat and barley cultivation, herding, dairying and horse exploitation.

Around 3300 BP, farmers from the Liaodong–Shandong area migrated to the Korean peninsula, adding rice, barley and wheat to millet agriculture. This migration aligns with the genetic component modelled as Upper Xiajiadian in our Bronze Age sample from Korea and is reflected in early borrowings between Japonic and Koreanic languages. Archaeologically it can be associated with agriculture in the larger Liaodong–Shandong area without being specifically restricted to Upper Xiadiajian material culture.

In the third millennium BP, this agricultural package was transmitted to Kyushu, triggering a transition to full-scale farming, a genetic turn-over from Jomon to Yayoi ancestry and a linguistic shift to Japonic. By adding unique samples from Nagabaka in the southern Ryukyus, we traced the farming/language dispersal to the edge of the Transeurasian world. Demonstrating that Jomon ancestry stretched as far south as Miyako Island, our results contradict previous assumptions of a northward expansion by Austronesian populations from Taiwan. Together with the Jomon profile discovered at Yokchido in Korea, our results show that Jomon genomes and material culture did not always overlap.

By advancing new evidence from ancient DNA, our research thus confirms recent findings that Japanese and Korean populations have West Liao River ancestry, whereas it contradicts previous claims that there is no genetic correlate of the Transeurasian language family[13].

Although some previous research regarded the Transeurasian zone as beyond the area suitable for farming[20], our research confirms that the farming/language dispersal hypothesis remains an important model

for understanding Eurasian population dispersals[21]. Triangulation of linguistics, archaeology and genetics resolves the competition between the pastoralist and farming hypotheses and concludes that the early spread of Transeurasian speakers was driven by agriculture.

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

## Methods

### Linguistics

**Bayesian phylogenetics.** Combining dictionary search with fieldwork, we collected a comparative dataset including 3,193 datapoints representing 254 basic vocabulary concepts for 98 Transeurasian languages, including contemporary and historical varieties (Supplementary Data 1). These concepts are based on a merger of the Leipzig–Jakarta 200 (ref. [43]) and Jena 200 (ref. [44]) lists (Supplementary Data 2). The Turkic and Tungusic basic vocabulary included is based on a revision of recently published datasets[45,46]. Cognate coding is supported by an inventory of basic vocabulary etymologies and sound correspondences across the Transeurasian languages presented in Supplementary Data 2.

We performed a Bayesian phylogenetic analysis with cognates encoded as binary data[47]. Because the data were collected such that at least one cognate was present, the data were ascertained to not contain any sites having all zeros. Ascertainment correction was applied to cater for this[47].

We considered the following substitution models, which govern the evolutionary process of cognates along branches of a tree: continuous time Markov chain (CTMC), which assumes a constant rate of mutations; covarion, which assumes a slow and fast rate and the model switching between these two states; and the pseudo Dollo covarion model, which is based on the Dollo principle that a cognate can only appear once, but can be lost many times. Detailed descriptions of the CTMC and covarion models[47] and the pseudo Dollo covarion model[48] are available in the literature. For all models, we assume that each meaning class has its own relative rate to capture the variation between rates of evolution of different words.

Although language evolves on average at a constant rate, we find that there can be considerable variation in rates between branches on a tree[47,48]. Such variation can be captured using the uncorrelated relaxed clock[49], assuming rates are log-normally distributed.

A birth death model is used to describe the generative process of language creation. As the data contain ancient languages that may be ancestral to current languages, we allow the tree to have ancestral nodes. A fossilized birth death model[50], which allows such ancestral nodes, is used as prior on the tree. Language family node ages were informed by age priors (Japonic 2100 BP ± 175, Koreanic 800 BP ± 175, Turkic 2100 BP ± 175, Mongolic 750 BP ± 50, Tungusic 1900 BP ± 275). These calibrations are supported by chronological estimations proposed in linguistic literature (Supplementary Data 18). We found that these node age priors helped to reduce uncertainty slightly in the root age distribution.

We compared the fit of different models by estimating the marginal likelihoods using nested sampling[51] (Supplementary Data 18), and conclude that the pseudo Dollo covarion model with a relaxed clock has the best fit, and covarion with relaxed clock the next best fit. Both models produce compatible time estimates, though covarion estimates tend to have larger uncertainty (that is, have larger 95% HPD intervals). Time estimates of the CTMC model with relaxed clock are still compatible but even wider, and tend to have a higher mean.

All posterior estimates were performed using BEAST v.2.6[52] using adaptive coupled Markov chain Monte Carlo (MCMC)[53]. Detailed specification of the models, priors, hyperpriors and settings used to run these models can be found in the BEAST XML files (Supplementary Data 19). The results of our Bayesian analysis are visualized as a dated phylogenetic tree of the Transeurasian languages (Supplementary Data 24).

**Bayesian phylogeography.** We assumed that the dispersal of people through Eurasia can be described as a random walk, so is best captured by diffusion on a sphere[54]. To get an impression about the uncertainty in locating origins by such model, we performed a post hoc analysis using the posterior tree set from the lexical analysis. We assigned point

positions to the tips and randomly sampled trees from the posterior while estimating geographical parameters through MCMC. Even in this relatively restricted set-up, the uncertainty in root location does not allow us to distinguish the different geographical origin hypotheses. The results of our analysis are represented on a map (Supplementary Data 3). As Bayesian phylogeography must contend with a number of limitations[55,56], we complemented it with other homeland detection methods such as linguistic palaeontology and the diversity hotspot principle to reach a balanced location for the homelands of the root and nodes of the Transeurasian family (Supplementary Data 4).

**Linguistic palaeontology.** We compiled comparative agropastoral vocabularies for each Transeurasian subfamily: Turkic (Supplementary Data 5a), Mongolic (Supplementary Data 5b), Tungusic (Supplementary Data 5c), Koreanic (Supplementary Data 5d) and Japonic (Supplementary Data 5e). We applied linguistic reconstruction, a procedure for inferring an unattested ancestral state of a language on the evidence of data that are available from a later period, to corresponding words (Supplementary Data 5).

To distinguish between inherited and borrowed correspondence sets, we used standard criteria based on the phonology, semantics, morphology and distribution of the word involved, as specified in Supplementary Data 5. Dividing our dataset into inherited versus borrowed subsistence vocabulary, we determined distinctive spatiotemporal and cultural patterns for each category (Supplementary Data 5).

We applied linguistic palaeontology to our subsistence vocabulary, a historical comparative method that enables us to study human prehistory by correlating our linguistic reconstructions with information from archaeology about the culture of the ancient speech communities that used these words. In this way, we drew inferences about the subsistence strategies available to speakers of the different Transeurasian proto-languages in the Neolithic and Bronze Age (Supplementary Data 5) and identified a plausible location for the homeland of the ancient speech communities involved (Supplementary Data 4).

**Diversity hotspot principle.** To estimate the location of the ancient speech communities involved, we combined Bayesian phylogeography and linguistic palaeontology with the diversity hotspot principle. The principle is based on the assumption that the homeland is closest to the greatest diversity with regard to the deepest subgroups of the language family. We located these areas on the map and took them as an approximation of the area where a certain proto-language began to diversify (Supplementary Data 4). Although this method must contend with certain limitations (Supplementary Data 4), taken together with the other techniques for homeland location discussed here, it can give us a reasonably robust estimation of the location of an ancient speech community.

### Archaeology

**Archaeological database.** We scored 172 cultural traits for 255 Neolithic–Bronze Age archaeological sites or phases from the West Liao river basin (36), the Amur (Jilin, Heilongjiang and inland Liaoning) (32), the Primorye (4), the Liaodong peninsula (37), the eastern steppes (1), the Shandong peninsula (4), the Yellow River basin (2), the Korean peninsula (58) and the Japanese islands (85).

Sites with several major cultural phases were scored separately. The sites date from 8400–1700 BP and include the Early Neolithic to Bronze Age in northeast China, the Middle Neolithic Zaisanovka culture in the Primorye, the Middle–Late Neolithic Chulmun and Bronze Age Mumun cultures in Korea, and the Late Neolithic–Bronze Age Final Jomon and Yayoi cultures in western Japan. Categories of cultural traits scored comprised ceramics (70), stone tools (38), buildings (9), plant and animal remains (26), shell and bone artefacts (17) and burials (12). Definitions of scored features are found in Supplementary Data 6 (sheet 2) and further discussion of scoring methods can be found in Supplementary

Data 7. All features were scored as present (1) or absent (0) following published site reports or other literature.

The database was used to analyse changes in the distribution of Neolithic and Bronze Age artefacts over time, especially in relation to the spread of agricultural systems in Northeast Asia (Supplementary Data 7).

In addition, the cultural data in our archaeological database were analysed using Bayesian phylogenetic methods. There is a large amount of phylogenetic work with archaeological data[57], some parsimony-based[58], others distance-based[59]. The benefit of Bayesian approaches is that they are model-based, have sound formal mathematical foundations in probability theory allowing us to estimate uncertainty around all estimates, and allow integration of information from various sources in a single analysis (like cognate and geographic data) based on probability theory. BEAST is aimed specifically at inferring rooted time trees, and uncertainty of time estimates, which sets it apart from other Bayesian packages that target unrooted trees. Furthermore, BEAST supports models that are currently not available in other packages, hence the use of this package.

The cultural data are encoded as a binary alignment, and we applied the same substitution and clock models as for the lexical data. The pseudo Dollo model with relaxed clock fits the data best (Supplementary Data 20). Because the coefficient of variation of the relaxed clock exceeded 1, which indicates a considerable amount of variation, we also ran the analysis with the standard deviation capped at 1, which only slightly affected time estimates.

The large number of sampling dates and uncertainty on number of missing cultures made it hard to apply the fossilized birth death prior, so we opted for the flexible Bayesian skyline plot instead[60]. Timing information is based on sampling dates of archaeological finds. As there is uncertainty in dating these findings, tip dates were uniformly sampled in these intervals during the MCMC. In line with previous archaeological studies[61–63], we constrained the clades 'Xinglongwa–Zhabaogou–Hongshan' and 'Yabuli–Primorye' to be monophyletic (Supplementary Data 8). All analyses were performed in BEAST v.2.6[52] using adaptive coupled MCMC[53]. Details on models, priors, hyperpriors and settings can be found in the BEAST XML (Supplementary Data 21). The results of our Bayesian analysis are visualized as a phylogenetic tree of archaeological cultures in Northeast Asia (Supplementary Data 25) and interpreted in Supplementary Data 8.

**Archaeobotanical database.** In addition to the database of archaeological features, we compiled a list of the earliest crop remains from each region of Northeast Asia directly dated by radiocarbon (Supplementary Data 9). This list comprises 269 samples (China, 82; Primorye, 12; Korea, 31; Japan (excluding Ryukyus), 120; Ryukyu Islands, 24). Radiocarbon dates in this database were re-calibrated using OxCal v.4.4. We used kernel density mapping to plot the spread of cereals in this database over time Supplementary Data 7). Our databases were supplemented by published datasets for faunal remains[64,65], dolmens[66] and spindle whorls[67].

**Genetics**
**Laboratory procedures.** Ancient DNA wet laboratory work, including DNA extraction and library preparation, was performed in a dedicated ancient DNA clean room facility at the Max Planck Institute for the Science of Human History (MPI-SHH) and in an ancient DNA laboratory at Jilin University following established protocols[68]. A double-stranded library was built with 8-mer index sequences at both P5 and P7 Illumina adapters. Four individuals from China characterized in Jilin were directly shotgun-sequenced on the Illumina HiSeq X10 instrument in the 150-bp paired-end sequencing design to obtain an adequate coverage. Eighty-three double-stranded libraries for 33 individuals from Korea and Japan were generated and characterized in the MPI-SHH either by shotgun sequencing or by insolution capture at approximately

1.2 million informative nuclear single-nucleotide polymorphisms (SNPs). After initial screening of the preservation of those libraries, a further 108 single-stranded libraries were built aiming at retrieving more endogenous DNA from the samples, and again, those libraries were directly shotgun-sequenced and in-solution-captured at around 1.2 million SNPs (Supplementary Data 17) and sequenced on the Illumina HiSeq 4000 platform following the manufacturer's protocols.

**Sequence data processing.** Raw sequencing reads were processed by an automated workflow with the EAGER v.1.92.55 programme[69]. Illumina adapter sequences were trimmed from the sequencing data and overlapping pairs were merged with AdapterRemoval v.2.2.0[70]. We mapped the merged reads with a minimum of 30 bp to the human reference genome (hs37d5; GRCh37 with decoy sequences) using BWA v.0.7.12[71]. We removed PCR duplicates by DeDup v.0.12.2[60]. To minimize the effect of post-mortem DNA damage on genotyping, we masked 2 bp for nonUDG libraries and 10 bp for half-UDG libraries on both ends per read using the trimbam function on bamUtils v.1.0.13[72]. The cleaned reads with both base quality (Phred-scale quality) and mapping quality (Phred-scale mapping quality) over 30 were piled up by SAMtools 1.3[60] with the mpileup function. We called pseudo-diploid genotypes using the pileupCaller program (https://github.com/stschiff/sequenceTools) against SNPs in the '1240k' panel[73,74] under the random haploid calling mode. For C/T and G/A SNPs, we used the masked BAM files; for the rest we used the original unmasked BAM files.

**Reference datasets.** We compared our ancient individuals to three sets of world-wide genotype panels, one based on the Affymetrix HumanOrigins Axiom Genome-wide Human Origins 1 array ('HumanOrigins'; 593,124 autosomal SNPs)[75], the '1240k' panel[73], and the 'Illumina' dataset[76]. We augmented these datasets by adding the Simons Genome Diversity Panel[77] and published ancient genomes (Supplementary Data 11).

**Ancient DNA authentication.** We applied multiple criteria to confirm the authentication of the newly published ancient genomes from Korea and Japan. First, we characterized the post-mortem chemical modifications characteristic for ancient DNA using mapDamage v.2.0.6[78]. Second, we estimated mitochondrial contamination rates for all individuals using Schmutzi v.1.5.1[79]. Third, we measured the nuclear genome contamination rate in males on the basis of X chromosome data as implemented in ANGSD v.0.910[80]. As males have only a single copy of the X chromosome, mismatches between bases, aligned to the same polymorphic position, beyond the level of sequencing error are considered as evidence of contamination. Fourth, we assessed the potential West Eurasian contamination with all reads available and the damage-restricted reads on single-stranded libraries implemented in the PMDtools[81] with a PMD score of at least 3 and compared their positions in a Eurasia PCA with all reads and damaged reads alone. Fifth, we applied qpAdm[74] per individual to further characterize the West Eurasian contamination with West Eurasian characteristic groups such as Sintashta_MLBA or LBK_EN as sources (see Supplementary Data 17, 22 for details).

**Population structure analysis.** We performed a PCA with the smartpca v.16000[82] using a set of 2,077 present-day Eurasian individuals from the 'HumanOrigins' dataset and the '1240kIllumina' dataset with the option 'lsqproject: YES' and 'shrinkmode: YES'. We used outgroup-$f_3$ statistics[83,84] to obtain a measurement of genetic affinity between two populations since their divergence from an African outgroup. We calculated $f_4$ statistics with the 'f4mode: YES' function in admixtools[31]. Both $f_3$ and $f_4$ statistics were calculated using qp3Pop v.435 and qpDstat v.755 in the admixtools package.

**Genetic sexing and uniparental haplogroup assignment.** We determined the molecular sex of our ancient samples by comparing the ratio of X and Y chromosome coverages to autosomes[85]. For women, we

would expect an approximately even ratio of X to autosome coverage and a Y ratio of 0. For men we would expect roughly half of the coverage on X and Y than autosomes.

**Admixture modelling with qpAdm.** We modelled the ancient individuals in this study using the qpWave/qpAdm framework (qpWave v.410 and qpAdm v.810) in the admixtools v.5.1 package[74]. We used the following 7 populations in '1240k' datasets as outgroup ('OG'): Mbuti, Onge, Iran_N, Villabruna, Karitiana, Naxi and Funadomari Jomon. This set includes an African outgroup (Mbuti), Andamanese islanders (Onge), early Neolithic Iranians from the Tepe Ganj Dareh site (Iran_N), late Pleistocene European hunter-gatherers (Villabruna), indigenous Karitiana from Brazil, a Tibetan-Burman speaking group from southern China (Naxi) and ancient hunter-gatherers from Japan (Funadomari Jomon) (Supplementary Data 13, 16).

## Triangulation

The term 'triangulation' is borrowed from a navigational technique that determines a single point in space with the convergence of measurements taken from two other distinct points. In qualitative research it designates a method used to capture different dimensions of the same phenomenon by using evidence from three distinct scientific disciplines. To avoid circularity in the argumentation, data collection, analyses and results are performed or reached within the limits of each individual discipline, independently from the other two. Only in the final phase of the triangulation process are the inferences drawn by the three disciplines mapped on each other by comparing a number of variables describing the phenomenon. The purpose of triangulation is to increase the credibility and validity of the results by evaluating the extent to which the evidence from the three disciplines converges and by identifying correlations, inconsistencies, uncertainties and potential biases across the different perspectives on the investigated phenomenon.

Building on previous applications of triangulation in anthropology[86], we applied the method to the dispersal of the Transeurasian languages, integrating linguistics, archaeology and genetics to contribute a better understanding of the phenomenon. We collected different datasets and applied the methods described above to draw independent inferences with regard to a number of variables such as location, chronology, migratory dynamics, continuity versus diffusion, and subsistence (Supplementary Data 26). Each discipline inferred the most parsimonious model involving these variables on the basis of the application of tools internal to its own field, whether qualitative or quantitative, based on direct or indirect evidence. Taken by itself, a single discipline alone cannot conclusively resolve the question about farming/language dispersals, but taken together the three disciplines increase the credibility and validity of this scenario. Aligning the evidence offered by the three disciplines, we gained a more balanced and richer understanding of Transeurasian migration than each of the three disciplines could provide us with individually.

## Reporting summary

Further information on research design is available in the Nature Research Reporting Summary linked to this paper.

## Data availability

Linguistic and archaeological datasets are available through the Supplementary Information. Files that require applications were uploaded to FigShare. The links to FigShare are as follows: Supplementary Data 3: Bayesian phylogeographic analysis modelling the spatiotemporal expansion of the Transeurasian languages (https://figshare.com/s/b9c67ca3ea47faf51d48); Supplementary Data 19: BEAST XML files specifying the models, priors, hyperpriors and settings used to run the analyses of the linguistic database (https://

figshare.com/s/748bf751fe3ba7752046); Supplementary Data 21: BEAST XML files specifying the models, priors, hyperpriors and settings used to run the analyses of the archaeological database (https://figshare.com/s/99f5aab9a2e43eb2ffd4); Supplementary Data 24: dated Bayesian phylogeny of the Transeurasian languages (https://figshare.com/s/709f239fa45982911b87); and Supplementary Data 25: Bayesian phylogenetic analysis of the archaeological database (https://figshare.com/s/65615dddc0817bc0184f). The link to the figtree application is: https://github.com/rambaut/figtree/releases/tag/v1.4.3 For our genetic datasets, the DNA sequences reported in this paper have been deposited in the European Nucleotide Archive (ENA) under accession PRJEB46162. Haploid genotype data of ancient individuals in this study on the '1240k' panel are available in the EIGENSTRAT format from the following link: https://edmond.mpdl.mpg.de/imeji/collection/59JGAaOpSxRb96Vh.

## Code availability

Readers can access the code that underlies our Bayesian analyses of linguistic and cultural datasets through the Supplementary Information. The files in Supplementary Data 19 relate to languages and those in Supplementary Data 21 to cultures. The web-links are: Supplementary Data 19: BEAST XML files specifying the models, priors, hyperpriors and settings used to run the analyses of the linguistic database (https://figshare.com/s/748bf751fe3ba7752046); Supplementary Data 21: BEAST XML files specifying the models, priors, hyperpriors and settings used to run the analyses of the archaeological database (https://figshare.com/s/99f5aab9a2e43eb2ffd4).

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

**Acknowledgements** The research leading to these results has received funding from the European Research Council under the European Union's Horizon 2020 research and innovation programme (grant agreement no. 646612) granted to M.R. R.B. was supported by a Marsden grant 18-UOA-096 from the Royal Society of New Zealand. We thank N. Adachi, T. Kakuda, E. Savelyeva, W. Lawrence, S. Wichmann, C. Wang, M. Burri, N. Klyuev, I. Zhushchikhovskaya, M. Byington, H. Miyagi, Y. Vostretsov, A. Jarosz, J.-O. Svantesson, M. Levy, J. Lefort, M. Miller, K. Mischenkova, E. Perekhvalskaya, I. Nikolaeva, P. Czerwinski, N. Aralova, A. Francis-Ratte, I. Joo, R. Máté, T. Pellard and the Korean National Museum for helping to compile, analyse or interpret data.

**Author contributions** The research was conceptualized by M.R. Linguistic datasets were collected by A.S., J.D., S.O., B.D., R. Bjørn, S.R., K.-D.A., I.G., O.M., J.R.B. and M.R. The linguistic database was scored by M.R. and analysed by M.R. and R. Bouckaert. Etymologies were established by M.R. The archaeology database was scored by T.L., M.C., T.K., G.K., J.U. and L.G., and analysed by M.J.H., R. Bouckaert, M.R., M.C. and I.R.B. The Nagabaka site was excavated by T.K. and K.-Y.Y. under the direction of M.J.H. with advice from M.K. and H.I. Post-excavation analyses of materials from Nagabaka were analysed by K.-Y.Y., T.K., N.S., H. Tomita, H. Takamiya, J.U., P.R., R.F. and M.Y. Y.C. shared the Angangxi data, D.I.-A. and J.-H.K. the ancient Korean data, K.i.S. the Yayoi data and H.I., R.K., T.S. and H.O. the modern Ryukyu data. Wet laboratory works for ancient DNA data from Korea and Japan were carried out by R.A.B. and M.H. Genetic data analyses were carried out by C.N. with input from H.K.-K. and F.Z. The writing was done by M.R., M.J.H. and C.N.

**Funding** Open access funding provided by Max Planck Society.

**Competing interests** The authors declare no competing interests.

**Additional information**
**Correspondence and requests for materials** should be addressed to Martine Robbeets, Mark J. Hudson or Chao Ning.

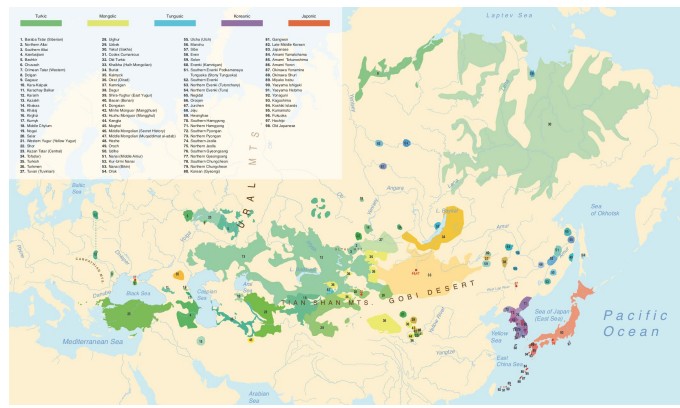

**Extended Data Fig. 1 | Legend for Fig. 1.** Detailed legend to accompany main Fig. 1.

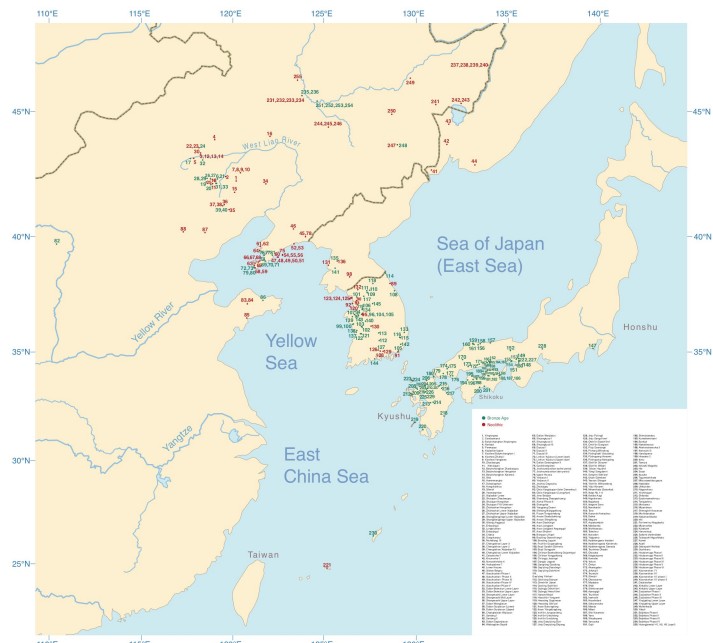

**Extended Data Fig. 2 | Legend for Fig. 2.** Detailed legend to accompany main Fig. 2.

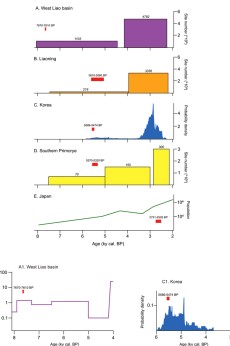

**Extended Data Fig. 3 | Demographic changes with agriculture in Neolithic and Bronze Age. Northeast Asia.** A1 shows changes following the adoption of millet farming ca. 8000–4000 BP, using quantity of pottery for the West Liao[29] and B2 shows these changes using radiocarbon proxy dates for Korea[87]. Figures A to E show long-term dynamics ca. 8000–2000 BP following the integration of millet with rice, barley and wheat in the Bronze Age and based on site numbers for NE China[88], radiocarbon dates for Korea[87] and site numbers for Japan[89]. For references and methods used to derive demographic information from the proxies, see Supplementary Data 7.

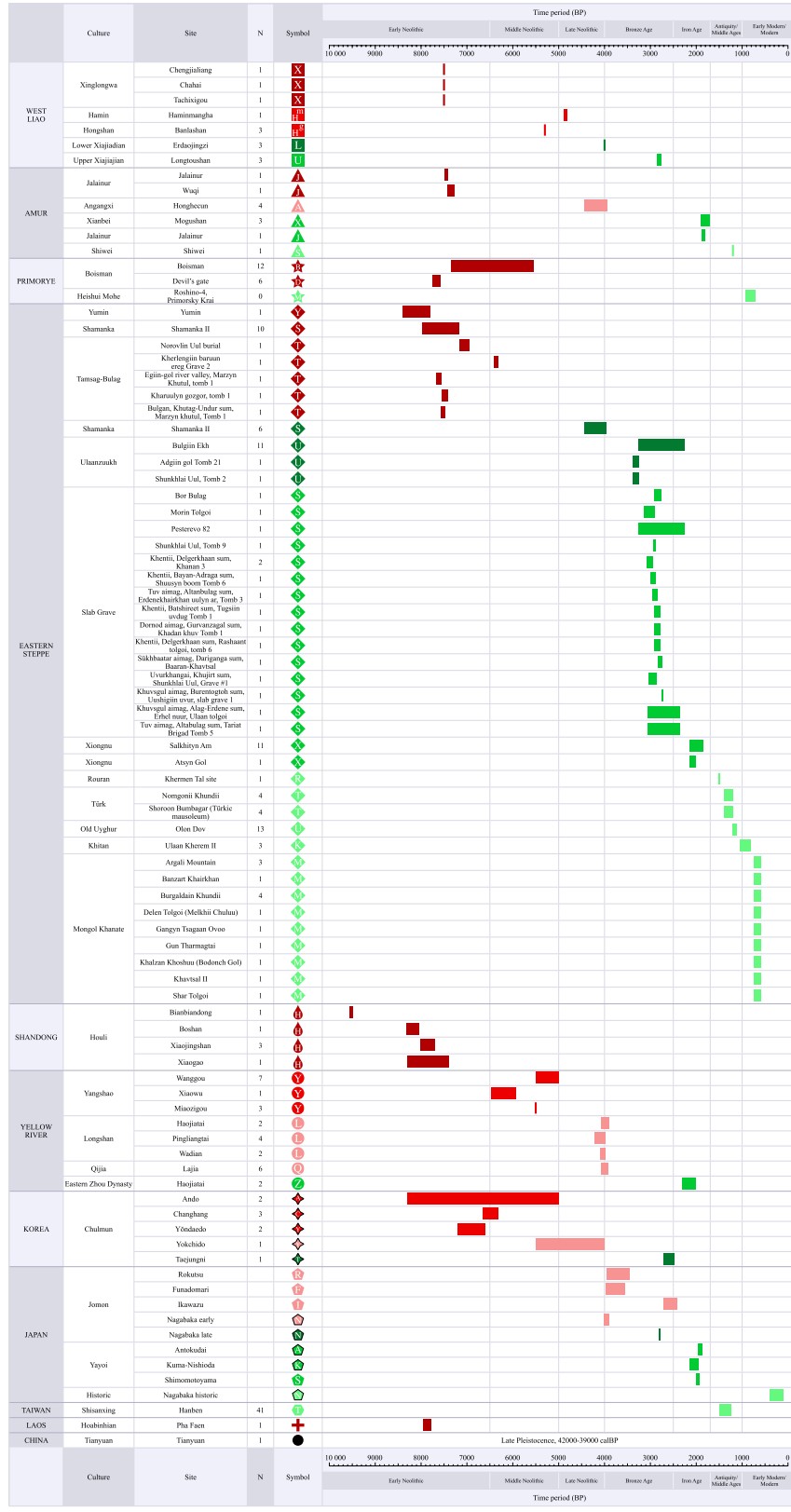

**Extended Data Fig. 4 | Ancient genomes located in time and space.** Includes detailed legend to accompany main Fig. 3 and Extended Data Figs. 7–10.

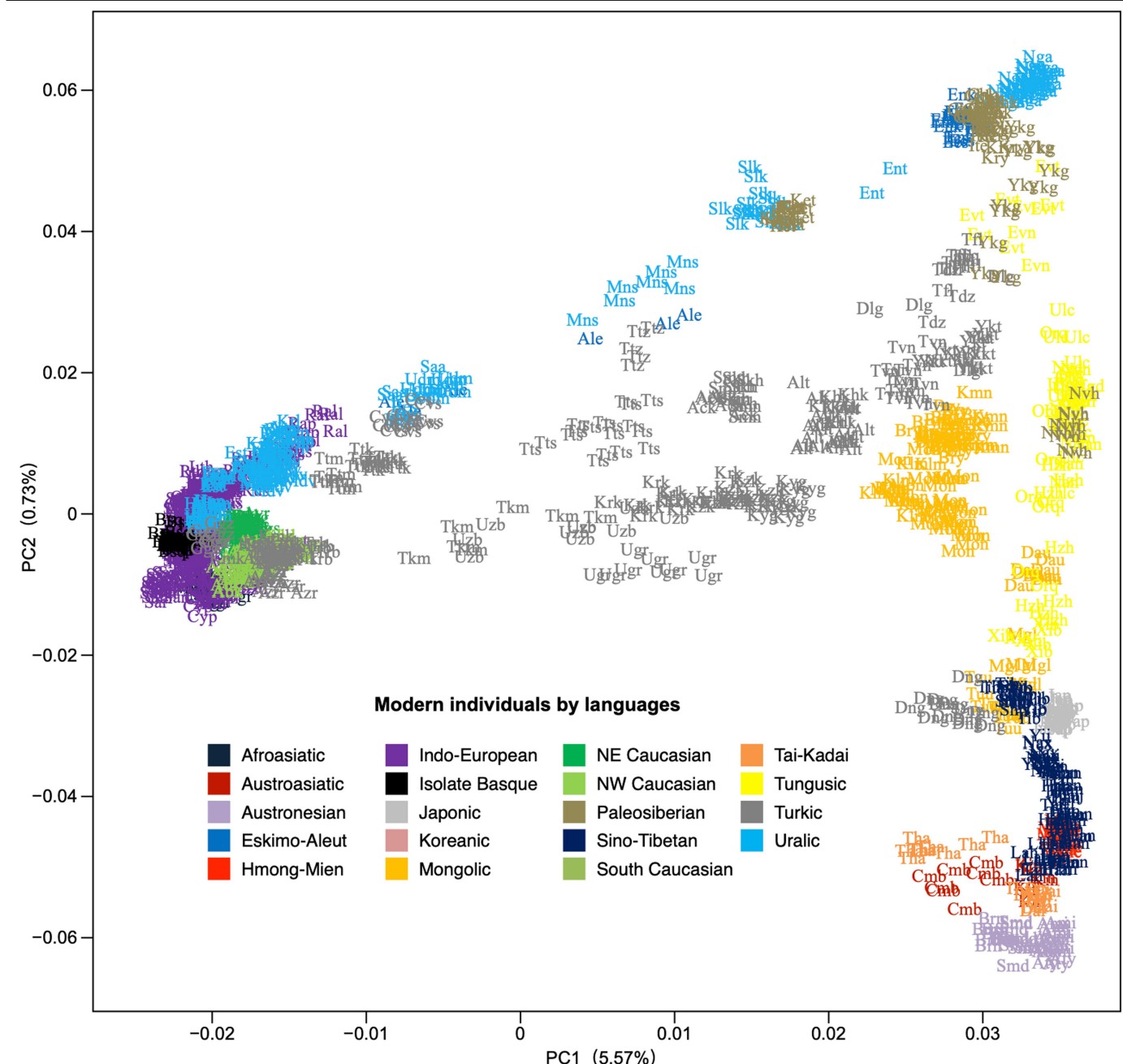

**Extended Data Fig. 5 | PCA displaying the genetic structure of present-day Eurasians.** PC1 separates Western and Eastern Eurasian populations, PC2 Southern and Northern Eurasian populations. Transeurasian populations are coloured according to subfamily (Turkic in grey, Mongolic in orange, Tungusic in yellow, Koreanic in pink, Japonic in light grey). Non-Transeurasian populations are coloured according to families. Populations are labelled with three letters, for a list of abbreviations, see Supplementary Data 10.

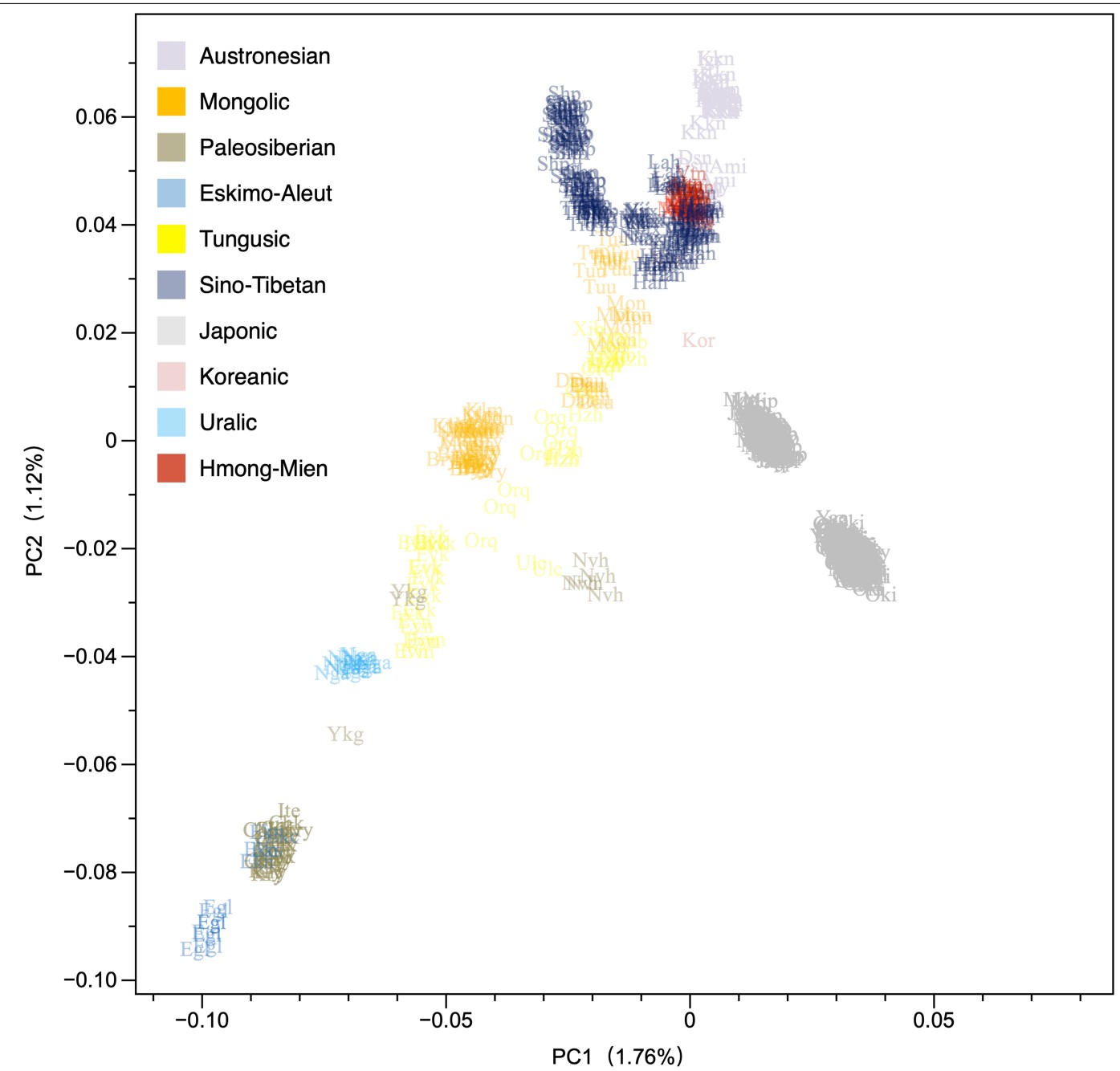

**Extended Data Fig. 6 | PCA displaying the genetic structure of present-day East Asians.** Populations are labelled with three letters, for a list of abbreviations, see Supplementary Data 10.

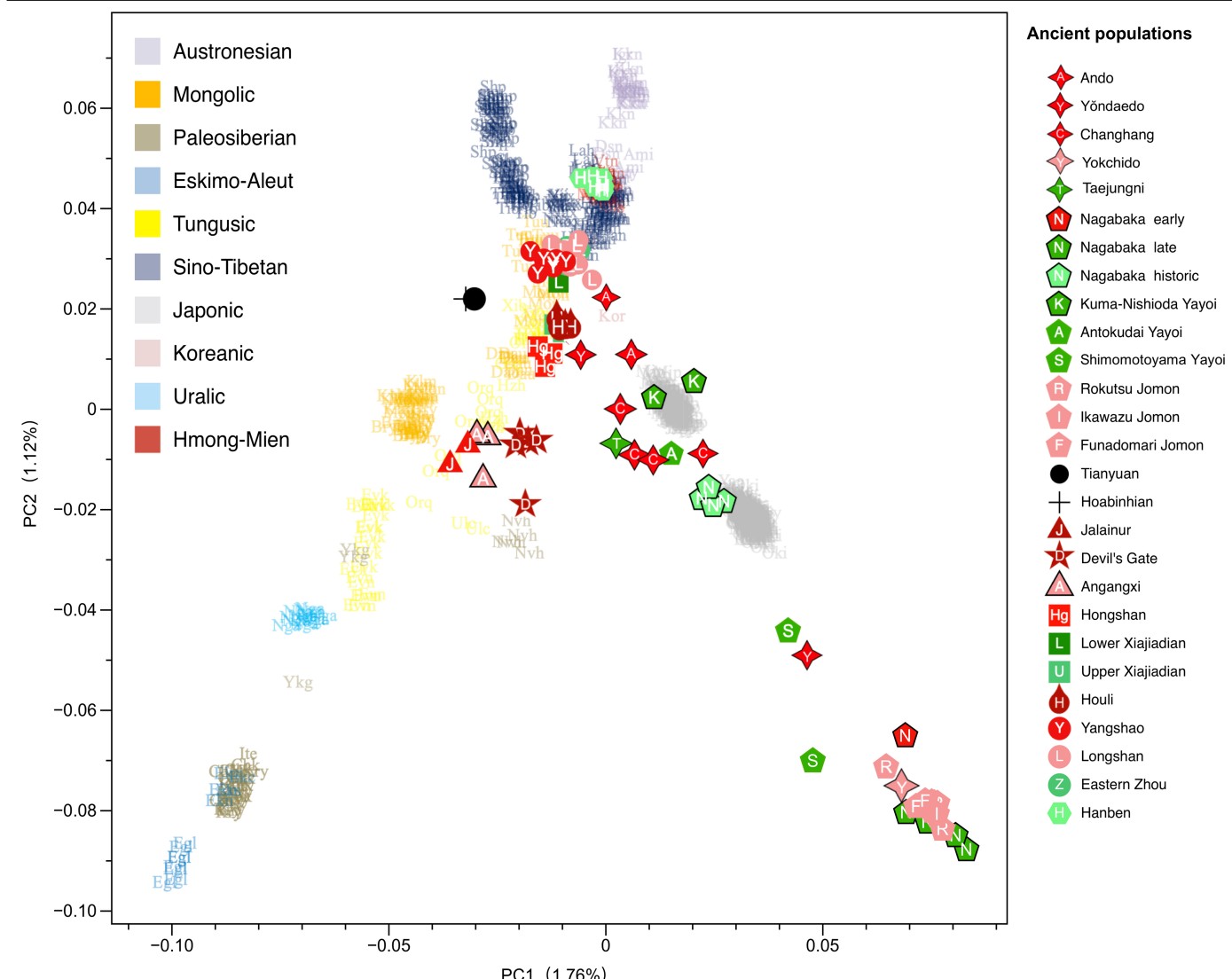

**Extended Data Fig. 7 | Ancient genomes plotted on PCA displaying the genetic structure of present-day East Asians.** For a detailed legend, see Extended Data Fig. 4.

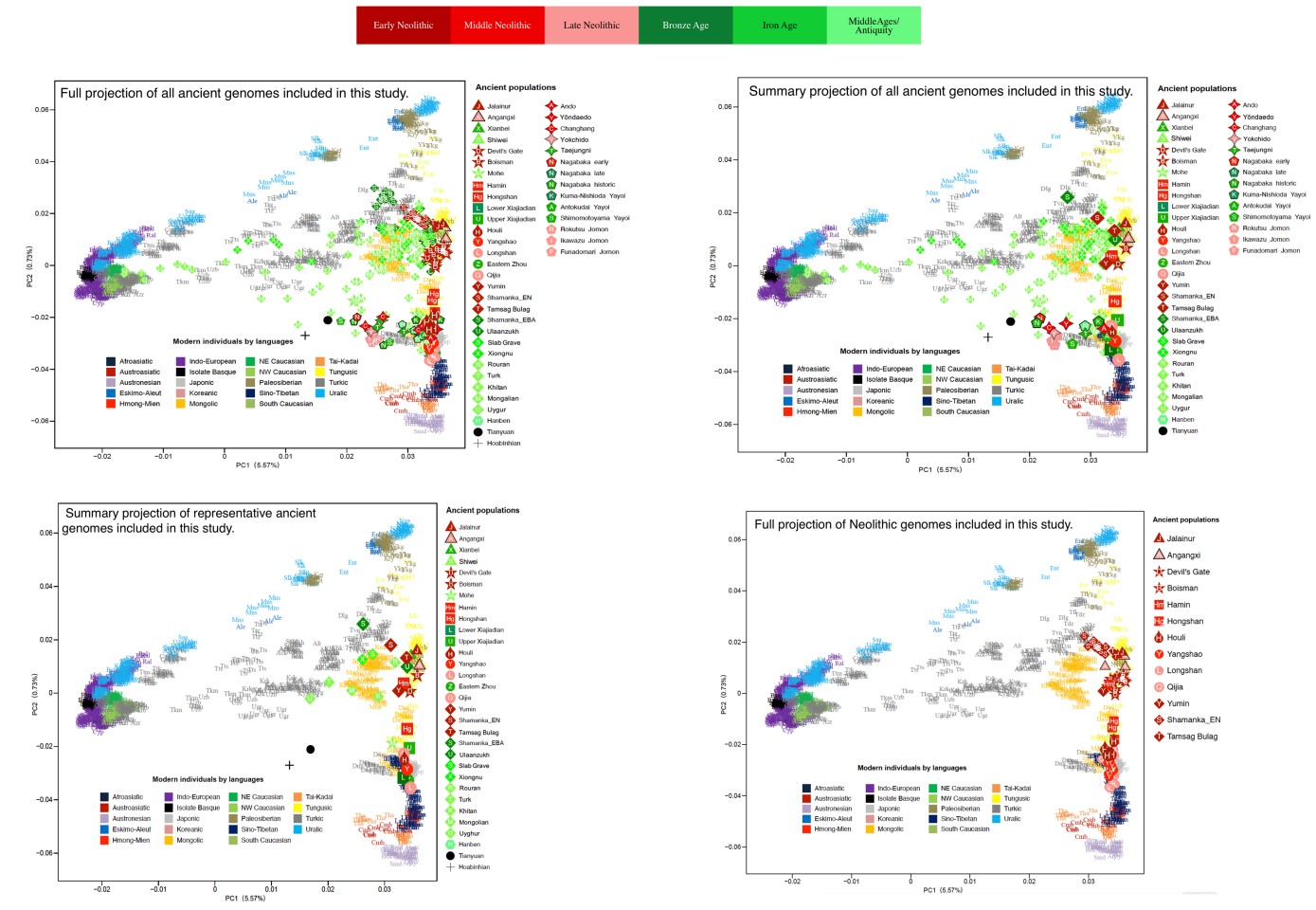

**Extended Data Fig. 8 | Ancient genomes plotted on PCA displaying the genetic structure of present-day Eurasians.** For a detailed legend, see Extended Data Fig. 4.

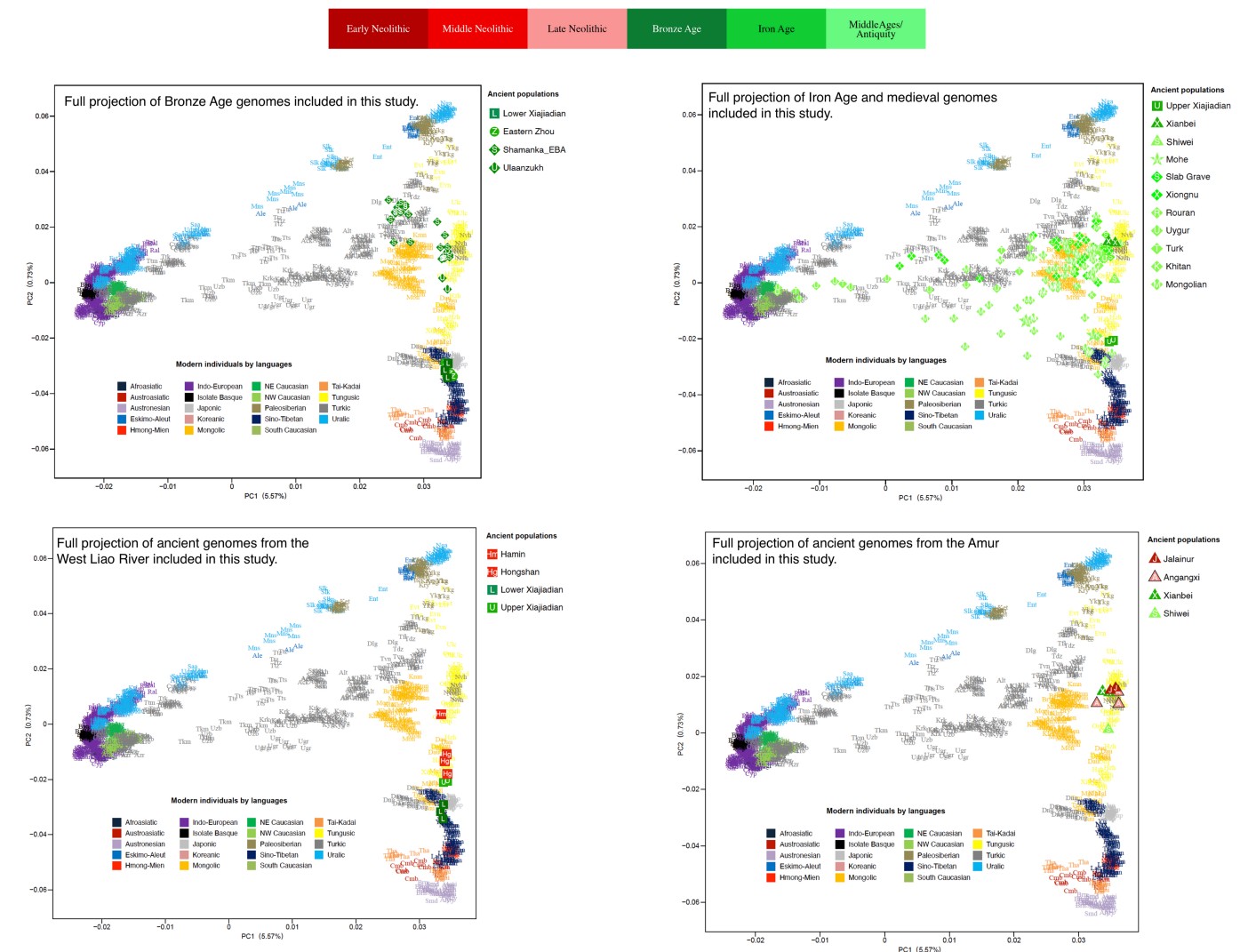

**Extended Data Fig. 9 | Ancient genomes from Bronze Age, Iron Age, West Liao and Amur plotted on PCA displaying the genetic structure of present-day Eurasians.** For a detailed legend. see Extended Data Fig. 4.

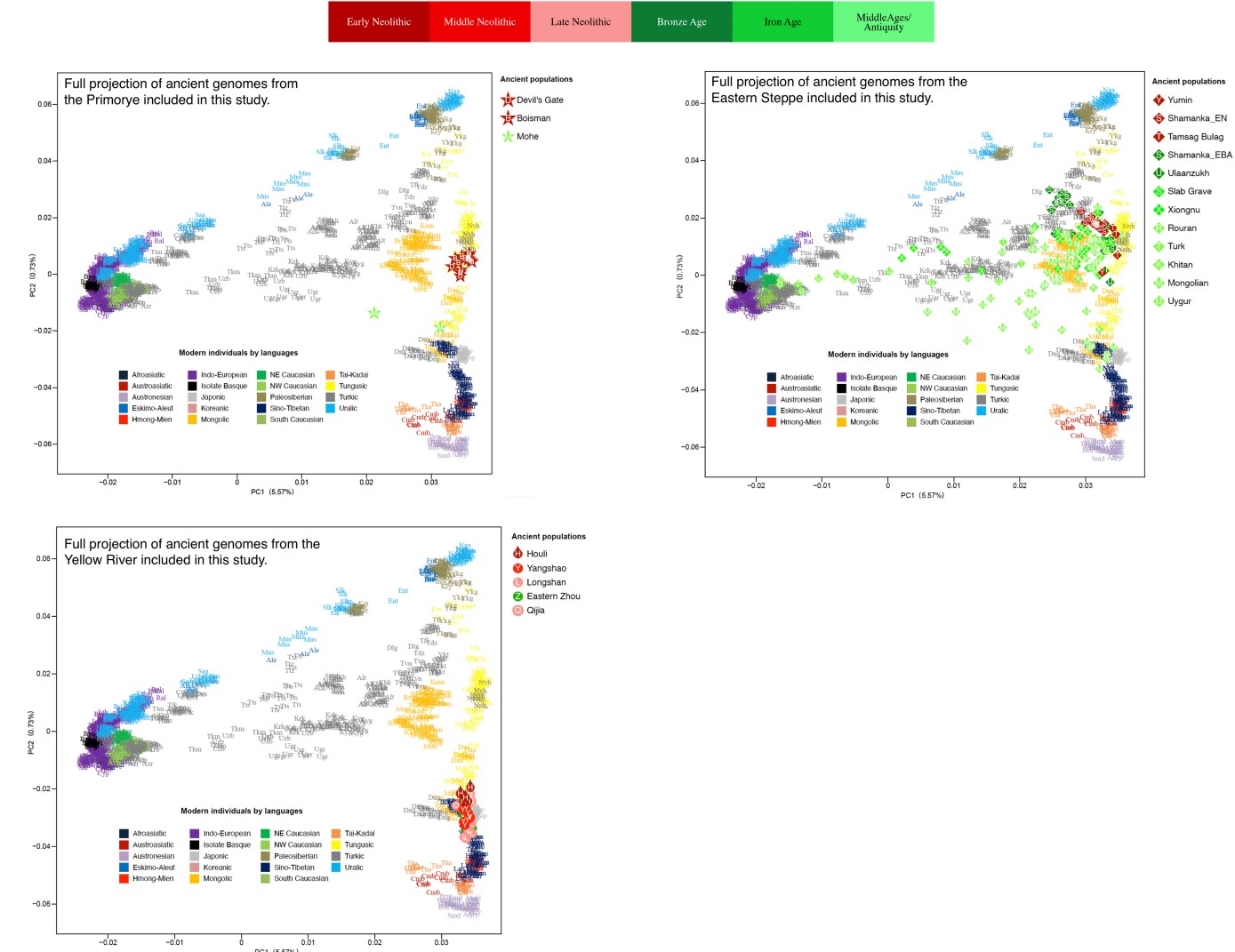

**Extended Data Fig. 10 | Ancient genomes from Primorye, eastern steppe and Yellow River plotted on PCA displaying the genetic structure of present-day Eurasians.** For a detailed legend, see Extended Data Fig. 4.

# Reporting Summary

Nature Research wishes to improve the reproducibility of the work that we publish. This form provides structure for consistency and transparency in reporting. For further information on Nature Research policies, see our Editorial Policies and the Editorial Policy Checklist.

## Statistics

For all statistical analyses, confirm that the following items are present in the figure legend, table legend, main text, or Methods section.

| n/a | Confirmed | |
|---|---|---|
| ☐ | ☒ | The exact sample size (*n*) for each experimental group/condition, given as a discrete number and unit of measurement |
| ☐ | ☒ | A statement on whether measurements were taken from distinct samples or whether the same sample was measured repeatedly |
| ☐ | ☒ | The statistical test(s) used AND whether they are one- or two-sided *Only common tests should be described solely by name; describe more complex techniques in the Methods section.* |
| ☒ | ☐ | A description of all covariates tested |
| ☐ | ☒ | A description of any assumptions or corrections, such as tests of normality and adjustment for multiple comparisons |
| ☒ | ☐ | A full description of the statistical parameters including central tendency (e.g. means) or other basic estimates (e.g. regression coefficient) AND variation (e.g. standard deviation) or associated estimates of uncertainty (e.g. confidence intervals) |
| ☐ | ☒ | For null hypothesis testing, the test statistic (e.g. *F*, *t*, *r*) with confidence intervals, effect sizes, degrees of freedom and *P* value noted *Give P values as exact values whenever suitable.* |
| ☐ | ☒ | For Bayesian analysis, information on the choice of priors and Markov chain Monte Carlo settings |
| ☒ | ☐ | For hierarchical and complex designs, identification of the appropriate level for tests and full reporting of outcomes |
| ☒ | ☐ | Estimates of effect sizes (e.g. Cohen's *d*, Pearson's *r*), indicating how they were calculated |

*Our web collection on statistics for biologists contains articles on many of the points above.*

## Software and code

Policy information about availability of computer code

| Data collection | The code used in the Bayesian analysis of the linguistic and cultural topologies is fully referenced. Readers can access the code underlying our Bayesian analyses of linguistic and cultural datasets through the supplementary information. The files in SI 19 relate to languages and those in SI 21 to cultures; see https://figshare.com/s/748bf751fe3ba7752046 and https://figshare.com/s/99f5aab9a2e43eb2ffd4 Illumina sequence data were processed using the following programs to obtain genotype data used in the analysis: EAGER v1.92.55, AdapterRemoval v2.2.0, BWA v0.7.12, DeDup v0.12.2, bamUtils v1.0.13, pileupCaller (https://github.com/stschiff/sequenceTools), mapDamage v2.0.9, ANGSD v0.910, Schmutzi v1.5.1. These programs are publicly available. . |
|---|---|
| Data analysis | The code used in the Bayesian analysis of the linguistic and cultural topologies is fully referenced. Population genetic data analysis in this study was performed using the following publicly available programs: Smartpca v16000, ADMIXTURE v1.3.0, PLINK v1.90, lcMLkin v0.5.0, qp3Pop v435, qpDstat v755, qpWave v410, qpAdm v810, DataGraph v4.5.1. Non-default parameters used in our analysis are described in the Methods section. The base map in Figure 1 was downloaded from the Nature Earth map dataset (https://www.naturalearthdata.com/), granted for the public domain use and is free for use in any type of project. Calibration of AMS 14C dating results was done by OxCal v4.4, using the IntCal20 database. |

For manuscripts utilizing custom algorithms or software that are central to the research but not yet described in published literature, software must be made available to editors and reviewers. We strongly encourage code deposition in a community repository (e.g. GitHub). See the Nature Research guidelines for submitting code & software for further information.

## Data

Policy information about availability of data

All manuscripts must include a data availability statement. This statement should provide the following information, where applicable:

- Accession codes, unique identifiers, or web links for publicly available datasets
- A list of figures that have associated raw data
- A description of any restrictions on data availability

All linguistics and archaeological datasets are available through the supplementary information. Files that require applications were uploaded on two external sources, i.e. GitHub (https://github.com/rbouckaert/Eurasia3angle) and FigShare. For our genetic datasets, the DNA sequences reported in this paper have been deposited in the European Nucleotide Archive (ENA) under accession PRJEB46162. Haploid genotype data of ancient individuals in this study on the 1240k panel are available in the EIGENSTRAT format from the following link:https://edmond.mpdl.mpg.de/imeji/collection/59JGAaOpSxRb96Vh

# Field-specific reporting

Please select the one below that is the best fit for your research. If you are not sure, read the appropriate sections before making your selection.

☒ Life sciences  ☐ Behavioural & social sciences  ☐ Ecological, evolutionary & environmental sciences

For a reference copy of the document with all sections, see nature.com/documents/nr-reporting-summary-flat.pdf

# Life sciences study design

All studies must disclose on these points even when the disclosure is negative.

| | |
|---|---|
| Sample size | No sample-size calculation was performed. The study proceeding by attempting to sample ancient DNA from contexts that were not previously analyzed and every new sample contributed meaningful new information. The uncertainties due to limited sample size are clearly indicated when there are concerns. |
| Data exclusions | Data were excluded for analysis based either on evidence for sample contamination, or low coverage data. We clearly indicate these cases. |
| Replication | As our study is an evolutionary analysis of language, culture and genes and the evolutionary process only proceeds once, replication was not possible. |
| Randomization | This is not relevant to our study because we are dealing with an evolutionary process not a human-designed experiment. |
| Blinding | Blinding was not possible for this study because the analysts needed to understand the historical background of the samples. |

# Reporting for specific materials, systems and methods

We require information from authors about some types of materials, experimental systems and methods used in many studies. Here, indicate whether each material, system or method listed is relevant to your study. If you are not sure if a list item applies to your research, read the appropriate section before selecting a response.

### Materials & experimental systems

| n/a | Involved in the study |
|---|---|
| ☒ | ☐ Antibodies |
| ☒ | ☐ Eukaryotic cell lines |
| ☐ | ☒ Palaeontology and archaeology |
| ☒ | ☐ Animals and other organisms |
| ☒ | ☐ Human research participants |
| ☒ | ☐ Clinical data |
| ☒ | ☐ Dual use research of concern |

### Methods

| n/a | Involved in the study |
|---|---|
| ☒ | ☐ ChIP-seq |
| ☒ | ☐ Flow cytometry |
| ☒ | ☐ MRI-based neuroimaging |

## Palaeontology and Archaeology

| | |
|---|---|
| Specimen provenance | Skeletal samples newly analysed in the study are under the custodianship of archaeologists or anthropologists in our team who contributed them to the study and whose permission to analyse the samples is indicated through co-authorship of the manuscript. |
| Specimen deposition | The analyzed samples are under the custodianship of the co-authors who contributed them to the study; the provenance of each sample is described in SI 11 and SI 12. Our co-authors will give access to the parts of the samples remaining after ancient DNA and radiocarbon analysis to anyone who requests it. We also shared photos in SI 13 and commit to sharing more photographic material of skeletal samples before and after sampling. |

Dating methods

We dated the root of our linguistic family and the nodes in the family using Bayesian estimation methods, based on calibrating against known time spans provided by dated written records; see Extended Data Fig 1 and BEAST XML files in SI 19.We further report existing radiocarbon dates of archaeological specimens and new radiocarbon dates on bone in this paper; see SI 14 and SI 15.

☒ Tick this box to confirm that the raw and calibrated dates are available in the paper or in Supplementary Information.

Ethics oversight

No ethical approval or guidance was required because we did not perform research on living human participants or animals.

Note that full information on the approval of the study protocol must also be provided in the manuscript.

