## [Peer Review File · Nature]

Manuscript Title: Triangulation supports agricultural spread of the Transeurasian languages

Reviewer Comments & Author Rebuttals

Reviewer Reports on the Initial Version:

Referee #1 (Remarks to the Author):

A. Summary of the key results

The Neolithic protolanguages, viz. Proto-Transeurasian, Proto-Altaic, Proto-Mongolo-Tungusic and Proto-Japano-Koreanic reflect a small core of inherited words relating to cultivation ('field', 'sow', 'plant', 'grow', 'cultivate', 'spade'), millets but not rice or other crops ('millet seed', 'millet gruel'), food production and preservation ('ferment', 'grind', 'crush to pulp', 'brew'), wild foods suggestive of sedentism ('walnut', 'acorn', 'chestnut'), textile production ('sew', 'weave cloth', 'weave with a loom', 'spin', 'cut cloth', 'ramie', 'hemp'), and pigs and dogs as the only domesticated animals. By contrast, individual subfamilies that separated in the Bronze Age, viz. Turkic, Mongolic, Tungusic, Koreanic and Japonic, inserted new subsistence terms relating to the cultivation of rice, wheat and barley, dairying, domesticated animals such as cattle, sheep, and horses, farming or kitchen tools, and textiles such as silk. These words are borrowings resulting from linguistic interaction between Bronze Age populations speaking various Transeurasian and non-Transeurasian languages. In sum, the age, homeland, original agricultural vocabulary and contact profile of the Transeurasian macrofamily support the 'Farming Hypothesis' and exclude the 'Pastoralist Hypothesis'. These conclusions obtained from comparative linguistics are in correlation with recent results of archaeology and genetics.

B. Originality and significance: if not novel, please include reference

The 'Farming Hypothesis' verified in the contribution is quite new and rather surprising, but only for the first view. The shift of farmers to pastoralists is known e.g. from the Near East. Early Semitic civilisation was based on agriculture, but some its bearers changed into pastoralists with regard to their migrations for more arid areas (Arabian Peninsula). And the founder of the modern concept of the Nostratic hypothesis, Illič-Svityč, connected the hypothetical Nostratic protolanguage with the Near Eastern Neolithics. If the Transeurasian protolanguage was one of its descendants, its lexicon should preserve some terms connected with agriculture.

C. Data & methodology: validity of approach, quality of data, quality of presentation

The authors apply standard methods to their research. New is their combination called aptly 'triangulation' and number and quality of used data in the field of linguistics, archaeology and genetics. Admirable is also their effort to keep a neutral distance to eliminate any prejudice.

D. Appropriate use of statistics and treatment of uncertainties

Statistical approaches play very important role in the present research. The authors prefer Bayesian statistics operating with conditional probabilities.

E. Conclusions: robustness, validity, reliability

The results represent of maximum of possible with regard to both quality and quantity.

F. Suggested improvements: experiments, data for possible revision

It is difficult to arrange any experiments in research of any hypothetical ethnic group and its no less hypothetical protolanguage, with exception of testing the correlation between the linguistic and archaeological results. And just this approach was applied here.

G. References: appropriate credit to previous work?

During last two decades the leader of the research team, Martine Robbeets, made the enormous work, first evaluating the older studies and later testing new hypotheses. She is able to work both alone and in the team, always preferring discussion. She always seeks best solution, never dogma.

H. Clarity and context: lucidity of abstract/summary, appropriateness of abstract, introduction and conclusions

All parts of the text are formulated comprehensibly and clearly, the used argumentation is convincing and correct.

Referee #2 (Remarks to the Author):

Robbeets et al. combine three different disciplines (linguistics, archaeological dates and ancient genetics) to explore by a "triangulation" approach the origin and dispersal of Transeurasian languages (including Japanese, Korean, Tungusic, Mongolic and Turkic) and placed it with the advent and spread of millet farming, around the Amur river (North East Asia). Genetics of ancient Korean genomes as well as other from Ryukyu islands, altogether with previously published data, also supports the view that agriculture was associated to expanding ancestries. These findings contradict previous hypothesis that related the dispersal of Transeurasian languages to Bronze Age population movements that, although detected here also from genetics, postdate the original Neolithic movements.

This is a monumental work that tries to be a synthesis of three disciplines in an integrative way, and come to their conclusions by searching a geographic and temporal overlap among all evidence. It is of course complex to read because, among other things, I assume no one can be an expert on the three different fields and I am unsure on how controversial the linguistic and archaeological findings might be. Therefore, I will comment more specifically on the genetics section. This said, I find the effort and the general conception of this work appealing and with the possibility of becoming a landmark study on East Asian human population history, provided they can explain it in a clear way to non experts in Asian prehistory.

My main concern is the potential overlapping of this work with another previously published at Nature (February 22th, 2021) that shares some of the coauthors (Chuan-Chao Wang et al. Genomic Insights into the Formation of Human Populations in East Asia). The focus is not that interdisciplinary, but for instance the authors link genetics with linguistics also to conclude that Yellow River Basin farmers likely spread the Sino-Tibetan languages at ca 3000 years BCE. The authors cite this work when it was at BioRxiv (ref 7), but I think they should mention at least this conclusion, as this seems to provide genetic evidence for dispersion centres for farming in Asia, this one in particular I guess associated to rice farming.

Also Chao Ning et al. (2020, Nat Comm, ref 6, again shared with some coauthors), partly overlap with the subject of this work; as they provide evidences of correlation between archaeolinguistic signatures and past human migrations in northern China. Again, I think the genetics section (or the Discussion) would benefit of mentioning explicitly these previous findings that preclude some of the findings of the current work.

In the main text, Gakuhari et al. (2020) Comm. Biol. is not mentioned; in this paper the authors sequence a 2,500 years-old Jomon genome from the main island of Japan and conclude she has affinities with mainland Upper Paleolithic hunter-gatherer populations from South Asia and connections to Taiwan (thus, favouring a southern settlement route). According to them, this Jomon individual clusters with previously published Jomon individuals from Hokkaido (Kanzawa-Kiriyama et al. 2019, again not referenced in the main text). They date the divergence of the Jomon ancestry from that of mainland Asia to have occurred between 40,000 and 26,000 years ago. In contrast the current paper states that Jomon ancestry was common in Korea by 6,000 y BP. I might be wrong but I understand both findings are not necessarily contradictory (the previous lack of ancient Korean genomes might prevent to find this Jomon-like signal up north in the mainland). However, it is unclear from the reading of the current manuscript if the authors support the previous UP connection to South East Asia (through Taiwan, and I assume also through Ryukyu) of the Jomon ancestry; if this need to be revised, maybe the authors should mention it in the main text or in the suppl. If not -because they are dealing with much more recent events, related to farming- maybe they should mention the previous work on Jomon ancestry,

considering it is quite relevant for this paper.

In conclusion, I feel the authors could try to integrate all this available paleogenetic information in a comprehensive and general scenario (maybe in the Discussion that right now almost lacks any reference to previous works).

Minor points

The paper starts with a sentence on previous paleogenetic papers dealing with linguistic and migrations in western Eurasia. In my view the choices of the four examples is a bit strange; ref 4 deals with Iron Age genomes from Tianshan (Central Asia, technically not western Eurasia), ref 1 is from 2018 and is posterior to ref 2 (2015). I would reorganize this as: ref 2,3,1 and maybe 4.

The choice of ref 5 to illustrate the few truly interdisciplinary works -strongly focussed on genetics- is a bit odd; it is a kind of general review published in a Russian journal that provides no new genetic data whatsoever.

Some SI pdfs have labels cut because they were converted from excel tables with limited-space columns. If they have to be published only as excel files, then it will be fine.

Referee #3 (Remarks to the Author):

A. This paper brings together linguistic, archaeological and genetic (including aDNA) datasets to locate in time and space the origins of the far-flung Transeurasian language family. It applies similar cladistic methods to each of these datasets. All three datasets point towards an centre of origins and divergence focused in Northeast China from which various migrations are indicated by genetics, dispersals are indicated for cultural attributes, including agricultural and textile practices, and from which the various subclades of the language family also originate. Taken together these data argue for an initial expansion and break up of the Trans-Eurasian family connected with the origins and spread of agriculture, based on millet (*Panicum miliaceum*), possibly a ramie (*Boehmeria*) [note that this is presumably *Boehmeria silvestrii*, but not *B. nivea*, the more widespread ramie crop, which originated further south like in the Yangtze).

This makes TransEurasian an apparently clear instance of the language-farming dispersal hypothesis.

B. I found these results novel and compelling. For many years lip service has been paid to the need to integrate linguistics, genetics and archaeology, but there are actually few studies that have done this successfully. The Language-Farming dispersal hypothesis is normally attributed to either Colin Renfrew's work on European Neolithic & Indo-European or Peter Bellwood's Austronesian origins work. But neither of them actually did a good job of analytically integrating these datasets (in terms of detailed etymologies, population genetic relationships and archaeology). Of course, recent years has made the genetic story much more accessible as aDNA has become available. Still, I think the current "triangulation" paper is best study I have seen that empirically integrates these datasets.

C./D. The data set is large and impressive. The analyses, especially on linguistics and genetics follow current, established analytical methods and these are well presented. The supplementary tables provide a wealth of detail to explore. Tables of etyma and cognates are provided to back up the linguistic analysis. Details of the genetics sequences and the time and space distribution of

archaeological skeletons used for aDNA. Also a larger data table of archaeological traits (presence/absence of artefact types, crops and animals), as well as directly dated crops. There is a wealth of data here.

My main criticisms and suggestions for improvement, however, relate to making clear issues around the compilation of the archaeological dataset and statistical methods for its analysis; these are not as well justified as those applied to languages and genomes- which follow well-trodden procedures. What is more methodologies for inferring demography from archaeological data, including radiocarbon dates, are not clarified. Nor is reference made to robust Bayesian statistical protocols that have become established for summed radiocarbon dates in the last few years. I also found a lack of reference to previous cladistic approaches to archaeological data, which either should be cited as methodological precursors or flagged as differing from the methods used in this study, if they do differ. (see below)

E. The conclusions are clear, supported by the evidence.

F. Suggested improvements:

In the main paper it is stated only that "Building on previous studies, we provided an overview of demographic changes associated with the introduction of millet farming across the regions in our study"... And that there were population increased in the Neolithic and again in the Bronze Age. This is supported by reference to Extended data Fig. 3. But it is this extended data Fig 3, and the associated discussion of methods, that require improvement.

Supplementary Information 7 describes in detail the compilation of the archaeological dataset, and provides an interpretative summary of this information. It does not, however, detailed the methods used to derive demographic information from the proxies in Fig. S3.

More information, use of appropriate citation and methods are needed on these. There appear to be three kinds of demographic proxies, all used, without reference to any drawbacks (1) site counts in a given region over time; (2) summed probability distribution of radiocarbon dates; (3) a population estimate in human numbers for Japan (derivation unclear). All of these are potentially problematic without clarification, and probably some statistical reworking.

(1) some discussion of representativeness of site counts is in order. Just because a site falls into an archaeological period that lasts 500 or 1500 years does not mean the site was occupied for that entire span, and given that difference archaeological periods are difference lengths this will tend to overestimate populations for longer periods. There are two ways to deal with this, either to number of sites/100 yrs (as used in Stevens and Fuller 2017 or Leipe et al 2019, *Scientific Adv.* – both already cited); or take a aoristic approach (which divides sites counts into fractional units to correct for time differences), an approach introduced in

Palmisano, A., Bevan, A. and Shennan, S., 2017. Comparing archaeological proxies for long-term population patterns: An example from central Italy. *Journal of Archaeological Science*, 87, pp.59-72.

(2) SPDs of radiocarbon dates are now pretty standard fare, but it is inadequate just to sum them, as this may over represent sites or cultures that have had more dating (and more funding) thrown at them. More appropriate is the bin sites and phases and to represent the dates of each sites as a single summed unit before adding these to other sites. Also patterns in these data need to be tested for significance, both against fluctuations that result from the radiocarbon calibration curve and from nulls models of no demographic affect (which usually assumes background steady population growth). Both can be tested using now well-established Bayesian methods. These were introduced in 2013/2014 mainly in the context of European Neolithic, esp.

Timpson, A., Colledge, S., Crema, E., Edinborough, K., Kerig, T., Manning, K., Thomas, M.G. and Shennan, S., 2014. Reconstructing regional population fluctuations in the European Neolithic using radiocarbon dates: a new case-study using an improved method. *Journal of Archaeological Science*, 52, pp.549-557.

This approach has standard R script that are now available that were brought out with the study of Bevan et al (2017)

Bevan, A., Colledge, S., Fuller, D., Fyfe, R., Shennan, S. and Stevens, C., 2017. Holocene fluctuations in human population demonstrate repeated links to food production and climate. *Proceedings of the National Academy of Sciences*, 114(49), pp.E10524-E10531.
Also see a recent issue of *Holocene* devoted to demographic proxies of the Mediterranean including site count based and SPD based approaches: *The Holocene* 29(5) [2019]

(3) In this case it is not clear what these population estimates are actually based on; perhaps they refer to counts of pit dwellings, but this should be clarified.

As it stands the caption for Fig S3 is misleading, C14 SPD is not same as population density, as will be seen from the sources above at best SPDS gives direction of travel of population growing or falling) and not absolute size. Site count data can give absolute size, but one also has to factor site size (or assume that all sites are roughly the same size?). In any case this can be improved in revision.

Finally, the authors cladistic approach to an archaeological data matrix appears to follow the same methods as that applied to languages data, using current Bayesian methods/ BEAST, etc. This seems fine, but it should be justified against the background of how phylogenetic methods have been used in the past. These were all the rage 10-15 years ago (pre-BEAST I think) but some reference to previous methods and how this approach is similar/ different/ better would seem in order.

Examples include:

Collard, M., Shennan, S.J. and Tehrani, J.J., 2006. Branching, blending, and the evolution of cultural similarities and differences among human populations. *Evolution and Human Behavior*, 27(3), pp.169-184.

Coward F, Shennan SJ, Colledge S, Conolly J, Collard M. 2008. The spread of Neolithic plant economies from the Near East to Northwest Europe: a phylogenetic analysis. *J. Archaeol. Sci.* 35:42-56

Buchanan, B. and Collard, M., 2007. Investigating the peopling of North America through cladistic analyses of Early Paleoindian projectile points. *Journal of Anthropological Archaeology*, 26(3), pp.366-393.

Chickens. In the archaeological review (§3.1) they site the alleged aDNA evidence for chickens in NE China. This evidence is specious. Bones sequenced were from pheasants and produce galliform but *Gallus* sequences, as I understand it. The critique needs to be cited:

Peters, J., Lebrasseur, O., Best, J., Miller, H., Fothergill, T., Dobney, K., Thomas, R.M., Maltby, M., Sykes, N., Hanotte, O. and O'Connor, T., 2015. Questioning new answers regarding Holocene chicken domestication in China. *Proceedings of the National Academy of Sciences*, 112(19), pp.E2415-E2415.

Also the genomic data is clear that Chickens comes from the South (SE Asia or thereabouts), and spread relatively late (Bronze Age).

Huang, X.H., Wu, Y.J., Miao, Y.W., Peng, M.S., Chen, X., He, D.L., Suwannapoom, C., Du, B.W., Li, X.Y., Weng, Z.X. and Jin, S.H., 2018. Was chicken domesticated in northern China? New evidence from mitochondrial genomes. *Sci Bull Fac Agric Kyushu Univ*, 63, pp.743-6.

Wang, M.S., Thakur, M., Peng, M.S., Jiang, Y., Frantz, L.A.F., Li, M., Zhang, J.J., Wang, S., Peters, J., Otecko, N.O. and Suwannapoom, C., 2020. 863 genomes reveal the origin and domestication of chicken. *Cell research*, 30(8), pp.693-701.

Referee #4 (Remarks to the Author):

This is an important submission which breaks new ground in its high level of collaboration between linguists, archaeologists and geneticists. Such a high level of collaboration has so far been lacking in many articles that deal with broad scale human population history, especially those emanating from the new and growing field of ancient DNA analysis.

This paper is clearly a response to a very recent publication in *Nature* by Chuanchao Wang et al., published at <https://doi.org/10.1038/s41586-021-03336-2> (2021). This Wang et al. article is not referenced in the current contribution, presumably because it only appeared in the last week or so. Wang et al. have a section entitled "Refining the Transeurasian Hypothesis", in which they suggest that Transeurasian (henceforth TE) speakers did not have a single genetic origin. Rather, they suggest that Japanese and Koreanic speakers had a Liao Basin Neolithic origin, while Mongolic, Turkic and Tungusic speakers spread later from a separate source population in the Amur Basin. Thus, they challenge the existence of a unified TE language family as a relevant concept in understanding the history of NE Asian human populations. The Wang et al. paper, however, is focused on ancient DNA analysis, and does not discuss in detail any evidence from archaeology or linguistics.

The current contribution by Robbeets et al. opposes the view of Wang et al. and suggests a unified origin for the whole TE family in the Liao Basin Neolithic. These two papers are therefore at loggerheads on this issue, although I must confess to some confusion in seeing some authors listed between both papers, thereby presumably contributing their information willingly to the genesis of opposed conclusions. This is not of particular concern to me and I can understand that this situation is quite common with massively multi-authored papers, such as the two under discussion here, where individual authors do not necessarily contribute anything to the actual writing of the paper. This observation does not affect the contents of the new Robbeets et al., but it has always struck me as being a little odd. My own perspective on the issue suggests to me that Robbeets et al. are correct as far as TE origins are concerned.

I need to point out here that I am an archaeologist with a broad general knowledge of Asian population history. I do not have specialised skills in linguistics, genetics or statistics. However, the general conclusions of this paper, that the TE language family originated amongst early cultivators of millet in the Liao Basin, and then underwent initial Neolithic expansions eastwards towards the Amur basin and Korea, followed by more recent Bronze and Iron Age expansions involving the ancestral Mongolic, Tungusic, Turkic and Japanese speakers, are strongly supported and clearly stated.

There is no single reason why I prefer the conclusions of Robbeets et al. over those of Wang et al., but it seems to me that Robbeets et al. present more detailed analysis of relevant northeast Asian data from all three of the main research fields involved in the debate, especially with regard to the TE origins issue. They also make a good case for the root genetic component of the early TE speaking population being Amur River related, and for this component becoming later mixed with genetic contributions from Yellow River (presumably Sino-Tibetan speaking) populations. This all makes good sense in terms of the geography and early Neolithic archaeology of the whole region – Yellow/Liao/Amur.

On reading through the paper I had a few minor comments.

Line 95: the two references here should presumably be 40 and 41, not 43 and 44, which do not exist.

The caption to Figure 1b would better read Reconstructed locations of Transeurasian ancestral languages....., rather than giving the impression (as now) that these were once real boundaries. Also, the distribution of Proto-Turkic seems very large for a protolanguage, especially its western extension.

Line 114, The sentence "These dates estimate the time depth of the break-up of a given language family into its subfamilies" is a little unclear. Presumably it should state something like "These dates estimate the time depth of the initial break-up of a given language family into more than one foundation subgroup/subfamily".

Line 121: "After a primary break-up of the family in the Neolithic, further dispersals took place in the Bronze Age". The dates listed in this paragraph are hardly Bronze Age. I suggest this be changed to "late Neolithic and Bronze Age". The same applies in the other direction to line 138, since many of the listed expansions were Iron Age.

Lines 152-155: remove the duplication of "...in northern China, the Primorye, Korea and Japan".

Line 169: just listing 'stone tools' seems rather unconvincing. What kinds of stone tools are referred to?

Line 238: the suggestion of a Nagabaka origin amongst the Jomon rather than Austronesians from Taiwan makes good sense to me from an archaeological perspective. The Ryukyus do not really have an archaeological presence from Taiwan.

Lines 291 and 292: "Together with the Jomon profile discovered at Yokchido in Korea, our results show that Jomon genomes and material culture did not always overlap". This sentence would make more sense if readers could be told what was the material culture at Yokchido. Chulmun? Jomon? Are there traces of a former Jomon presence in Korea?

Extended data Figure 1 puzzled me a little since the caption states "Dated Bayesian phylogeny...". I cannot see in the figure any numbers that I would recognise as dates. I think more explanation is needed here.

I hope that Nature can give this article very serious consideration. It presents a different perspective on an issue just covered by Wang et al. in Nature, and in my view it breaks new ground in that it actually brings together linguists, archaeologists and geneticists to decide amongst themselves what exactly they wish to express about an issue of considerable importance in human population history. I find the unified origins view of TE ancestry put forward by Robbeets et al. to be the most convincing explanation for the situation in terms of comparative evidence from the linguistic and ancient DNA origin scenarios currently being put forward for many other language families across the world.

The authors should also note that another recent paper by archaeologists supports a very early expansion of Neolithic people with millet cultivation from the lower Yellow River in Shandong (Guiyun Jin et al., *Antiquity*, December 2020, p. 1426–1443 – on the Beixin culture). This is perhaps not directly relevant for TE issues, but it does support the idea that early farmers in northern China were entering an expansion mode as early as 6500 BC.

Referee #5 (Remarks to the Author):

1. Summary of the key results

This study proposes a method or technique of "triangulation" whose aims to combine the linguistics, archaeology and genetics evidences to explain the origin and dispersal of Transeurasian languages. It is quite attractive approach, given that the whole picture can theoretically be reconstructed from the proper integrations of these three interdisciplinary aspects. Despite the ambition, this work is quite far from what would be expected. In fact, the work provided linguistic, archaeological and genetic evidences independently but not integrally. It is therefore not suitable to be published in Nature.

Detailed comments as follows:

- (1) Bayesian phylogeny of Transeurasian languages dated the first divergence in 9,181 BP and inferred its homeland in the West Liao River in Northeast Asia. In fact, these phylogenetic results are not established on the reliable time calibrations on the internal nodes of the phylogeny (e.g. divergence/origin of Tungusic). These calibrations seemed self-determined without any reference supported and listed in this manuscript.
- (2) Using a large vocabulary list rather than using Swadesh 100 or 200 word-lists is not an advisable choice. Especially, the Swadesh 100 word-list show a high linguistic quality and great explanatory power for language evolution in previous studies. Also, there is no clear definition for 'the datapoints' in line 1-7 in P.107. Do they refer to the cognate sets? It seems not a standard terminology in the linguistics field.
- (3) According to the supplementary file of language tree, some Transeurasian languages showed unstable evolutionary rates along the branches of the phylogeny, such as middle_Mongolian_Secret_History. It is suspected that the heterogeneity in evolutionary rates of languages should not be affected by the prior settings of clock model in BEAST, but could be strongly biased to the reliable divergence time estimation.
- (4) The descriptions of the phylogenetic reconstruction is very poor in the manuscript. Even, it does not provide the number of iteration, burn-in setting, and autocorrelation and convergence status after the reconstruction, posterior values for model parameters estimated in BEAST, and the parameter (i.e. Bayes Factor) for the model comparison and selection....
- (5) The homeland of Transeurasian languages are inferred via three methods. But the rationale of these methods is to find out the center of high linguistic diversity. The efficiency and accuracy of these inference approaches are recently evaluated and questioned (Wichmann and Rama, 2020, Testing methods of homeland detection using synthetic data; Neureiter et al., 2021, Can Bayesian phylogeography reconstruct migrations and expansions in linguistic evolution?). However, the manuscript seems to ignore their discussions.
- (6) Note that there are no strong linguistic evidence supporting 'Farming Hypothesis'. In fact, the manuscript does not provide strong linguistic evidence to completely reject the alternative hypothesis of pastoralist although they use the traditional historical linguistic comparative method to reconstruct proto-type of the words associated with cultivation, food production and preservation, millet and rice, and domesticated animals such as dog. The homology of these words can probably be generated by the lexical borrowings in the ancient period.
- (7) The samples in language phylogeny cover the large geographic areas across the Europe and Asia. In contrast, the archaeological analyses in this paper make concerns on the area of Northern China. In my intuition, the samples in Northern China cannot represent the whole Transeurasian languages or populations even under the Pastoralist Hypothesis.
- (8) From archaeological perspective, the manuscript posts the archaeological data of Neolithic and Bronze Age sites in northern China compiled from the several literature and documents of, and extracted and coded the these sites. This collection and quantitative work is very essential for the "triangulation" study. However, similar to the aspect of linguistic analysis, the manuscript did not provide more detailed descriptions about the data collection and criteria of feature selection.
- (9) Based on the binary-coding archaeological features, the study reconstructed the Bayesian phylogeny via BEAST. I do not suspect the efficiency of the program BEAST, but MrBayes could be

more suitable for reconstructing cultural evolution of archaeological sites.

(10) I am confused that in the main body of manuscript, authors used 23 individuals (perhaps ancient people? Or contemporary persons?) in line 193 but there is a lack of detail of sample collection procedure for these individuals and their authentication.

(11) The manuscript attempt to map the language populations and ancient populations (including ancient samples) from the genetic perspective, aiming to investigate the association between the ancient populations and contemporary ones speaking specific languages. It is very important to point out that such mapping cannot authenticate the languages of ancient populations because the scenario of language shift frequently occurred in the ancient periods. It is a very dangerous explanation based on this mapping.

2. Originality and significance: if not novel, please include reference

(1) Originality: not novel. It seems a synthesis of the previous works done by the authors like Ning et al. 2019, Current biology; Li et al. 2020. Archaeological Research in Asia; Ning et al. 2020, Nature Communication and so on.

(2) Significance: good.

3. Data & methodology: validity of approach, quality of data, quality of presentation

(1) Validity of approach: Bayesian phylogenetic analysis for linguistics is the state of the art in the field of language evolution. Applying MrBayes on the archaeological data is more reasonable than using BEAST. The approaches of genetic analyses are valid.

(2) Quality of data: no confirmation for linguistic data. High for archaeological data. And no confirmation for genetic data due to lack of the explicit genetic quality report for the ancient DNA.

(3) Quality of presentation: Not clarity for the data collection and method usage.

4. Appropriate use of statistics and treatment of uncertainties

(1) The use of statistics is right. But there is no explicit statistical descriptions for the bars in Fig 1b.

5. Conclusions: robustness, validity, reliability

Robustness: No

Validity: No

Reliability: No

6. References: appropriate credit to previous work?

No.

7. Clarity and context: lucidity of abstract/summary, appropriateness of abstract, introduction and conclusions

Lucidity of abstract/summary: Yes

Appropriateness of abstract: Yes

Appropriateness of introduction: No

Appropriateness of conclusions: No

Author Rebuttals to Initial Comments:

**Documentation of revision**
**“Triangulation supports agricultural spread of the Transeurasian languages”**

(1) Reviewer 2: Overlapping with Wang et al. (Genomic Insights into the Formation of
Human Populations in East Asia). The authors cite this work when it was at BioRxiv (ref 7),
but I think they should mention at least this conclusion [i.e. Yellow River Basin farmers
likely spread the Sino-Tibetan languages], as this seems to provide genetic evidence for
dispersion centres for farming in Asia, this one in particular I guess associated to rice
farming. Integrate information in a comprehensive and general scenario, maybe in the
Discussion that right now almost lacks any reference to previous works.

We adapted the reference in the manuscript as follows

Wang, C. C., Yeh, H. Y., Popov, A. N., *et al.* Genomic insights into the formation of human populations in East
Asia. *Nature* **591**, 413-419 (2021).

On line 214-217 we added: “This contradicts a recent genetic study (**Wang et al. 2021**)”, which concludes
the absence of Yellow River influence in ancient genomes from Mongolia and the Amur does not support the
West Liao genetic correlate of the Transeurasian language family”.

On line 224 we added a reference to Wang et al. 2021: “As Amur-related ancestry can be traced back
to speakers of Japanese and Korean (**Wang et al. 2021**)”

One line 299-302, we added: “By advancing the first evidence from ancient DNA, our research thus
confirms recent findings that Japanese and Korean populations have West Liao River ancestry, while it
contradicts previous claims that there is no genetic correlate of the Transeurasian language family (**Wang et al.**
**2021.**)”

On line 271-275, we added: “In line with recent associations between the Sino-Tibetan family estimated at
8000 BP (Sagart et al. 2019; Zhang et al. 2020) and Neolithic farmers from the Upper and Middle Yellow River
(Ning et al. 2020, **Wang et al. 2021**), our results associate the two centers of millet domestication in Northeast
Asia with the origins of two major language families: Sino-Tibetan on the Yellow River and Transeurasian on
the West Liao River.”

(2) Reviewer 2: Partial overlapping with Chao Ning et al. (2020, Nat Comm, ref 6, again
shared with some coauthors) as they provide evidences of correlation between
archaeolinguistic signatures and past human migrations in northern China. Again, I think the
genetics section (or the Discussion) would benefit of mentioning explicitly these previous
findings that preclude some of the findings of the current work

We added additional reference to Ning et al. in the genetics session and in the discussion as
follows:

In the genetics section, line 206: “Contemporary Tungusic as well as Nivkh speakers in the Amur form a
tight cluster (**Ning et al. 2020**)”

In the genetics section, line 212 “West Liao Neolithic millet farmers show a considerable proportion of
Amur-like ancestry with a gradual shift towards the Yellow River genome over time (Extended data Fig. 7, Fig.
3b). (**Ning et al. 2020**)”

In the discussion, on line 271-275, we added: “In line with recent associations between the Sino-
Tibetan family estimated at 8000 BP (Sagart et al. 2019; Zhang et al. 2020) and Neolithic farmers from the
Upper and Middle Yellow River (**Ning et al. 2020**, Wang et al. 2021), our results associate the two centers of

millet domestication in Northeast Asia with the origins of two major language families: Sino-Tibetan on the
Yellow River and Transeurasian on the West Liao River.”

(3) Reviewer 2: Mention Gakuhari et al. (2020) *Comm. Biol.*

The reviewer correctly notes that our findings with respect to ancient DNA from Korea raise
potential new questions about Jomon origins and population history. As also noted by the
reviewer, our paper deals primarily with more recent events related to farming expansions in
the Neolithic and Bronze Age. The origins of the Jomon people are outside of the main focus
of the present research. Nevertheless, we have added several new comments on Korea in SI
13. We have cited Gakuhari et al (2020) and Kanzawa-Kiriyama et al. (2019) in the
supplementary information. Since the number of references for the main text is strictly
limited by *Nature*, and because our paper requires references to all three disciplines of
linguistics, archaeology and genetics, we hope the reviewer will understand our not citing
these papers in the main text.

(4) Reviewer 2: Reorganize ref 2,3,1 and delete reference 4

In line with the reviewer’s advice, we reorganized the references in chronological order as 1
Haak et al. 2015, 2 Allentoft et al. 2015 and 3 Damgaard et al. 2018. We deleted the
reference to Ning et al. 2019 as we agree with the reviewer that it is not essential here.

(5) Reviewer 2: The choice of ref 5 to illustrate the few truly interdisciplinary works -strongly
focussed on genetics- is a bit odd; it is a kind of general review published in a Russian
journal that provides no new genetic data whatsoever.

The reviewer is right. We added the description “or limited to reviewing existing datasets”
for our reference to Mallory et al. 2019.

(6) Reviewer 3: a ramie (*Boehmeria*) : this is presumably *Boehmeria silvestrii*, but not *B.*
*nivea*

We thank the reviewer for pointing this out and we changed the name in the agropastoral
vocabulary in SI 5.

(7) Reviewer 3: Extended data Fig 3, and the associated discussion of methods, require
improvement. Supplementary Information 7 describes in detail the compilation of the
archaeological dataset, and provides an interpretative summary of this information. It does
not, however, detail the methods used to derive demographic information from the proxies in
Fig. S3. More information, use of appropriate citation and methods are needed on these. There
appear to be three kinds of demographic proxies, all used, without reference to any
drawbacks (1) site counts in a given region over time; (2) summed probability distribution of
radiocarbon dates; (3) a population estimate in human numbers for Japan (derivation
unclear). All of these are potentially problematic without clarification, and probably some
statistical reworking.

A description of the three kinds of proxies has been added to §3.9 (Demography) in SI 7 to
explain the various methods used for Extended Data Fig. 3. The data used for the graph were
extracted from the published literature and are based on different criteria (site numbers for
China and the Primorye, radiocarbon SPDs for Korea, and population estimates from site
numbers for Japan). In the new text we discuss in some detail how the various numbers were

derived. We have also added an explanation that our intention here is to simply show
demographic *trends* over time, not to provide an original analysis of this topic. Given that the
original data sets derive from different methods, we have not attempted any statistical re-
working of the data.

We added the following reference to the caption of Extended Data Fig. 3
“For references and methods used to derive demographic information from the proxies, see SI 7.”

The text added is as follows

“We collated published demographic estimates for Neolithic and Bronze Age Northeast Asia and
plotted them together in Extended Data Fig. 3. The red bars in the figure show the first evidence for millet
cultivation based on direct radiocarbon dates. The site numbers for the West Liao basin and Liaoning province
were extracted from an analysis of 51,074 archaeological sites in China dating to ca. 8000 – 500 BC. These data
were taken from the 25 published volumes of the *Atlas of Chinese Cultural Relics*. Site numbers for the eastern
Inner Mongolia Autonomous Region are used as a proxy for the West Liao basin. The site numbers shown in
ED Fig. 3 for the West Liao and Liaoning are totals for each time period.

For Korea we used summed probability distributions of radiocarbon dates analysed in a recent study¹⁴³.
Building on the high number of radiocarbon dates available in South Korea, that study used 3127 dates from
Chulmun and Mumun pit houses. Some 94% of the dates were on charcoal thought to derive from the wooden
structures of the houses which were likely re-built every decade or so. Multiple dates from the same house were
combined and temporal outliers and samples with a standard error >100 years were removed to leave 2190 dates
from 513 sites ranging between 5900 – 2200 cal BP. The generated SPD was tested using methods from
Timpson et al. (2014) and Crema et al. (2016).

For the southern Primorye, we used estimates of site numbers provided by Irina Zhushchikhovskaya of
the Institute of History, Archaeology and Ethnology of the Peoples of the Far East. These estimates are 55-70
for the pre-millet farming Neolithic Boisman and Rudnaya cultures, around 150 for the Zaisannovka culture,
and 200-300 for the Bronze Age Yankovskaya culture. In Fig. 3 we used the higher end of these estimates since
archaeological survey work is still scarce in the region and more sites would plausibly be found in future
research.

For Japan, we used the population estimates made by Koyama in the 1970. Koyama collected data on
site numbers and size for mainland Japan excluding Hokkaido and Okinawa from the Initial Jōmon to the Yayoi
periods. Population was calculated using the relationship between site size and populations as recorded in early
historic tax records with a correction factor applied for Jōmon sites. Our graph shows Koyama’s estimates for
the total population of mainland Japan. Methodologically, Koyama’s estimates are perhaps the weakest of the
proxies used for Fig. 3. However, later research has supported the general trends identified by Koyama using a
range of approaches at various regional scales, including site and pit house numbers as well as radiocarbon dates

The graphs A1 and C1 in Extended Data Fig. 3 show proxies for demographic change following the
initial adoption of millet farming in the West Liao basin and Korea. The West Liao graph in A1 shows changes
in pottery found during surface surveys covering a total of 1339 km² in the Chifeng and Fuxin areas. Varying
quantities of pottery found during these surveys are hypothesised to scale to population density. The graph for
Korea in C1 is taken from the same radiocarbon SPD analysis used for the graph in C.

Extended data Fig. 3 is designed to show broad demographic *trends* in Northeast Asia in the Neolithic
and Bronze Ages. Despite the use of different proxies in the published literature, the trends identified are
comparable, especially the large population increase reconstructed for the Bronze Age in all regions.”

(8) Reviewer 3: Nor is reference made to robust Bayesian statistical protocols that have
become established for summed radiocarbon dates in the last few years.
SPDs of radiocarbon dates are now pretty standard fare, but it is inadequate just to sum them,
as this may over represent sites or cultures that have had more dating (and more funding)
thrown at them. More appropriate is the bin sites and phases and to represent the dates of
each sites as a single summed unit before adding these to other sites. Also patterns in these

data need to be tested for significance, both against fluctuations that result from the
radiocarbon calibration curve and from null models of no demographic effect (which usually
assumes background steady population growth). Both can be tested using now well-
established Bayesian methods. These were introduced in 2013/2014 mainly in the context of
European Neolithic, esp.

Timpson, A., Colledge, S., Crema, E., Edinborough, K., Kerig, T., Manning, K., Thomas,
161 M.G. and Shennan, S., 2014. Reconstructing regional population fluctuations in the European
Neolithic using radiocarbon dates: a new case-study using an improved method. *Journal of*
*Archaeological Science*, 52, pp.549-557.

This approach has standard R script that are now available that were brought out with the
study of Bevan et al (2017)

Bevan, A., Colledge, S., Fuller, D., Fyfe, R., Shennan, S. and Stevens, C., 2017. Holocene
fluctuations in human population demonstrate repeated links to food production and climate.
*Proceedings of the National Academy of Sciences*, 114(49), pp.E10524-E10531.

Also see a recent issue of *Holocene* devoted to demographic proxies of the Mediterranean
including site count based and SPD based approaches: *The Holocene* 29(5) [2019]

As noted also in the preceding response to question 7 and in our revision of SI 7, the data we
used for Korea derives from a published analysis of Neolithic and Bronze Age radiocarbon
dates by Oh and colleagues. In that study, the generated SPD was tested using methods from
Timpson et al. (2014) and Crema et al. (2016).

(9) Reviewer 3: I also found a lack of reference to previous cladistic approaches to
archaeological data, which either should be cited as methodological precursors or flagged as
differing from the methods used in this study, if they do differ.

The authors cladistic approach to an archaeological data matrix appears to follow the same
methods as that applied to languages data, using current Bayesian methods/ BEAST, etc. This
seems fine, but it should be justified against the background of how phylogenetic methods
have been used in the past. These were all the rage 10-15 years ago (pre-BEAST I think) but
some reference to previous methods and how this approach is similar/ different/ better would
seem in order.

We thank the reviewer for pointing this out. By adding the following explanation of the
merits of the Bayesian approach to the methods section, line 111-120, we hope to have
addressed the concerns:

"There is a large amount of phylogenetic work with archaeological data done (Mace et al, 2005), some
parsimony based (O'Brien et al, 2002), or distance based (Allaby et al, 2008). The benefit of Bayesian
approaches is that it is model based, has sound formal mathematical foundations in probability theory allowing
193 us to estimate uncertainty around all estimates, and it allows integration of information from various sources in a
194 single analysis (like cognate and geographic data) based on probability theory, unlike the other approaches.
BEAST aims specifically at inferring rooted time trees, and uncertainty of time estimates, which sets it apart
from other Bayesian packages that target unrooted trees. Furthermore, BEAST supports models that are not
available in other packages at the moment, hence the use of this package."

Allaby, R.G., Fuller, D.Q. and Brown, T.A., 2008. The genetic expectations of a protracted
model for the origins of domesticated crops. *Proceedings of the National Academy of*
*Sciences*, 105(37), pp.13982-13986.

O'Brien, M.J. and Lyman, R.L., 2002. Evolutionary archeology: Current status and future
prospects. *Evolutionary Anthropology: Issues, News, and Reviews: Issues, News, and*
*Reviews*, 11(1), pp.26-36.

(10) Reviewer 3: Some discussion of representativeness of site counts is in order. Just because a site falls into an archaeological period that lasts 500 or 1500 years does not mean the site was occupied for that entire span, and given that difference archaeological periods are difference lengths this will tend to overestimate populations for longer periods. There are two ways to deal with this, either to number of sites/100 yrs (as used in Stevens and Fuller 2017 or Leipe et a 2019, Scientific Adv. – both already cited); or take a aoristic approach (which divides sites counts into fractional units to correct for time differences), an approach introduced in Palmisano, A., Bevan, A. and Shennan, S., 2017. Comparing archaeological proxies for long-term population patterns: An example from central Italy. Journal of Archaeological Science, 87, pp.59-72.

Further details can be found in the response to question 7 on demography and in our revision of SI 7. We used two sets of demographic proxy data based on site numbers. For NE China, the study by Hosner et al (2016) gives site totals per period (Neolithic & Bronze Age) for the regions relevant to our present paper (eastern Inner Mongolia and Jilin). For the Primorye, our data was by archaeological culture but these were combined into Neolithic and Bronze Age totals to match the Hosner et al. study

(11) Reviewer 3: Chickens. In the archaeological review (§3.1) they site the alleged aDNA evidence for chickens in NE China. This evidence is specious. Bones sequenced were from pheasants and produce galliform but Gallus sequences, as I understand it. The critique needs to be cited; se references. Also the genomic data is clear that Chickens comes from the South (SE Asia or thereabouts), and spread relatively late (Bronze Age).

We thank the reviewer for these references, which we have added to SI 7 §3.1.

(12) Reviewer 4: opposes the view of Wang et al. Make more prominent in discussion. See also our response to question (1).

On line 214-217 we added: “This contradicts a recent genetic study (Wang et al. 2021), which uses the absence of Yellow River influence in ancient genomes from Mongolia and the Amur to deny the West Liao genetic correlate of the Transeurasian language family.”

One line 299-302, we added: “By advancing the first evidence from ancient DNA, our research thus confirms recent findings (Wang et al. 2021) that Japanese and Korean populations have West Liao River ancestry, while it contradicts the claim made in that study that there is no genetic correlate of the Transeurasian language family.”

(13) Reviewer 4: Line 95: the two references here should presumably be 40 and 41, not 43 and 44, which do not exist.

We thank the reviewer for pointing this out and corrected the numbers of the references in Line 95.

(14) Reviewer 4: The caption to Figure 1b would better read Reconstructed locations of

254 Transeurasian ancestral languages....., rather than giving the impression (as now) that these
255 were once real boundaries. Also, the distribution of Proto-Turkic seems very large for a
256 protolanguage, especially its western extension.

The reviewer is right. We changed the caption on line 102 to “Fig. 1b. Reconstructed locations of
Transeurasian ancestral languages...”

For the reconstruction of homelands, we integrated three different homeland detection
methods, as explained in SI 4. In general, the validity and credibility of the locations
increases by integrating the different methods. However, in the case of Proto-Turkic,
integration leads to a rather large distribution due to the fact that Bayesian phylogeography
and cultural reconstruction situate the homeland too much to the west. This is because the
contemporary distribution of Turkic languages is not representative of the earlier distribution,
as most descendants of West Old Turkic have been erased by descendants of East Old Turkic.
Moreover, due to its early break-up from Altaic, Proto-Turkic covers a long period from the
Middle Neolithic to the Early Iron Age. We should thus interpret the Proto-Turkic homeland
on the map as a dynamic entity, gradually expanding from Southeast to Northwest from the
Middle Neolithic to the Early Iron Age.

In order to direct the reader to this information, we added a reference in the caption of Fig.
1b. on line 103-104: “For detailed homeland detection, see SI 4.”

In SI 4, we added the following explicitation on line 232-235: “The integration of the three
methods of homeland detection will thus lead to a rather large distribution, stretching from present-day Inner
Mongolia and Shanxi in the east to North Kazakhstan in the west; see Section 4.10.2.”

In SI 4, line 496-504, we explained: “In the case of Proto-Turkic, the bias caused by the domination
of the descendants of Eastern Old Turkic led to a rather large distribution of the homeland, stretching from
present-day Inner Mongolia and Shanxi in the east to North Kazakhstan. The southeastern homeland indicated by
cultural reconstruction probably represents the best approximation of the original location of Proto-Turkic, at a
time following its separation from Altaic. Due to this early break-up, Proto-Turkic covers a long period from the
Middle Neolithic to the Early Iron Age. Therefore, the Proto-Turkic homeland on the map in Fig. SI 4.10 can be
considered as a dynamic entity, gradually expanding from Southeast to Northwest from the Middle Neolithic to
the Early Iron Age.”

(15) Reviewer 4: Line 114, The sentence “These dates estimate the time depth of the break-
up of a given language family into its subfamilies” is a little unclear. Presumably it should
state something like “These dates estimate the time depth of the initial break-up of a given
language family into more than one foundation subgroup/subfamily”.

The formulation proposed by the reviewer is indeed more precise. On line 116-117, we
reworded as follows: “These dates estimate the time depth of the initial break-up of a given language family
into more than one foundation subgroup.”

(16) Reviewer 4: Line 121: “After a primary break-up of the family in the Neolithic, further
dispersals took place in the Bronze Age”. The dates listed in this paragraph are hardly Bronze
Age. I suggest this be changed to “late Neolithic and Bronze Age”. The same applies in the
other direction to line 138, since many of the listed expansions were Iron Age.

We added ‘in the late Neolithic’ to line 125.

(17) Reviewer 4: Lines 152-155: remove the duplication of “..in northern China, the
Primorye, Korea and Japan”.

We removed the reduplication on line 157-158

(18) Line 169: just listing ‘stone tools’ seems rather unconvincing. What kinds of stone tools
are referred to?

On Line 169 we have changed to ‘stone tools for cultivation and harvesting’

(19) Lines 291 and 292: “Together with the Jomon profile discovered at Yokchido in Korea,
our results show that Jomon genomes and material culture did not always overlap”. This
sentence would make more sense if readers could be told what was the material culture at
Yokchido. Chulmun? Jomon? Are there traces of a former Jomon presence in Korea?

Further details on the Jomon material culture found at Yokchido have been added to SI 12,
line 147-154:

“In addition to local Chulmun ceramic wares, several types of Jōmon artefacts from Kyushu have been found at
Yokchido. These include Middle Jōmon Funamoto II and small quantities of Late Jōmon pottery (several styles
related to the Adaka type), arrowheads and harpoons made of obsidian and sanukite, and a so-called Kamasaki-
type scraper. Yokchido has also produced two small, crudely-made clay figurines interpreted as depicting a wild
boars from Middle Chulmun contexts. These are plausibly interpreted as an influence from Japan where more
than 70 boar figurines are known from the Jōmon period. Specifically, the Yokchido figurines have been
compared to a similar figurine from the Miyashita shell midden on Gotō island, Nagasaki”

The site is basically Chulmun but with some imported Jomon pottery and stone tools. As
regards the question of a ‘former Jomon presence in Korea’, previous research has assumed
that artefacts shared between Korea and Japan (mainly Kyushu) during the Jomon resulted
from trade and exchange between the two regions, with Japanese obsidian for example being
brought to the peninsula.

In addition, we added information about Jomon influences observed at other sites on the
Korean peninsula. For Ando shell midden, line 69-70: “The site also has sherds of Initial Jōmon
pottery and obsidian, presumably from sources in Japan.”; for Yōndaedo, line 174-178: “Evidence of
interaction with northern Kyushu consists of Early Jōmon Todoroki B and Middle Jōmon Kasuga pottery. Large
quantities of obsidian are also reported, both as raw materials and finished tools including arrowheads (86% of
tools) and scrapers. Sanukite debitage was also found together with sanukite arrowheads, scrapers and harpoon
heads including a harpoon type typical of northwest Kyushu”

As another reviewer has pointed out, the DNA analysis in our paper suggest for the first time
the possibility of a broader and deeper Jomon presence on the Korean peninsula. We have
added some notes on this issue in SI 13.

The text added is as follows:

SI, 13Line 64-72: “However, it is presently unclear to what extent this admixture reflects extensive ancient
Jōmon ancestry on the peninsula or more recent exchange. Archaeologists have so far only discussed the latter
possibility. There is evidence for the exchange of pottery, obsidian, sanukite, fishing tools and ornaments
between Neolithic Korea and Kyushu. Previous research has identified 23 or 24 sites in South Korea with
evidence of interaction with Jōmon Japan and 27 Jōmon sites in Kyushu with Chulmun pottery from the
peninsula. Despite such interaction, many archaeologists have previously concluded that a basic cultural or
ethnic boundary was maintained between Kyushu and Korea.”

SI 13, Line 78-87: “The broader question of Jōmon genetic ancestry on the Korean peninsula requires
further research. The idea of a Southeast Asian origin for Jōmon populations has a long history, drawing
originally on von Eickstedt’s concept of ‘Palaeomongoloids’ and ethnological theories of links between
Austronesians and the Ainu¹. By the 1970s, studies of skeletal and dental morphology were showing links
between Jōmon and Southeast Asian populations, although alternative interpretations included links with
Polynesians. The Southeast Asian origin of the Jōmon continues to be discussed from a genomic perspective.
However, whatever their ultimate origins, it is likely that many Jōmon populations reached the Japanese
archipelago via Korea, presenting the possibility of a very ancient Jōmon genetic heritage on the peninsula.”

(20) Extended data Figure 1 puzzled me a little since the caption states “Dated Bayesian
phylogeny...”. I cannot see in the figure any numbers that I would recognise as dates. I think
more explanation is needed here.

The dates in units of 100 years BP are marked on the nodes of the tree (i.e. root at 91.8311
means 9183 BP). Given their small size, they are hardly visible. In addition to the pdf, we
have the same file in a tree format, that allows the reader to play with the data, make
probabilities, credible intervals, branch lengths, etc. clearly appear or disappear. However,
this format was not successfully submitted to the Nature submission site. Therefore, we will
send a separate *wetransfer* to the Nature senior editor including all special formats.

(21) Reviewer 5: Bayesian phylogeny of Transeurasian languages dated the first divergence
in 9,181 BP and inferred its homeland in the West Liao River in Northeast Asia. In fact, these
phylogenetic results are not established on the reliable time calibrations on the internal nodes
of the phylogeny (e.g. divergence/origin of Tungusic). These calibrations seemed self-
determined without any reference supported and listed in this manuscript.

We added Table SI 18.1 Time calibrations of internal nodes in the Transeurasian
phylogenetic tree to SI 18 Bayesian phylogenetic analysis of the linguistic dataset
(38_Eurasia3angle_synthesis_SI 18_Bayesian linguistics) along with a motivation of the
calibrations and added a reference to this in the method section of the manuscript.

Methods Line 31-32: “These calibrations are supported by chronological estimations proposed in linguistic
literature (SI 18).”

SI 18 Line 5-63:

**“Table SI 18.1 Time calibrations of internal nodes in the Transeurasian phylogenetic tree.**

Node	Calibration (B)CE	Calibration BP
Japonic	150 BCE +/- 175	2100 BP +/- 175
Koreanic	1150 CE +/- 175	800 BP +/- 175
Tungusic	50 CE +/- 275	1900 BP +/- 275
Mongolic	1200 CE +/- 50	750 BP +/- 50
Turkic	150 BCE +/- 175	2100 BP +/- 175

We informed the node ages of the individual language families under discussion by the age priors given in Table
SI 18.1 above. We used the dates generally proposed in linguistic literature to support our estimates of the time
intervals for each node.

Our chronological estimation of Proto-Japonic at 2100 BP +/- 175 is in line with Lee and Hasegawa’s (2011)
Bayesian phylogenetic analysis, dating Proto-Japonic divergence at 2182 BP. Classical linguistic dating
methods estimate the break-up of Proto-Japonic before 1500 BP: between 2000 and 1500 BP according to
Hattori (1976: 43) and before 1500 BP according to Frellesvig and Whitman (2008). However, the former
dating is based on somewhat controversial lexicostatistic dating methods and the latter on the association of

certain language changes with specific historical events, notably the introduction of Buddhism, which yields a
terminus ante quem, a ceiling but not a floor for the time-depth of separation. Since we know that the primary
branches Proto-Ryukyuan and Japanese remained in intensive contact after the break-up of Proto-Japonic until
at least the 8th-9th Century AD (Pellard 2015), we expect more similarities between both languages than in case
the connectivity would have been completely broken. Therefore, lexicostatistic dating methods tend to be biased
towards more shallow break-up times than the real times. By consequence, we consider the earlier part of the
spectrum of previous time estimations to be more reliable than the later part and estimate Proto-Japonic at 2100
BP +/- 175.

Since the Silla kingdom unified the Korean peninsula politically and linguistically in AD 668, erasing all
pre-existing linguistic diversity, Proto-Koreanic is much shallower than Proto-Japonic. Modern Korean dialects
show limited variation, most of which can be regarded as derived from Late Middle Korean (1446-1600)
(Whitman 2011). An exception is the Cheju dialect, which preserves some phonological and lexical features that
is thought to preserve elements from Early Middle Korean (918-1446) (Shin et al. 2020). Therefore, we
estimated Proto-Koreanic, as the ancestor of all contemporary Korean dialects at 800 BP +/- 175.

Previous research on the time-depth of Proto-Tungusic largely confirm Janhunen's (2012: 8) proposal of an
Iron Age dating (2450-1450 BP), among others based on the reconstruction of the term 'iron' to Proto-Tungusic.
This large interval of about one millennium has been reduced by Pevnov (2012: 32) estimating on the basis of a
rough measure of mutual intelligibility that Proto-Tungusic could not be younger than two thousand years and
by Robbeets situated the break-up of Proto-Tungusic in the Han period (2156-1730 BP) on the basis of
ethnonym shifts. Recently, Oskolskaya et al. (2021) propose a dating around 1500 BP using Bayesian methods.
However, this date was calculated on the basis of a separation time of 351 BP for the split between Jurchen and
Manchu-Xibe, while a realistic separation time that aligns with the model should be at least 765 BP. This
difference of at least 400 years is expected to yield a later date for the calculated time depth of Proto-Tungusic
than the real one. Therefore, we can safely estimate the Tungusic node age at 1900 BP +/- 275.

Proto-Mongolic is nearly equivalent with the language spoken by the historical Mongols around the time of
the Mongol Empire (1206-1368), which is documented in historical sources, written in several different scripts
and collectively termed Middle Mongolian. As all contemporary varieties can be derived from Middle
Mongolian, the depth of the Mongolic family is no more than 700 to 800 years (Rybatzki 2003, Robbeets et al.
2020) and we calibrate at 750 BP +/- 50. Even if there were historical languages related to Mongolic spoken in
Northeast China, such Khitan, the dynastic language of the Liao Empire (907-1125), the written traces of these
groups are too fragmentary to be included in our dataset. By consequence, these languages cannot contribute to
the calibration of the Proto-Mongolic node.

Based on evidence from contact linguistics, the earliest split between the two principal branches of Turkic,
i.e. Bulgharic and Common Turkic, is usually dated to 2450-1950 BP (Janhunen 2010). Turkic phylogenies
relying on quantitative methods basically support the lower estimate. The following dates are obtained by
lexicostatistic calculations: 2250 BP (Tenišev et al. 2001), 2070 BP (Mudrak 2009) and 1950 BP (Dybo 2007).
Savelyev and Robbeets (2019) date the split of Proto-Turkic to 2074 BP on the basis of a Bayesian analysis.
Our calibration of 2100 BP +/- 175 thus lies within the bounds generally proposed as the time-depth of the
Turkic language family.”

(22) Reviewer 5: Using a large vocabulary list rather than using Swadesh 100 or 200 word-
lists is not an advisable choice. Especially, the Swadesh 100 word-list show a high linguistic
quality and great explanatory power for language evolution in previous studies.

Our wordlist is based on the traditional Swadesh 200 word list, but it is complemented and
merged with the Leipzig-Jakarta 200 list (Haspelmath and Tadmor 2009). The Leipzig-
Jakarta list is a recently developed alternative to the Swadesh 200 word list, which takes a
more systematic and empirical approach to the basic vocabulary. As our 254 word list
includes BOTH lists it is an improvement to the Swadesh 200 word list. This is explained in
SI 2 Basic vocabulary etymologies across the Transeurasian languages, line 18-30.

“Our 254 basic vocabulary concepts are based on a merger of the Leipzig-Jakarta 200 list (Haspelmath and
Tadmor 2009) and the Jena 200 list (Anderson and Heggarty n.d.). The underlying Leipzig-Jakarta 200 list is
reduced to 195 items due to the merger of a few meanings (e.g., ‘breast’ and ‘chest’). It is supplemented by the
Jena 200 list, which is an updated version of the Swadesh 200 list, currently applied to a comparison of Indo-
European languages in the CoBL (Cognacy in Basic Lexicon) project by Anderson and Heggarty. Given the
large overlap between the two lists, the final list amounts to 254 concepts.

Compared to the traditional Swadesh list, which is mainly based on intuition, the Leipzig-Jakarta list takes a
more systematic and empirical approach to the basic vocabulary because it is based on a quantitative
comparison of stable words in the languages across the world. The strength of basic vocabulary is not in the
stability of a single concept, but in the overall stability of the body of concepts as a whole.”

(23) Reviewer 5: Also, there is no clear definition for ‘the datapoints’ in line 1-7 in P.107.
Do they refer to the cognate sets? It seems not a standard terminology in the linguistics field.

We replaced the term ‘datapoints’ in Line 107 of the manuscript text and in the title of SI 1
by term ‘cognate sets’, which will be less ambiguous for readers with expertise in linguistics.
The term ‘cognate sets’ refers to the rows in the spread sheet of SI 1. Comparative dataset
including 3193 cognate sets representing 254 basic vocabulary concepts for 98 Transeurasian
languages (16_Eurasia3angle_synthesis_SI 1_BV 254.xls).

(24) Reviewer 5: According to the supplementary file of language tree, some Transeurasian
languages showed unstable evolutionary rates along the branches of the phylogeny, such as
middle_Mongolian_Secret_History. It is suspected that the heterogeneity in evolutionary
rates of languages should not be affected by the prior settings of clock model in BEAST, but
could be strongly biased to the reliable divergence time estimation.

The uncorrelated relaxed clock employed indeed allows for uncorrelated variation among
branches, which has been shown excellent performance in dating, judging from the 5000+
citations for the Ho et al 2005 paper alone. Most of the higher rates are on the branches
among language families. Reducing the variation among rates would reduce the rate among
these branches, and as a consequence the estimate of the root age would be somewhat older,
resulting in even more support for an agricultural expansion about 8000kya.

Priors for the clock model were chosen to be uninformative. Note that uncertainty in root age
estimate is quite high and even though the posterior is more concentrated than the prior, only
substantial changes to the prior (i.e. a substantially more informative prior) will have
substantial impact on the root age estimate.

(25) Reviewer 5: The descriptions of the phylogenetic reconstruction is very poor in the
manuscript. Even, it does not provide the number of iteration, burn-in setting, and
autocorrelation and convergence status after the reconstruction, posterior values for model
parameters estimated in BEAST, and the parameter (i.e. Bayes Factor) for the model
comparison and selection....

We apologise for the inconvenience but our impression is that the reviewer was not able to
access supplementary files submitted in kml, tree and XML format. This may be due to
automatic pdf conversion of uploaded files on the Nature submission site. Therefore, we will

resubmit the entire folder of supplements to the Nature senior editor by *wetransfer*, so that he
can distribute them among the reviewers in the appropriate format.

Unfortunately, the word limit on the manuscript does not allow for extensive elaboration on
these details, but we believe that the requested information is now more clearly available in
supplementary information. Though most of the information concerning detail of the BEAST
runs is encoded in the XML files accompanying the submission, we understand that this can
be hard to read. Therefore, we added additional information in the README files SI 19 and
SI 21 detailing the BEAST analyses. (See 39_Eurasia3angle_synthesis_SI 19_XML
files_README and 41_Eurasia3angle_synthesis_SI 19_XML files_README)

Convergence was checked by running each analysis 4 times and verifying in Tracer that all
ESSs exceeded 200, and all runs converged to the same distribution by inspecting the overlap
of marginal distributions (also in Tracer). This information is added to the README as well.

Bayes factors for model comparison were provided in SI 18 and SI 20 for the language and
archeological analyses respectively. The analysis can be replicated running the XML files
available in SI 19 and SI 21 with BEAST v2.6.3 and CoupledMCMC, nested sampling and
Babel packages required for the analysis.

(26) Reviewer 5: The homeland of Transeurasian languages [is] inferred via three methods.
But the rationale of these methods is to find out the center of high linguistic diversity. The
efficiency and accuracy of these inference approaches are recently evaluated and questioned
(Wichmann and Rama, 2020, Testing methods of homeland detection using synthetic data;
Neureiter et al., 2021, Can Bayesian phylogeography reconstruct migrations and expansions
in linguistic evolution?). However, the manuscript seems to ignore their discussions.

We thank the reviewer for referring to these two useful recent papers on the topic of
homeland detection. We integrated the discussion of these papers in SI 4 (Integration of
qualitative assessment methods and Bayesian phylogeography in identifying the ancestral
homelands of Transeurasian 19_Eurasia3angle_synthesis_SI 4_homelands.docx), Section 10,
Line 401-505.

“10.1. Methods and limitations

In order to estimate the location of the ancient speech communities involved, we combined Bayesian
phylogeography, the diversity hotspot principle and cultural reconstruction in a single approach. Bayesian
phylogeography models how speech communities may have moved using a phylogenetic tree as a guideline, by
using the diffusion rate at which languages change their location to infer the spatial location of the root and all
ancestral nodes. The diversity hotspot principle assumes that the homeland is located in the geographical area
with the highest linguistic diversity with regard to the deepest subgroups in the family. Cultural reconstruction
delimits an area where the natural elements, material objects, concepts and practices revealed in the linguistic
reconstructions converge to the exclusion of other regions.

Each of these three methods has its own limitations. Bayesian methods provide a set of plausible homelands,
an area associated with the highest likelihood or posterior probability, rather than single locations. Wichmann
and Rama (2020) argue that BEAST tends to infer homeland areas that are too large to be of use, possibly
because this software involves the simultaneous inference of phylogeny and phylogeography and is
overparametrized. They suggest that other Bayesian methods, in particular BayesTraits, are more efficient.
Second, “migrations” (Neureiter et al. 2021) or “early jumps” (Wichmann and Rama 2020), where one place is
abruptly left for another are a major source of error. Bayesian methods are thought to produce more reliable

results in the case of “expansions”, whereby speech communities gradually expand their territory. Finally,
Bayesian random walks cannot easily deal with directional trends emerging from geographical or ecological
constraints, such as oceans, mountains and deserts (Wichmann and Rama 2020, Neureiter et al. 2021).

The identification of a homeland through the diversity hotspot principle depends on the location of the
deepest subgroups and therefore, on how robustly the internal structure of a given family has been established.
Uncertainty with regard to the primary branching in the family, will does lead to uncertainty in the
reconstruction of the homeland. Besides, the contemporary hotspot of linguistic diversity may diverge from an
earlier one, for instance, due to the loss of diversity in the homeland due to later spreads.

Finally, cultural reconstruction relies on the assumptions that meanings of words can be accurately
reconstructed to a proto-language, but is a fact that semantic reconstruction is less precise than phonological
reconstruction and leaves room for subjectivity and interpretation. In addition, it is not sufficient to reconstruct a
single cultural or natural item to an ancestral language, but our reconstruction should be backed up by other
members of the semantic domain to which it belongs. It may therefore be hard to come by a sufficient number of
proto-words that are truly geographically diagnostic.

*10.2. Overcoming the limitations specific to the Transeurasian case*

Given these limitations and considering cases in which one method is not foolproof or even not applicable, the
three methods serve to complement each other.

In our Bayesian phylogeographic approach, we use BEAST v2.6 with adaptive coupled MCMC. The best fit
is provided by the pseudo Dollo covarion model with a relaxed clock. As this model is not available in
BayesTraits and because BEAST is specialised in time trees, we opted for these settings and software. Our
Bayesian phylogeographic analysis uses the posterior tree set from our lexical analysis. We assigned point
positions to the tips and randomly sampled trees from the posterior while estimating geographical parameters
through MCMC. The resulting maps with circles corresponding to the 95% highest posterior density intervals
are shown in SI 3. In line with the reservations expressed by Wichmann and Rama 2020, these intervals, are
indeed too large to be informative. We solve this issue by representing the dots with the highest posterior
probability estimated by the Bayesian phylogeography in red on the maps above and by integrating these dots
with the locations suggested by diversity hotspot and cultural reconstruction. The resulting map in Fig. SI 4.10
with the linguistic homelands proposed for the root and the nodes in the Transeurasian family, therefore gives a
realistic approximation of where the homelands were situated.

The population movements in the Transeurasian case are not to be understood as a permanent movement of
an entire population, involving a whole language family leaving its original homeland. Rather, growing
populations requiring more resources cause certain subgroups to leave the original homeland and to colonize
new territory. This causes inhabitants of the new territory abandon their local language and adopt the incoming
language of the colonizers. Similar to most cases of farming/language dispersal, this scenario is in line with the
expansion model rather than with the migration or “early jumps” model as defined by Neureiter et al. (2021) and
Wichmann and Rama (2020). Therefore, we can expect a reliable performance of Bayesian homeland detection
methods in the Transeurasian case.

For a few nodes in our tree the predicted issue with Bayesian random walks not being able to respond
properly to geographical or ecological constraints indeed appears to be the case, such as the situation of Proto-
Altaic in the Northeastern Altai Mountains or Proto-Japonic in the Kagoshima Bay immediately south of
Kyushu. However, these are only minor aberrations that are easily wiped out by integrating the Bayesian results
with those of the hotspot diversity principle and cultural reconstruction. The situation of the Proto-Transeurasian
homeland in the uncultivable sand dunes of the Gobi desert may seem problematic in view the connection with
agriculture proposed in our research. However, Yang et al. (2015) demonstrated that this area was used as
farmland before it became a desert ca. 4200 years ago through groundwater capture by the Xilamulun River.

There are several nodes in our tree, for which the application of the diversity hotspot principle would be
misleading if we considered only contemporary linguistic diversity. This is the case for Proto-Japano-Koreanic,
Proto-Koreanic, Proto-Mongolic and Proto-Turkic because we know from historical sources that earlier
linguistic diversity has been erased by dominant linguistic expansions by the end of the first and the beginning
of the second millennium AD, such as the Silla unification in Korea, the Mongolic unification under Jinhgis
Khan and the expansion of Common Turkic over the Eurasian Steppe. However, taking indications of earlier
diversity into account, the results of the diversity hotspot principle reach a better convergence with the other
homeland detection methods used in this study.

Conversely, when the data for cultural reconstruction is limited or the archaeological record is slim, then that
method may be less reliable and the other methods can step in. In this way, we gain by the interaction of the
three homeland detection methods.

It could be argued that the different methods are not completely independent from each other: The diversity
hotspot method depends on tree structure, so in that sense there is a certain overlap with Bayesian
phylogeography. Phylogeography also relies on the logic of bottom-up reconstruction, which overlaps

somewhat with cultural reconstruction using proto-forms. Nevertheless, the three detection methods applied in
 our study are sufficiently independent of each other as they do not only make use of different principles of
 analysis but also rely on different lines of evidence. As shown in the tables in each section above, the Bayesian
 approach is based on a dataset of basic vocabulary, whereas the diversity hotspot principle starts from
 geographic observations about the distribution of the languages involved and cultural reconstruction relies on
 the reconstruction of natural and agropastoral vocabulary. As such, the homeland detection applied here
 advances convergent evidence and not just convergent methods.

Fig. SI 4.10 infers a location for the homelands of the root and nodes of the Transeurasian family based on
 the integration of the three homeland detection methods applied in our research.”

 We included reference to the two sources (Neureiter et al. 2021, Wichmann and Rama 2020)
 and the SI 4 in the methods section of the manuscript:

“Since Bayesian phylogeography must contend with a number of limitations (Neureiter et al. 2021, Wichmann
 and Rama 2020,) we complemented it with other homeland detection methods such as linguistic palaeontology
 and diversity hotspot principle in order to reach a balanced location for the homelands of the root and nodes of
 the Transeurasian family (SI 4).”

 (27) Reviewer 5: (6) Note that there [is] no strong linguistic evidence supporting ‘Farming
 Hypothesis’. In fact, the manuscript does not provide strong linguistic evidence to completely
 reject the alternative hypothesis of pastoralist although they use the traditional historical
 linguistic comparative method to reconstruct proto-type of the words associated with
 cultivation, food production and preservation, millet and rice, and domesticated animals such
 as dog. The homology of these words can probably be generated by the lexical borrowings in
 the ancient period.

We added a section to SI 5 (Inherited and borrowed correspondence sets for agropastoral
 vocabulary across the Transeurasian languages. 20_Eurasia3angle_synthesis_SI
 5_subsistence.docx), line 2817- 2860, that summarizes the arguments for the complete
 rejection of the alternative pastoralist hypothesis. On Line 146 of the manuscript we added
 reference to this (SI 5).

 **“4. Rejection of the Pastoralist Hypothesis**

 Our research challenges the traditional ‘Pastoralist Hypothesis’ that identifies the primary dispersals of the
 Transeurasian languages with nomadic expansions starting in the eastern Steppe near the Altai mountains in the
 fourth millennium BP (Menges 1977, Miller 1990, Dybo 2013) by proposing a ‘Farming hypothesis’. In order
 to determine which hypothesis is most suited, we tested different predictions with regard to the homeland, age,
 inherited vocabulary and contact profile, summarized in Table SI 5.1 below.

Table SI 5.1 Testing the validity of the Pastoralist Hypothesis as compared to the Farming Hypothesis on the
 basis of spatiotemporal and comparative linguistic predictions.

	Pastoralist Hypothesis predicts	Farming Hypothesis predicts	Our research shows
Homeland	Grasslands: Eastern steppe near the Altai mountains	Center of domestication: West Liao River/ Yellow River	West Liao River (SI 4)
Age ceiling	5 000 BP	9 000 BP	9181 BP 5595-12793 95% HPD (Extended data Fig. 1)
Inherited vocabulary	Pastoral	Agricultural	Agricultural /No pastoral (SI 5)
Contact profile	Agricultural terms are borrowed or inherited	Pastoral terms are borrowed	Pastoral terms are borrowed Agricultural terms are borrowed or inherited

 The Pastoralist Hypothesis predicts a homeland in the grasslands of the Eastern steppes near the Altai
 mountains, a location that originally lead to the coining of the label "Altaic" in reference to the Transeurasian
 languages. By contrast, the "Farming Hypothesis" predicts a homeland at a nuclear center where plant
 domestication emerged, such as the West Liao River or Yellow River areas in Northeast Asia.

According to Taylor et al. (2020), evidence for livestock-based herding subsistence on the Eastern Steppe is
 only found from the late 5th and early 4th millennia BP onwards, while horse-based pastoralism is as late as
 3200 BP. If Proto-Transeurasian indeed was the ancestral language of pastoralists on the steppe, 5000 BP is a
 ceiling for its existence. Since agriculture is thought to have emerged from around the 9th millennium BP
 onwards in the domestication centres in Northeast Asia, 9000 BP serves as a ceiling for the Farming Hypothesis.

As far as inherited vocabulary is concerned, the ancestral language of pastoralists is expected to contain
 pastoral vocabulary relating, for instance, to domesticated ruminants, herding, dairying or horse exploitation. By
 contrast, ancient farmers are expected to be familiar with terms for agriculture, such as those relating to plant
 cultivation, agricultural tools or domesticated crops.

As agriculture preceded pastoralism in Northeast Asia, all pastoral terms are expected to represent later
 innovations through language-internal change or borrowing under the Farming Hypothesis, whereas a Pastoral
 Hypothesis leaves room for agricultural terms to be inherited or borrowed.

In reality, different homeland detection methods such as Bayesian phylogeography, diversity hotspot
 principle and cultural reconstruction converge on the location of the Transeurasian homeland in the West Liao
 River area, as proposed by the Farming Hypothesis. Even taking into account the large credible interval (5595 -
 12793 95%HPD), the dating of the first break-up in the Transeurasian family excludes the Pastoral Hypothesis
 since the interval predates the development of nomadic pastoralism on the Eastern Steppe.

Linguistic criteria applied above attribute all similarities between pastoral words across the Transeurasian
 languages to borrowing, while some of the similarities in the agricultural vocabulary are residue of inheritance.
 Therefore, it is safe to exclude the Pastoralist Hypothesis as an explanation of the linguistic homology.”

 (28) Reviewer 5: The samples in language phylogeny cover the large geographic areas across
 the Europe and Asia. In contrast, the archaeological analyses in this paper make concerns on
 the area of Northern China. In my intuition, the samples in Northern China cannot represent
 the whole Transeurasian languages or populations even under the Pastoralist Hypothesis.

 Our archaeological data and analyses also include the Russian Far East, Korea and Japan, and
 are not limited to Northern China. In our study, what is key for further triangulation is the
 linguistic distribution of the languages in the Neolithic and Bronze Age that are associated
 with the eastward spread of farming and not that of the contemporary languages as a whole.

 We made this clear in the caption of Figure 2: “The distribution of archaeological sites in Fig. 2 is
 smaller than that of contemporary languages in Fig. 1 because we focus on the early dispersal of the linguistic
 subgroups in the Neolithic and Bronze Age and on the links between the eastward spread of farming and
 language dispersal”

 (29) Reviewer 5: From [an] archaeological perspective, the manuscript pos[i]ts the
 archaeological data of Neolithic and Bronze Age [sites] in northern China compiled from the
 several literature and documents of, and extracted and coded [the] these sites. This collection
 and quantitative work is very essential for the “triangulation” study. However, similar to the
 aspect of linguistic analysis, the manuscript did not provide more detailed descriptions about
 the data collection and criteria of feature selection.

We thank the reviewer for pointing out that further information on the selection of the
archaeological features is appropriate for an interdisciplinary paper such as this. An extended
discussion to this effect has been added to SI 7 (Qualitative analysis of the archaeological
database 27_Eurasia3angle_synthesis_SI 7_qualitative analysis) under §1 Methods.

The text added is as follows:

“For the database of archaeological features (SI 6), we scored a wide range of artefact categories
available across the Northeast Asia region. The features scored comprised those considered likely to
be connected to agricultural dispersals in the region. At the same time, given different histories of
archaeology and preservation contexts across Northeast Asia, scoring was designed to be as broad and
inclusive as possible. In archaeology, ceramics are generally regarded as one possible indicator of
population movement. In certain areas of Northeast Asia, previous research has examined links
between ceramic typologies and migration in the Neolithic and Bronze Age but no comprehensive
regional analysis has yet been attempted. Pottery is also a key functional aspect of the transition from
hunter-gatherer to farming in Northeast Asia as new ceramic forms including tripods and steamers
were adopted with agriculture. Ceramics were not scored on established typologies but by using basic
decorative techniques and vessel shapes as a way to facilitate inter-regional comparisons. For
example, steamers were scored as one feature even though as many as six different types have been
identified in Chinese Neolithic sites.

Stone tools are linked to farming in Northeast Asia through a distinctive series of tools
including harvesting knives and sickles. Though less diagnostic, certain types of stone axes and hoes
are also thought to be related to cultivation. The stone tool category scored here includes jade and
obsidian, raw materials both known to be widely distributed in Northeast Asia.

The category of buildings includes common domestic architecture in prehistoric Northeast
Asia such as pit and surface dwellings, post holes and hearths. Public architecture, fortifications and
ditches were also scored, reflecting the growing complexity of Neolithic and Bronze Age settlements
in the region.

Twelve plants and 14 animals were scored to obtain an overall picture of the subsistence
economy of the scored sites. These included 10 wild species and 16 thought to have been
domesticated.

Burials are a way to understand changing social and ideological contexts as well as cultural
interactions during the Neolithic and Bronze Age in Northeast Asia. The interpretation of mortuary
customs can be complex, but here we scored the main types of burial architecture from Northeast Asia
during these periods, with the exception of adult burial jars which are not found outside western
Japan. One burial category, dolmens, has been much discussed in terms of possible links to
migrations, though controversies remain”

(30) Reviewer 5: Based on the binary-coding archaeological features, the study reconstructed
the Bayesian phylogeny via BEAST. I do not suspect the efficiency of the program BEAST,
but MrBayes could be more suitable for reconstructing cultural evolution of archaeological
sites.

BEAST has many users and the various versions obtained tens of thousands of citations. It is
specialised in time trees, unlike MrBayes, so we do not think there is reason to suspect
validity of the results. We demonstrated that the pseudo-Dollo covarion model is the best
fitting model used for the evolution of binary characters compared to other models commonly

used for inference in cultural evolution, so that model is used for the final analysis. This
model is not available in MrBayes, Bayes Traits or other packages we are aware of, but only
available in BEAST.

(31) Reviewer 5: I am confused that in the main body of manuscript, authors used 23
individuals (perhaps ancient people? Or contemporary persons?) in line 193 but there is a
lack of detail of sample collection procedure for these individuals and their authentication.

We thank the reviewer for pointing this out. We added ‘ancient’ on line 193 of the
manuscript text. Given the space restrictions in the main text, we provided a detailed
description of the sample collection procedure for these 23 ancient individuals
in Supplementary Information 13. As the reviewer is correct in pointing out that it was not
adequately referenced, we included a reference to S 13 on line 196 of the manuscript

(32) Reviewer 5: The manuscript attempt to map the language populations and ancient
populations (including ancient samples) from the genetic perspective, aiming to investigate
the association between the ancient populations and contemporary ones speaking specific
languages. It is very important to point out that such mapping cannot authenticate the
languages of ancient populations because the scenario of language shift frequently occurred
in the ancient periods. It is a very dangerous explanation based on this mapping.

Our findings do not exclude a scenario of language shift but rather embrace it. This is implied
by how we model the Framing/Language dispersals in our paper, for instance in SI 4 line
462-464: “The population movements in the Transeurasian case are not to be understood as a permanent
movement of an entire population, involving a whole language family leaving its original homeland. Rather,
growing populations requiring more resources cause certain subgroups to leave the original homeland and to
colonize new territory. This causes inhabitants of the new territory to abandon their local language and adopt the
incoming language of the colonizers.”

The subgroups moving eastward with agriculture, i.e. Koreanic, Tungusic and Japonic all
show traces of language shift. Ainu and Nivkh, languages spoken on the Asian North Pacific
Coast, systematically deviate from the structurally more homogeneous Transeurasian
languages. They have been regarded as "marginal" pockets of earlier structural types whose
lineages became isolated before the large-scale language spreads in Eurasia and that may
have links to the languages of northwestern America. There are indications of language shift,
in the structure of Koreanic, Tungusic and Japonic whereby some of the ancestral speakers
of Ainu and Nivkh abandoned their native language in favor of a Transeurasian target
language such as proto-Tungusic, proto-Koreanic and/or proto-Japonic. In other words, we
find indications of substratum interference in proto-Tungusic, proto-Koreanic and/or proto-
Japonic under influence of the ancestral states of Ainu and Nivkh. It concerns typological
features, such as initial velar nasal, absence of voicing distinction, initial consonant clusters,
2 distinctive tones, three laryngeal contrast sets for stops, strong verbal encoding of property
words, mixed strategies to mark distributive numerals, pro-drop, more than 2 demonstrative
distinctions, etc., which are more proto-typical of Ainu and Nivkh than of Transeurasian
languages. These typical features can often be explained as being implicational tendencies of
being of the head-marking type, with Ainu and Nivkh being typical head-marking languages,
while the Transeurasian languages are standard dependent-marking languages.

As this paper is already extremely loaded and rich in linguistic datasets, it would lead us to
far to add datasets demonstrating substratum interference from neighbouring languages into
the Transeurasian languages, supporting a scenario of language shift. This is in our view not

essential for the line of argumentation presented here, but it would without doubt make a
fruitful topic for publication in the future.

Reviewer Reports on the First Revision:

Referee #1 (Remarks to the Author):

- A. Apposite.
- B. Quite new and original idea, which could be supported by the parallel processes from the Near East.
- C. Extraordinary rich data, strict methodological approaches, presentation based on convincing arguments.
- D. Admirable application of various statistic methods, always with explanation, why.
- E. The conclusions are always exhaustively argued, frequently with discussion, why pro or contra.
- F. The updated version is satisfactory.
- G. Yes.
- H. With regard to the fact that the authors have summarized so many data, arguments and hypotheses, their material is relatively well-structured and orientation of a reader is quite good.

Referee #2 (Remarks to the Author):

I am satisfied with the answers from the authors to my comments in this revised version of the manuscript.

Referee #3 (Remarks to the Author):

This looks much improved. Concerns have been addressed and clarified in the rebuttal and main manuscript. I think this important paper is ready for publication.

I would not that SI #3 appeared to be missing from the list of files in the editorial manager-- obviously this needs to be sorted for publication. (So I could not reread this, but the indications from the rebuttal document indicate issues have been handled).

Referee #4 (Remarks to the Author):

This is the second time around for this paper, and the authors have taken aboard most of the suggestions I made in my first review a few months ago. But I still have a few issues of clarity:

1. The authors make a good case that the population at the base of the TE genetic tree had what they term an "Amur Valley" genetic profile, this presumably relating to the late Palaeolithic/pre-Neolithic populations of the general region. In lines 213-215 they state that this ancestry was present in the Baikal, Amur, Primorye, SE steppe and West Liao regions. This is a very large region for a single ancestry component. Does this need some comment? Furthermore, the suggestion in lines 213-214 should presumably be changed from "Amur-like ancestry thus likely represents the original genetic profile of Neolithic hunter-gatherers" to "Amur-like ancestry thus likely represents the original genetic profile of indigenous pre-Neolithic (or late Palaeolithic) hunter-gatherers".
2. Lines 156-7: "...and compiled an inventory of 157 early cereal remains with direct radiocarbon dates..." would read better as "...and compiled an inventory of 157 directly C14-dated early cereal remains..."
3. Lines 293-294 are a little confusing - the lower Amur Valley looks to me to be in the Primorye

region (at least from a map - I have never been there), so how can Amur ancestry be introduced to Primorye? Wasn't it there already, before the Neolithic?

4. Lines 304-305: on reading this, I am led to believe that the Upper Xiajiadian genetic component from Liaoning was involved in spreading Japonic languages to Korea and later from Korea to Japan. Is this a correct understanding? Perhaps it should be clarified. Is there material culture like Xiajiadian in Korea? See also the next point.

5. Lines 239-246 state:

"The lack of a significant Jomon component in Taejungni indicates that early populations without detectable Jomon ancestry linked to present-day Koreans, migrated to the Korean peninsula in association with rice farming, replacing Neolithic populations with some Jomon admixture although our genetic data currently do not have resolution to test this hypothesis due to limited sample size and coverage. We therefore associate the spread of farming to Korea with different waves of Amur and Yellow River gene-flow, modelled by Hongshan for the Neolithic introduction of millet farming and by Upper Xiajiadian for the Bronze Age addition of rice agriculture."

My understanding here is that the Hongshan movement went with early Koreanic languages, and the later Upper Xiajiadian one with early Japonic. Is this correct? If so, it raises the question of why Koreans do not speak Japonic languages now. Why did Japonic not replace the earlier Koreanic? I do not know the answer to this issue, but suggest the authors clarify what they really think.

6. Line 314 states: "By advancing the first evidence from ancient DNA" – surely the recent Wang et al. paper does this too, as do several others.

7. Finally, lines 318-319 state "While previous research on the Farming/Language Dispersal hypothesis regarded the Transeurasian zone as beyond the area of agriculture⁴¹". However, the reference given, First Farmers by Bellwood (published in 2005, and before the term "Transeurasian" was in common use), stated (p. 231) with respect to what was then termed "Altaic":

"Prior to this time, linguistic relationships are hard to reconstruct, but the ultimate roots of the whole family probably are to be found amongst the rich Neolithic cultures of Manchuria. Millet farmers with pottery and large villages, different in cultural tradition from the early farmers of the Yellow River, were well established on the fertile southern Manchurian plains by 6000 BC. Early agricultural dispersal for these pioneers, except into Korea, was probably circumscribed by decreasing rainfall in the west (Mongolia), decreasing temperature to the north (Siberia), and other farmers to the south (the early Sino-Tibetans)."

This statement does not suggest that the Transeurasian homeland was seen as being beyond the range of agriculture.

Referee #5 (Remarks to the Author):

I appreciate the authors' responses to the comments. However, some questions still remain for the computational approach, especially the Bayesian phylogenetic construction of languages.

For example, in the supplementary XML named tea254pdcov-ucln-fdn-constrained,

(1) Authors set the tip dates for several contemporary languages as 20. I am extremely confused that these languages are existing for use now. The settings of tip dates can traditionally be used for language extinction. Thus, if the languages are now alive, the dates for these should be set as 0 when you select "the dates are specified numerically as years before the present". Accordingly, I do not confirm the settings for tip dates correct.

(2) For the settings of priors including tree model and time calibration, authors choose the fossilized birth-death model because some languages in their samples are ancient. I am not clear

why they select the Weibull distributions but not the Normal distributions as the prior one for the time calibration of language divergence including Japonic, Kroeanic, Monogolian, Tungusic, and Turkic. Different prior distribution types for divergence time of languages could have potential impacts on the time estimation in the BEAST. A normal distribution is more widely used in phylolinguistics than Weibull distribution. According to the time calibration listed in the Supplementary Materials, it seems appropriate to choose the Normal distribution.

(3) The values of time calibration set in the BEAST are weird. Traditionally, the Mean value of the priors is set as the potential time of language divergence before the present (Calibration BP), but not the time point (Calibration (B)CE). For example, the Koreanic language divergence is referenced as 1150 CE +/- 175 (Calibration (B)CE), and equal to 800 BP +/- 175 (Calibration BP). Thus, the corresponding settings could be Normal distribution with mean = 8, and the sigma = 0.9. HOWEVER, the authors set the mean as 11.5, sigma = 1.75. The wrong settings are biased to the Bayesian phylogenetic estimation of divergence time.

In addition, there is no description for bar-plot in Figure 1b, such as 95% intervals for each bar, and especially what the basis of drawing this figure is (e.g. the BEAST results?)

Author Rebuttals to First Revision:

Documentation of second revision

13.07.2021

A Reply to the referees

Our replies are marked in blue, changes to the manuscript in red.

(1) Referee #3: I would note that **SI #3** appeared to be missing from the list of files in the editorial manager-- obviously this needs to be sorted for publication. (So I could not reread this, but the indications from the rebuttal document indicate issues have been handled).

SI 3. Bayesian phylogeographic analysis modelling the spatiotemporal expansion of the Transeurasian languages (18_Eurasia3angle_synthesis_SI 3_phylogeography.klm)

Because it requires applications, this file was uploaded to two external sources, i.e. GitHub (<https://github.com/rbouckaert/Eurasia3angle>) and FigShare <https://figshare.com/s/b9c67ca3ea47faf51d48>

(2) Referee #4: 1. The authors make a good case that the population at the base of the TE genetic tree had what they term an "Amur Valley" genetic profile, this presumably relating to the late Palaeolithic/pre-Neolithic populations of the general region. In lines 213-215 they state that this ancestry was present in the Baikal, Amur, Primorye, SE steppe and West Liao regions. This is a very large region for a single ancestry component. Does this need some comment?

This is an interesting remark. We are not aware about studies comparing the geographical extent of gene pools in the (pre-)Neolithic. The referee may be right that it is rather unusual to find such large regions for a single ancestry component. However, large gene pools are observed elsewhere, such as for instance the Yamnaya (Pontic Steppe) ancestry component in the Early Bronze Age, which covered a region from Western Europe to Xinjiang (Ning et al. 2019 *Current Biology*). A single Yamnaya component is even found in Afanasievo populations, situated in the Altai Mountains) populations, sharing an identical Yamnaya genetic profile in spite of separation over at least 2000 km.

The extent of regions covering a single ancestry component could be an interesting topic for future research, but considering the purpose for the present study, we did not insert a supplementary comment.

(3) Referee #4: Furthermore, the suggestion in lines 213-214 should presumably be changed from "Amur-like ancestry thus likely represents the original genetic profile of Neolithic hunter-gatherers" to "Amur-like ancestry thus likely represents the original genetic profile of indigenous pre-Neolithic (or late Palaeolithic) hunter-gatherers".

Indeed, the referee's suggestion is more accurate. On line 216-217 we specified accordingly: "Amur-like ancestry thus likely represents the original genetic profile of **indigenous pre-Neolithic (or late Palaeolithic) hunter-gatherers**"

(4) Referee #4: 2. Lines 156-7: "...and compiled an inventory of 157 early cereal remains with direct radiocarbon dates..." would read better as "...and compiled an inventory of 157 directly C14-dated early cereal remains..."

On line 159-160, we adapted the wording in line with referee's suggestion:
"and compiled an inventory of 157 directly C14-dated early cereal remains (SI 9)"

(5) Referee #4: Lines 293-294 are a little confusing - the lower Amur Valley looks to me to be in the Primorye region (at least from a map - I have never been there), so how can Amur ancestry be introduced to Primorye? Wasn't it there already, before the Neolithic?

The lower Amur is technically in Khabarovsk Krai, a bit further north than the Primorye. The term "Amur-like ancestry" is a genetic label. As the referee correctly points out, it covers a large gene pool that stretches from the Baikal to the Russian Far East and thus beyond the region that is geographically known as the Amur basin.

In order to avoid confusion, we rephrased lines 294-297 in the following way:

"Around the mid-6th millennium BP some of these farmers started to migrate eastwards, around the Yellow Sea into Korea and northeast into the Primorye, bringing Koreanic and Tungusic languages to these regions and bringing from the West Liao region additional Amur ancestries to the Primorye and mixed Amur-Yellow River ancestries to Korea ."

(6) Referee #4: 4. Lines 304-305: on reading this, I am led to believe that the Upper Xiajiadian genetic component from Liaoning was involved in spreading Japonic languages to Korea and later from Korea to Japan. Is this a correct understanding? Perhaps it should be clarified. Is there material culture like Xiajiadian in Korea? See also the next point.

In order to clarify the distinction between genetic modelling and sharing material culture, we made the following addition on the lines 307-310:

"Around 3300 BP farmers from the Liaodong-Shandong area migrated to the Korean peninsula, adding rice, barley and wheat to millet agriculture. This migration aligns with the genetic component modelled as Upper Xiajiadian in our Bronze Age sample from Korea and is reflected in early borrowings between Japonic and Koreanic languages. Archaeologically it can be associated with agriculture in the larger Liaodong-Shandong area without being specifically restricted to Upper Xiajiadian material culture."

(7) Referee #4: 5. Lines 239-246 state:

"The lack of a significant Jomon component in Taejungni indicates that early populations without detectable Jomon ancestry linked to present-day Koreans, migrated to the Korean peninsula in association with rice farming, replacing Neolithic populations with some Jomon admixture although our genetic data currently do not have resolution to test this hypothesis due to limited sample size and coverage. We therefore associate the spread of farming to Korea with different waves of Amur and Yellow River gene-flow, modelled by Hongshan for the Neolithic introduction of millet farming and by Upper Xiajiadian for the Bronze Age addition of rice agriculture."

My understanding here is that the Hongshan movement went with early Koreanic languages, and the later Upper Xiajiadian one with early Japonic. Is this correct? If so, it raises the question of why Koreans do not speak Japonic languages now. Why did Japonic not replace the earlier Koreanic? I do not know the answer to this issue, but suggest the authors clarify what they really think.

Yes, the referee's understanding is correct. We really think that this is a thought-provoking outcome of Farming/Language Dispersal and raised the issue in the supplementary information SI 5 (agropastoral). On the lines 198-211, we added the paragraph indicated in red, answering the referee's question:

“Why did the Neolithic expansion of millet farming cause pre-existing hunter-gatherers to abandon their native language and shift to the incoming Transeurasian target language, thus ensuring its continuity, while Bronze Age interaction involved maintenance of the native language with extensive borrowing of agropastoral vocabulary? In line with Renfrew’s (1987: 123–131) demography/subsistence model, the socio-economical context of the interaction may explain the outcome of the linguistic encounters. When farming is introduced to populations with relatively less successful subsistence strategies, we expect them to shift to the incoming culture and language because such a shift would involve a revolutionary potential for prosperity and demographic growth. By contrast, when groups of more or less equal socio-economic status meet, whereby certain crops are added to a pre-existing agricultural package or farming is complemented by pastoralism, the need to radically shift language and culture is less urgent. In such cases, the encounters are expected to result in borrowing.

The case of the spread of Koreanic and Japonic languages across the Korean peninsula represents a thought-provoking example of how (agri)culture may determine the linguistic outcome of interaction between speech communities. If millet agriculture and Hongshan genome were spread to the Peninsula with Proto-Koreanic languages in the Neolithic, while secondary crop agriculture and Upper Xiajiadian genome were spread with Proto-Japonic in the Bronze Age, it raises the question of why Koreans do not speak a Japonic language now. Why did Proto-Japonic not replace Proto-Koreanic on the Korean Peninsula? The answer to this question is probably determined by the socio-economical context of the interaction (Hudson & Robbeets 2020). When millet agriculture was introduced to the Korean Peninsula, farmers met hunter-gatherers. By contrast, when secondary crop agriculture was introduced, (rice, barley, wheat) farmers met (millet) farmers. From a socio-economical perspective, the first movement is more revolutionary than the second and will lead to language shift, while the second is expected to lead to linguistic borrowing between the pre-existing farmers and the incoming ones.

This explains why the Neolithic and Bronze Age migrations involved different linguistic dynamics and led to an interplay of two different outcomes, continuity and borrowing. This prehistorical layering of borrowed upon inherited words makes it difficult for historical linguists to distinguish between both transmission modes and is therefore at the base of the Transeurasian controversy.”

We added the following reference

Hudson, M. J. & Robbeets, M. Archaeolinguistic evidence for the farming/language dispersal of Koreanic. *Evolutionary Human Sciences* 2, e52 doi:10.1017/ehs.2020.49 (2020).

(8) Referee #4: 6. Line 314 states: "By advancing the first evidence from ancient DNA" – surely the recent Wang et al. paper does this too, as do several others.

On line 319, we replaced ‘the first’ by ‘new’ in the following way:

“By advancing **new** evidence from ancient DNA, our research thus confirms recent findings that Japanese and Korean populations have West Liao River ancestry”

(9) Referee #4: 7. Finally, lines 318-319 state "While previous research on the Farming/Language Dispersal hypothesis regarded the Transeurasian zone as beyond the area of agriculture⁴¹". However, the reference given, *First Farmers* by Bellwood (published in 2005, and before the term "Transeurasian" was in common use), stated (p. 231) with respect to what was then termed "Altaic":

"Prior to this time, linguistic relationships are hard to reconstruct, but the ultimate roots of the whole family probably are to be found amongst the rich Neolithic cultures of Manchuria. Millet farmers with pottery and large villages, different in cultural tradition from the early farmers of the Yellow River, were well established on the fertile southern Manchurian plains by 6000 BC. Early agricultural dispersal for these pioneers, except into Korea, was probably circumscribed by decreasing rainfall in the west (Mongolia), decreasing temperature to the north (Siberia), and other farmers to the south (the early Sino-Tibetans)."

This statement does not suggest that the Transeurasian homeland was seen as being beyond the range of agriculture.

The referee is right. On line 322-324 we rephrased as follows

“While some previous research regarded the Transeurasian zone as beyond the area suitable for farming [41 Heggarty & Beresford-Jones], our research confirms that the Farming/Language Dispersal hypothesis remains an important model for understanding Eurasian population dispersals [42 Bellwood].”

(10) Referee #5: Some questions still remain for the computational approach, especially the Bayesian phylogenetic construction of languages. For example, in the supplementary XML named tea254pdcov-ucln-fdn-constrained, (1) Authors set the tip dates for several contemporary languages as 20. I am extremely confused that these languages are existing for use now. The settings of tip dates can traditionally be used for language extinction. Thus, if the languages are now alive, the dates for these should be set as 0 when you select “the dates are specified numerically as years before the present”. Accordingly, I do not confirm the settings for tip dates correct.

Reading the referee’s feedback, we realize that our date setting in the analysis results in confusion.

The time units used in our analysis are indicated in centuries, so 1 time unit = 100 years. In the language analyses, time is going forward (as indicated by the attribute traitname="date-forward" in the XML), so a tip date set at 0 represents the year 0 AD, and 20 represents 2000 AD. Only if time was going backward contemporary languages should be assigned a tip date of 0.

However, the time depths presented in our results, such as the dates appearing in the node labels presented in Extended Data Fig. 1, are marked in centuries BP.

To avoid confusion, we added a comment to the read-me file saying:

"Note that time is in units of centuries in the language analyses and is going forward, so a tip date of 20 is interpreted as a date at the year 2000 AD, a calibration with mean -1.5 indicates the calibration has a mean at the year 150 BC."

For the read-me file, see:

https://raw.githubusercontent.com/rbouckaert/Eurasia3angle/master/39_Eurasia3angle_synthesis_SI%2019_XML%20files_README

(11) Referee #5: (2) For the settings of priors including tree model and time calibration, authors choose the fossilized birth-death model because some languages in their samples are ancient. I am not clear why they select the Weibull distributions but not the Normal distributions as the prior one for the time calibration of language divergence including Japonic, Kroeanic, Monogolian, Tungusic, and Turkic. Different prior distribution types for divergence time of languages could have potential impacts on the time estimation in the BEAST. A normal distribution is more widely used in phylolinguistics than Weibull distribution. According to the time calibration listed in the Supplementary Materials, it seems appropriate to choose the Normal distribution.

We are not sure where the Weibull distribution is mentioned for calibrations, but in the Supplementary Material describing the calibrations (38_Eurasia3angle_synthesis_SI_18_Bayesian linguistics.docx) there is no mention of a Weibull distribution for calibrations, and in fact we used a normal distribution for calibration as suggested by the reviewer. The

XML files are the ones used in the analyses and these are the authoritative sources, and these contain normal distribution.

(12) Referee #5: (3) The values of time calibration set in the BEAST are weird. Traditionally, the Mean value of the priors is set as the potential time of language divergence before the present (Calibration BP), but not the time point (Calibration (B)CE). For example, the Koreanic language divergence is referenced as 1150 CE +/- 175 (Calibration (B)CE), and equal to 800 BP +/- 175 (Calibration BP). Thus, the corresponding settings could be Normal distribution with mean = 8, and the sigma = 0.9. HOWEVER, the authors set the mean as 11.5, sigma = 1.75. The wrong settings are biased to the Bayesian phylogenetic estimation of divergence time.

We apologize for causing confusion by marking our calibrations in BC/AD as well as in BP in Table SI 18.1 . We based our analysis in the XML files on the BC/AD settings and put contemporary languages at 2000 AD. In order to avoid confusion, we now deleted the BP calibrations in Table SI 18.1 and in the explanatory text. For reason of consistency with other datasets, we now use the BC/AD convention in the table and the text in SI 18.

As explained in the response to the reviewer's question in (10), note that time is encoded to be going forward, and is encoded in centuries. So, the normal calibration with mean=11.5, and sigma=1.75 for Koreanic does properly represent the desired 1150 AD +/- 175 calibration.

(13) Referee #5: In addition, there is no description for bar-plot in Figure 1b, such as 95% intervals for each bar, and especially what the basis of drawing this figure is (e.g. the BEAST results?)

In line with the format requirements for figures in the main text, we reduced Figure 1 in such a way that it no longer includes the bar-plot. We added a reference to the BEAST results in Extended data Fig. 1 for the estimation of the time depth of the ancestral languages involved in the caption of Figure 1, as follows:

Fig. 1a. Geographical distribution of the 98 Transeurasian language varieties included in this study. Contemporary languages are represented by coloured surfaces, historical varieties by red dots. Fig. 1b. Reconstructed locations of Transeurasian ancestral languages spoken during the Neolithic (red) and Bronze Age and later (green). For detailed homeland detection, see SI 4. **The estimated time depth is based on Bayesian inference presented in Extended data Fig. 1.**

We maintained the figure with the bar-plot in Supplementary Information 4. Here we added the 95% intervals for each bar in the caption:

“Fig. SI 4.10 Reconstructed Transeurasian linguistic homelands, distinguishing between ancestral languages spoken during Neolithic times (red) and those spoken in the Bronze Age and later (green). **The estimated time depth represented in the bar plot is based on Bayesian inference presented in Extended data Fig. 1. The credible intervals for the break-up time of the ancestral languages represented in the bar plot are following: Proto-Transeurasian 9181 BP (5595 -12793 95%HPD); Proto-Japano-Koreanic 5458 BP (3335-8024 95%HPD); Proto-Altaic 6811 BP (4404-10166 95%HPD); Proto-Mongolo-Tungusic 4491 BP (2599-6373 95%HPD); Proto-Turkic 2195 BP (1882-2493 95%HPD); Proto-Mongolic 939 BP (871-1011 95%HPD); Proto-Tungusic 1950 BP (1499-2412 95%HPD); Proto-Koreanic 975 BP (528-1560 95%HPD) and; Proto-Japonic 2136 BP (1499-2412 95%HPD).”**

B Editorial format requirements

We performed an editorial check and made some changes to our paper so that it complies with your format requirements.

We updated Reporting Summary and Editorial Policy documents to reflect the revisions made and submitted with the revised manuscript.

We double-checked space requirements: summary and body text measured without captions and references is 2911 words (< 3000 max); title contains 73 characters (< 75 max) including spaces and does not contain punctuation; summary paragraph is 193 words (< 200 max) and complies with the requested structure; introduction has 299 words (< 500 max); subheadings are 12, 11, 9, 25 characters respectively (< 40 max); main text has 42 references (< 50 max). Figure legends contain 76, 107, 36 and 13 words respectively (< 300 max)

Figure captions are revised to begin with a brief title for the whole figure and include sample sizes

Figure legend is listed after the main text references

Method reference numbering continues from 42, the last reference in the main text

We added an information statement: “Correspondence and requests for materials should be addressed to Martine Robbeets.”

Some figures are still in JPEG format, but as our graphic designer is in vacation, we plan to convert to pdf/ai format when she returns. She will also adjust the lettering in all figures to a sans-serif font (Helvetica).

We added a SI Guide

We do not understand what is exactly meant with “ SI should not typically contain data figures”. In our linguistic SI, we have some maps situating homelands in space, in the archaeological SI, we have a few fieldwork pictures for illustration and in our genetic SI there are some plots in support of the argumentation. Are we allowed to leave them in? They are necessary for the sake of readability and transparency.

Reviewer Reports on the Second Revision:

Referee #4 (Remarks to the Author):

The authors have addressed my previous comments satisfactorily. I am happy to see this paper go ahead to publication.

Referee #5 (Remarks to the Author):

The authors take a lot of efforts to improve the manuscript. However, there remain the questions as followed:

The settings of time calibration in this paper are still questionable. There is a important tutorial of language phylogenies on the website (URL: <https://taming-the-beast.org/tutorials/LanguagePhylogenies/>). In the tutorial, the time calibrations are set as followed:

"...Next we want to add some priors for time calibration. According to (Wilmshurst, Hunt, Lipo, & Anderson, 2010), the settlement of New Zealand can be securely dated to between 1230-1282 A.D. We have two dialects of New Zealand Maori in these data: Maori and SouthIslandMaori. Let's operationalize this calibration like this: If we assume that the present is a nice round number like the year 2000 (this makes interpretation easier), then we convert this to before present:

2000 - 1230 = 770 years ago

2000 - 1282 = 718 years ago

So a prior incorporating this information could be a tight log-normal distribution with mean 744 (in real space) and a variance 0.02....

Another good calibration is East Polynesian: It's a well-attested linguistic group, and we have good archaeological evidence for when the initial settlement of East Polynesia began. However the ages are a bit controversial between "short" and "long" chronologies e.g.:

1025 - 1121 AD = 975 - 879 years ago

800 - 1000 AD = 1200 - 1000 years ago

The average of these estimates is about 1000 years ago, and they are spread on both sides by about 150- 200 years. This makes a great candidate for a Normal distribution with mean 1000 and variance 100. ..."

These settings are definitely different from those in this paper and the descriptions (12) in the response letter.

The author question is surrounding the linguistic classification of the Transeurasian languages. In this paper, the Transeurasian languages involve Japanese, Korean, Tungusic, Mongolic and Turkic. From the linguistic perspective, the first two languages are isolated, the others belong to Altaic language family. It is a unsolved problem whether the Altaic language is a great language family or a language alliance. However, the phyolinguistic analyses and related discussions on the

Transeurasian languages are here on the basis of the prerequisite that the isolated languages (Japanese, Korean) and Altaic languages (Tungusic, Mongolic and Turkic) constitute a tree-like typology. It is extremely conflicting with the existing linguistic knowledge, and is also lack of sufficient linguistic evidence except for the first author's statement.

Author Rebuttals to Second Revision:

Referee 5 (1): "The settings of time calibration in this paper are still questionable. There is a important tutorial of language phylogenies on the website (URL: <https://taming-the-beast.org/tutorials/LanguagePhylogenies/>). In the tutorial, the time calibrations are set as followed:

"...Next we want to add some priors for time calibration. According to (Wilmshurst, Hunt, Lipo, & Anderson, 2010), the settlement of New Zealand can be securely dated to between 1230-1282 A.D. We have two dialects of New Zealand Maori in these data: Maori and SouthIslandMaori. Let's operationalize this calibration like this: If we assume that the present is a nice round number like the year 2000 (this makes interpretation easier), then we convert this to before present:

2000 - 1230 = 770 years ago

2000 - 1282 = 718 years ago

So a prior incorporating this information could be a tight log-normal distribution with mean 744 (in real space) and a variance 0.02....

Another good calibration is East Polynesian: It's a well-attested linguistic group, and we have good archaeological evidence for when the initial settlement of East Polynesia began. However the ages are a bit controversial between "short" and "long" chronologies e.g.:

1025 - 1121 AD = 975 - 879 years ago

800 - 1000 AD = 1200 - 1000 years ago

The average of these estimates is about 1000 years ago, and they are spread on both sides by about 150- 200 years. This makes a great candidate for a Normal distribution with mean 1000 and variance 100. ..."

These settings are definitely different from those in this paper and the descriptions (12) in the response letter."

We can assure the reviewer that the calibrations have been encoded correctly in the XML.

To reiterate, there are two ways to encode time in BEAST: forward and backward. Which encoding is used depends on the "traitname" tag. For the language analyses, it is set to "date-forward", which means tip-dates and calibrations are interpreted as going forward, and a calibration with mean X is interpreted by BEAST as X centuries AD (because the unit of time is in centuries for the language analysis).

The tutorial uses "date-backward", which means tip-dates should be interpreted as ages. Just because the tutorial concentrates on a "date-backward" encoding does not mean that the encoding in our analysis is incorrect. In fact, we checked the calibrations encoded in the XML once again, and confirm these are indeed as intended and listed in Table SI 18.

Referee 5 (2): “The author question is surrounding the linguistic classification of the Transeurasian languages. In this paper, the Transeurasian languages involve Japanese, Korean, Tungusic, Mongolic and Turkic. From the linguistic perspective, the first two languages are isolated, the others belong to Altaic language family. It is a unsolved problem whether the Altaic language is a great language family or a language alliance. However, the phyolinguistic analyses and related discussions on the Transeurasian languages are here on the basis of the **prerequisite** that the isolated languages (Japanese, Korean) and Altaic languages (Tungusic, Mongolic and Turkic) constitute a **tree-like typology**. It is **extremely conflicting with the existing linguistic knowledge**, and is also **lack of sufficient linguistic evidence except for the first author’s statement.**”

This reservation, held back by the referee until now, brings a preconceived idea to the discussion, that is often encountered on popular sites like Wikipedia. Indeed, as the reviewer correctly points out, the question of the common origin and dispersal of Japanese, Korean, Tungusic, Mongolic and Turkic — the “Transeurasian” languages — is among the most hotly debated issues of historical linguistics and population history. That is why the current results of our interdisciplinary paper are of great scientific significance.

However, the referee’s impression that the genealogical relatedness of the Transeurasian languages is “*extremely conflicting with the existing linguistic knowledge*” is incorrect.

Gustaf J. Ramstedt is usually considered the founder of a geneological Transeurasian linguistics because he established a modern linguistic framework for Transeurasian comparison, supported by regular sound correspondences (1957) and morphological cognates (1952). While, until the late sixties, the field focused on the comparison of Turkic, Mongolic, and Tungusic on the one side, e.g. Poppe 1960, 1965, 1975, and of Korean and Japanese on the other, e.g. Martin 1966, in the seventies, Miller’s (1971) monograph ‘Japanese and the other Altaic languages’ increased the scholarly interest in the overall comparison of these languages. Clauson (1956) and Doerfer (1963–1975, 1966) raised substantial criticism against the geneological relatedness of these languages, which was mainly based on the alleged lack of shared basic vocabulary and the explanation of all correlations by borrowing.

Starostin et al. (2003) resurrected scholarly interest in the Transeurasian unity, accumulating a body of evidence that was far more impressive in quantity and rich in empirical material than the number and scope of etymologies proposed previously. However, these new matches were, in their turn, criticized for reason of phonological, morphological or semantic overpermissiveness, among others by Robbeets (2005), leaving room for a reduced core of reliable etymologies and by Vovin (2005), Georg (2007) and Fuente (2017), completely rejecting all evidence advanced so far.

Even if some of the objections are reasonable and a serious share of the evidence provided in the past is indeed questionable, there is nonetheless a core of reliable evidence for the classification of Transeurasian as a valid geneological grouping. As a result, the first author’s argumentation that the Transeurasian are geneologically related has recently gained acceptance in the field; see Gözaydin (2006); Rozycki (2006); Büyükmavi (2007, “proof for a close ...genetic relationship between Japanese and Altaic.); Kara (2007); Décsy (2007); Johanson (2010); Koch (2014); Ross (2014); Comrie (2014) “the most convincing demonstration to date of the unity of the Altaic family”), Dybo (2016); Eliasson (2016 “conceivable geneological links between the language groups in question”). These references may serve to attenuate the referee’s impression that the Transeurasian hypothesis is “*extremely*

conflicting with the existing linguistic knowledge”

Second, contrary to the referee’s impression, the “*tree-like typology*” is not a “*prerequisite*” but an outcome of our paper. Besides, the tree-like classification of Transeurasian is not a monopoly of the submitted paper, let apart of the first author alone. As illustrated in the figure below, numerous historical linguists have proposed a tree-like classification of Transeurasian in the past: A Polytopology (Vladimircov 1929: 44–47, Baskakov 1981: 14, Starostin et al. 2003: 236, Dybo et al. 2019), B Binary topology clustering Tungusic with Japonic (Miller 1971: 44, Blažek and Schwarz 2014: 90), C Binary topology with Tungusic separating first from Altaic (Ramstedt 1924: 440, Robbeets and Bouckaert 2018), and D Binary topology with Turkic separating first from Altaic (Poppe 1960b, 1965: 147, Street 1962: 95. Tekin 1994: 82, Robbeets 2015: 506 and confirmed by our Bayesian approach) The contribution of our paper is that our Bayesian approach solved the uncertainty in favor of figure D.

Finally, the referee's impression that there is a "lack of sufficient linguistic evidence" is insufficiently informed.

The Transeurasian hypothesis is an explanation that accounts for the sharing of certain linguistic properties between Turkic, Mongolic, Tungusic, Koreanic, and Japonic languages by inheritance from a common ancestral language called 'Proto-Transeurasian'. This hypothesis is empirically testable by applying the standard method of historical comparative linguistics. The scientific demonstration that two or more languages descend from a common ancestor consists first, in the establishment of a set of similarities holding between the languages concerned and second, in the demonstration that not all these similarities can be accounted for by an interplay of coincidence, universal principles in linguistic structuring and borrowing. Our supplementary information provides the basic evidence needed for testing the Transeurasian hypothesis: the similarities are established with the basic vocabulary etymologies and sound correspondences provided in SI 2, the ruling out of alternative accounts is done in SI 5. The evidence in the Supplementary Information alone warrants the conclusion that the Transeurasian languages are relatable within the limits of the classical historical comparative method.

However, apart from the efforts made in our article, more evidence has been provided by various scholars in the past. In line with the requirements of the classical comparative method of historical linguistics, the evidence consists not only of lexical etymologies including basic vocabulary and regular sound correspondences but also of shared morphology, among others, see for instance, Murayama (1957, 1958), Poppe (1972), Street (1978), Nasilov (1978) and Kormušin (1984), Miller (1981, 1982), Finch (1987), Itabashi (1991, 1993, 1996), Vovin (2001), and Robbeets (2015). We hope that this can serve to illustrate that the linguistic evidence for Transeurasian relatedness answers the requirements of the historical comparative linguistic method.

References mentioned in this reply

- Baskakov, Nikolaj A. (1981). *Altaiskaja sem'ja jazykov i ee izučenie* [The Altaic language family and its origins.] Moscow: Nauka.
- Blažek, Václav, and Michal Schwarz (2014). 'Jmenná deklinace v altajských jazycích [Nominal declension in Altaic]', *Linguistica Brunensia* 62.1: 89–98.
- Büyükmavi, Maud (2007). 'Rezension von Robbeets, Martine 2005. Is Japanese related to Korean, Tungusic, Mongolic and Turkic?', *Oriens Extremus* 46: 306–310.
- Clauson, Gerard (1956). 'The Case against the Altaic Theory', *Central Asiatic Journal* 2: 181–187.
- Décsy, Gyorgy. (2007). 'Review of Robbeets, Martine 2005. Is Japanese related to Korean, Tungusic, Mongolic and Turkic?', *Eurasian Studies Yearbook* 79: 157.
- Doerfer, G. (1966) 'Zur Verwandtschaft der altaischen Sprachen', *Indogermanische Forschungen* 71: 81-123.

- Doerfer, Gerhard (1963–1975). *Türkische und Mongolische Elemente im Neupersischen, unter besonderer Berücksichtigung älterer neupersischer Geschichtsquellen, vor allem der Mongolen- und Timuridenzeit*, vols. 1-4. [Turkic and Mongolic Elements in New Persian with Special Attention to Old New Persian Historical Sources, Mainly of the Period of the Mongols and the Timurids]. Wiesbaden: Franz Steiner
- Dybo, Anna (2013). 'Language and archeology: some methodological problems. 1. Indo-European and Altaic landscapes', *Journal of Language Relationship* 9: 69–92.
- Dybo, Anna (2016). New trends in European studies on the Altaic problem. *Journal of Language Relationship* 14: 41-106.
- Dybo, Anna, and George Starostin (2008). 'In Defense of the Comparative Method, or the End of the Vovin Controversy', *Aspects of Comparative Linguistics* 3: 119–258.
- Eliasson, Stig (2016). 'Review of Gardani, F., Arkadiev, P. And Amiridze, N. (eds) 2015. Borrowed morphology. Berlin: De Gruyter Mouton', *Studies in Language* 40(3): 722-731.
- Finch, Roger (1987). 'Verb Classes in the Altaic languages', *Sophia Linguistica* 26: 41–61.
- Fuente, José Andrés Alonso de la (2016), 'Review of Robbeets, Martine 2015. Diachrony of verb morphology: Japanese and the Transeurasian languages', *Diachronica* 33(4): 530-536.
- Georg, Stefan (2007). 'Review of Robbeets, Martine 2005. Is Japanese related to Korean, Tungusic, Mongolic and Turkic?', *Korean Studies*: 247–278.
- Gözyaydin, Nuretim (2006). 'Dergi ve kitap dünyasından', *Türk Dili* 653: 467–471.
- Itabashi, Yoshizo (1991). 'The origin of the Old Japanese genitive case suffixes *n/nö/na/Nga and the Old Korean genitive case suffix *i in comparison with Manchu-Tungus, Mongolian, and Old Turkic', *Central Asiatic Journal* 35: 231–278.
- Itabashi, Yoshizo (1993). 'A comparative study of the Old Japanese and Korean nominative case suffixes *i* with the Altaic third person singular pronouns', *Central Asiatic Journal* 37: 82–119.
- Itabashi, Yoshizo (1996). 'A comparative study of the Old Japanese locative case suffix *tu* with the Altaic locative and the related case suffixes', *Acta Orientalia*, 49: 373–394.
- Johanson, Lars (2010). 'The high and low spirits of Transeurasian language studies', in Johanson, Lars and Robbeets, Martine (eds), *Transeurasian verbal morphology in a comparative perspective: genealogy, contact, chance*. (Turcologica 78.) Wiesbaden: Harrassowitz
- Kara, Gyorgy (2007). 'Review of Robbeets, Martine 2005. Is Japanese related to Korean, Tungusic, Mongolic and Turkic?', *Anthropological Linguistics* 49: 95–98.
- Koch, Harold 2014. 'Morphological Borrowing and Genetic Relationship. A review article of Johanson and Robbeets (eds). 2012: 'Copies Versus Cognates in Bound Morphology' *Journal of Language Contact* 7: 408-424.
- Martin, Samuel Elmo (1966). 'Lexical evidence relating Korean to Japanese', *Language* 42: 185–251.
- Miller, Roy Andrew (1971). *Japanese and the other Altaic languages*. Chicago: The University of Chicago Press.
- Miller, Roy Andrew (1981). 'Altaic Origins of the Japanese Verb Classes', in Yoël Arbeitman (ed.), *Bono Homini Donum: Essays in Historical Linguistics in Memory of J. Alexander Kerns*. Amsterdam: Benjamins, 815–880.
- Miller, Roy Andrew (1982). 'Japanese Evidence for some Altaic Denominal Verb-stem Derivational Suffixes', *Acta Orientalia Academiae Scientiarum Hungaricae* 36: 391–403.

- Murayama, Shichiro (1957). 'Vergleichende Betrachtung der Kasus-Suffixe im Altjapanischen', in Omeljan Pritsak (ed.), *Studia Altaica. Festschrift für Nicholas Poppe zum 60. Geburtstag am 8 August 1956*. Wiesbaden: Otto Harrassowitz, 126–131.
- Murayama, Shichiro (1958). 'Einige Formen der Stammverkürzung in den altaischen Sprachen', *Oriens* 11: 225–230
- Nasilov, Dmitrij M. (1978). 'Formy vyraženija sposobov glagol'nogo deistvija v altaiskikh jazykakh (v svjazi s problemoy glagol'nogo vida) [Types of verbal action expressions in Altaic languages (with a reference to the problem of aspect)]', in Orest P. Sunik (ed.), *Očerki sravnitel'noi morfologii altaiskikh jazykov*. Leningrad, 77–177.
- Poppe, Nicholas (1960). *Vergleichende Grammatik der altäischen Sprachen* [Comparative Grammar of the Altaic Languages]. Teil I: Vergleichende Lautlehre. Wiesbaden: Otto Harrassowitz.
- Poppe, Nicholas (1965). *Introduction to Altaic Linguistics*. Wiesbaden: Harrassowitz.
- Poppe, Nicholas (1972). 'Über einige Verbalstammbildungssuffixe in den altaischen Sprachen', *Orientalia Suecana* 21: 119–141.
- Poppe, Nicholas (1975). 'Altaic linguistics: An overview', *Gengo no kagaku [Sciences of language]* 6: 130–186.
- Ramstedt, Gustaf J. (1924). 'A Comparison of the Altaic Languages with Japanese' *Transactions of the Asiatic Society of Japan. (Second Series)* 7, 41–54
- Ramstedt, Gustaf J. (1952–1957). *Einführung in die altaische Sprachwissenschaft I: Lautlehre; II: Formenlehre* [Introduction in Altaic Linguistics. I: Phonology, II: Morphology]. Helsinki: Suomalai-Ugrilainen Seura.
- Robbeets, Martine (2005). *Is Japanese related to Korean, Tungusic, Mongolic and Turkic?* Wiesbaden: Harrassowitz.
- Robbeets, Martine (2015). *Diachrony of verb morphology. Japanese and the Transeurasian languages*. (Trends in Linguistics 291.) Berlin: De Gruyter Mouton
- Robbeets, Martine, and Remco Bouckaert (2018). 'Bayesian Phylolinguistics reveals the Internal Structure of the Transeurasian Family', *Journal of Linguistic Evolution* 3(2): 145–162.
- Ross, Malcolm (2014) 'Affixes in language change. Copying, inheritance and accommodation. Review Essay of Johanson and Robbeets (eds). 2012: 'Copies Versus Cognates in Bound Morphology' *Language dynamics and change* 4: 271–284.
- Rozycki, William (2006). 'Review of Robbeets, Martine 2005. Is Japanese related to Korean, Tungusic, Mongolic and Turkic?', *Mongolian Studies* 28: 114–115.
- Starostin, Sergei, Anna Dybo, and Oleg Mudrak (2003). *Etymological Dictionary of the Altaic Languages*, I–III. Leiden: Brill.
- Street, John Charles (1962). 'Review of Nikolas Poppe, Vergleichende Grammatik der altaischen Sprachen, Teil I', *Language* 38: 92–98.
- Street, John Charles (1978). *Altaic Elements in Old Japanese, part 2*. Madison: Manuscript.
- Tekin, Talat (1994). 'Altaic Languages', in R. E. Asher (ed.). *The Encyclopedia of Language and Linguistics. Vol. 1*. Oxford: Pergamon Press, 82–85.
- Vladimirtsov, Boris J. (1929). *Sravnitel'naja grammatika mongol'skogo pišmennogo jazyka i xalxaskogo narečija. Vvedenie i fonetika* [A comparative grammar of Written Mongolian and Khalkha Mongolian. An introduction. Phonology]. Leningrad: Izd. Leningradskogo Vostočnogo Instituta im. Enukidze/Moscow: Nauka.
- Vovin, Alexander (2001). 'Japanese, Korean and Tungusic. Evidence for genetic relationship from verbal morphology', in David B. Honey and David C. Wright (eds), *Altaic Affinities*

(Proceedings of the 40th meeting of the PIAC, Provo, Utah 1997). Indiana University:
Research Institute for Inner Asian Studies, 183–202.

Vovin, Alexander (2005). 'The End of the Altaic Controversy (In Memory of Gerhard Doerfer)', *Central Asiatic Journal* 49: 71–132.